



# Stratospheric gravity-waves over the mountainous island of South Georgia: testing a high-resolution dynamical model with 3-D satellite observations and radiosondes

Neil P. Hindley[1,2], Corwin J. Wright[1], Alan M. Gadian[2], Lars Hoffmann[3], John K. Hughes[2], David R. Jackson[4], John C. King[5], Nicholas J. Mitchell[1], Tracy Moffat-Griffin[5], Andrew C. Moss[1], Simon B. Vosper[4], and Andrew N. Ross[2]

[1]Centre for Space, Atmospheric and Oceanic Science, University of Bath, Bath, UK
[2]School of Earth and Environment, University of Leeds, Leeds, UK
[3]Jülich Supercomputing Centre, Forschungszentrum Jülich, Jülich, Germany
[4]Met Office, Exeter, UK
[5]Atmosphere, Ice and Climate Group, British Antarctic Survey, Cambridge, UK

**Correspondence:** Neil Hindley
(n.hindley@bath.ac.uk)

**Abstract.** Atmospheric gravity waves are key drivers of the transfer of energy and momentum between the layers of the Earth's atmosphere. The accurate representation of these waves in General Circulation Models (GCMs) however has proved very challenging. This is because large parts of the gravity wave spectrum are at scales that are near or below the resolution of global GCMs. This is especially relevant for small isolated mountainous islands such as South Georgia (54°S, 36°W) in the Southern Ocean. Observations reveal the island to be an intense source of stratospheric gravity waves, but their momentum fluxes can be under-represented in global models due to its small size. This is a crucial limitation, since the inadequate representation of gravity waves near 60°S during winter has been linked to the long-standing "cold-pole problem", where the southern stratospheric polar vortex breaks up too late in spring by several weeks. Here we address a fundamental question: when a model is allowed to run at very high spatial resolution over South Georgia, how realistic are the simulated gravity waves compared to observations? To answer this question, we present a 3-D comparison between satellite gravity wave observations and a high resolution model over South Georgia. We use a dedicated high-resolution run (1.5 km horizontal grid, 118 vertical levels) of the Met Office Unified Model over South Georgia and coincident 3-D satellite observations from NASA AIRS/Aqua during July 2013 and June-July 2015. First, model winds are validated with coincident radiosonde observations. The AIRS observational filter is then applied to the model output to make the two data sets comparable. A 3-D $S$-transform method is used to measure gravity-wave amplitudes, wavelengths, directional momentum fluxes and intermittency in the model and observations. Our results show that although the timing of gravity wave activity in the model closely matches observations, area-averaged momentum fluxes are generally up to around 25% lower than observed. Further, we find that 72% of the total flux in the model region is located downwind of the island, compared to only 57% in the AIRS measurements. Directly over the island, the model exhibits higher individual flux measurements but these fluxes are more intermittent than in observations, with 90% of the total flux carried by just 22% of wave events, compared to 32% for AIRS. Observed gravity wave fluxes





also appear to dissipate more quickly with increasing height than in the model, suggesting a greater role for wave-mean flow interactions in reality. Finally, spectral analysis of the wave fields suggests that the model over-estimates gravity wave fluxes at short horizontal scales directly over the island, but under-estimates fluxes from larger horizontal scale non-orographic waves in the region, leading to a lower average value overall. Our results indicate that, although increasing model resolution is important, it is also important to ensure that variability in the background wind vector and role of non-orographic waves are accurately simulated in order to achieve realistic gravity wave activity over the Southern Ocean in future GCMs.

## 1 Introduction

Atmospheric gravity waves (GWs) are a key driver of the atmospheric circulation. These waves a play key role in many important chemical and dynamical processes throughout the atmosphere via the transportation of energy and momentum between atmospheric layers and across great distances (e.g. Fritts and Alexander, 2003; Fritts et al., 2006; Alexander et al., 2010). Gravity waves carry a flux of horizontal pseudo-momentum which is deposited when they break or dissipate, resulting in a drag or driving force on the background atmospheric flow. The large-scale effects of this forcing have significant impacts throughout the whole stratosphere and mesosphere.

Global circulation models (GCMs) used for numerical weather and climate forecasting must therefore include the effects of gravity waves. Indeed, it is now recognised GCMs must have a well-resolved stratosphere that includes realistic dynamics, including resolved gravity waves, in order to deliver accurate seasonal weather forecasts, predict long-term climate change and predict the future of the ozone layer (e.g. Baldwin et al., 2018).

However, gravity waves are notoriously difficult to represent, even in state-of-the-art numerical models. One reason for this is that they have a very wide spectrum of physical scale sizes, ranging from hundreds of metres to a few tens of kilometres in the vertical and from tens to many hundreds of kilometres in the horizontal. These short vertical and horizontal scales are too small to be resolved in GCMs so their effects are instead represented by parameterisations, where a momentum-forcing term is applied the background flow (e.g. Warner and McIntyre, 1996).

However, these parameterisations remain poorly constrained by observations. In some cases, there is a risk that gravity wave parameterisations are so poorly constrained that they begin to take on a role as simply a tuning parameter that is adjusted in order to reproduce realistic large-scale dynamics in reanalyses, such as zonal-mean zonal winds and a realistic quasi-biennial oscillation (QBO) (Alexander et al., 2010; Wright and Hindley, 2018), rather than being adjusted to match observational measurements.

At larger gravity wave scales of hundreds of kilometres, current operational GCMs can resolve waves directly in the stratosphere. Given the advances in computer power, GCMs are likely to operate at ever-finer resolution in the coming years, which will enable them to resolve more and more of the gravity wave spectrum. Indeed, offline high-resolution simulations are already being used to help tune gravity wave parameterisations in operational GCMs (e.g. Vosper, 2015; Vosper et al., 2016, 2020). A question then arises, as posed by Preusse et al. (2014): in the future, will ever higher spatial resolution in GCMs remove the need for gravity wave parameterisations altogether?





This question is especially significant for small, isolated mountainous islands whose physical scales of a few tens of kilometres are at or near the spatial grid size of current GCMs. In terms of orographic drag parameterisations, these islands lie in the "grey zone", where they are neither fully resolved nor fully subgrid (Vosper et al., 2016). As such, gravity wave generation by flow over small mountainous islands may be significantly underestimated in GCMs. Several recent studies have suggested that under-represented gravity wave momentum flux from isolated islands in the Southern Ocean may be a significant contributing factor to the wintertime "cold-pole problem", one of the largest and most long-standing biases common to virtually all major weather and climate models (Scaife et al., 2002; Butchart et al., 2011; McLandress et al., 2012; Alexander and Grimsdell, 2013; Garfinkel and Oman, 2018). This is in addition to significant non-orographic wave activity in the 60°S belt (Hendricks et al., 2014; Hindley et al., 2015, 2019).

The cold-pole problem refers to a simulated wintertime stratospheric polar vortex that, when compared to observations, is too cold by around 5 to 10 K, has winds that are too strong by around 10 ms[-1] and persists for some two to three weeks too long into spring before breaking up (e.g. Butchart et al., 2011). This bias causes difficulty in simulating systems such as the stratospheric ozone cycle (e.g. Garcia et al., 2017), global chemical transport (e.g. McLandress et al., 2012) and surface climate change in the Antarctic (Thompson et al., 2011). Poor simulation of GWs in this region has other important implications. For example, GWs play an important role in the dynamical balance of the stratospheric jet (Choi and Chun, 2013), which in turn affects surface storm track locations at mid- and high latitudes (Perlwitz, 2011). Gravity wave modulation of background winds can also affect the formation of polar stratospheric clouds (Höpfner et al., 2006), which can significantly reduce ozone concentrations, a principal driver of Antarctic climate change (McLandress et al., 2011; Garcia et al., 2017). There is thus a critical need to establish the sources, fluxes and variability of GWs from small islands near 60°S in order to determine their contribution to this region and so guide the development of GCMs.

Regarding the question posed by Preusse et al. (2014), it is possible that if models are run at sufficiently high spatial resolution, gravity wave fluxes from small mountainous islands will be accurately represented and the cold-pole bias may be reduced. This raises a follow-on question, which is the focus of our study: when a model is allowed to run at very high spatial resolution over a small mountainous island in the Southern Ocean, how realistic are the simulated gravity waves compared to observations?

In this study, we address this question for one such island: South Georgia (54°S, 36°W). South Georgia is a classic example of an isolated mountainous island in the Southern Ocean that experiences intense surface wind conditions during winter, so is an ideal subject for our study. The island is approximately 170 km long, 25 km wide and lies more than 600 km from any other major islands and more than 2000 km from the nearest continent of South America. The island is entirely mountainous and has interior peaks that reach heights of 3000 m. The sharpness and orientation of the topography relative to the prevailing wind, combined with the great distance from other potential orographic sources, make the island a ideal natural laboratory for orographic gravity wave generation. Previous studies of gravity waves over South Georgia and other small islands in the Southern Ocean that have been conducted using a diverse range of observations and models (e.g. Alexander et al., 2009; Hoffmann et al., 2013; Vosper, 2015; Vosper et al., 2016; Hoffmann et al., 2016; Moffat-Griffin et al., 2017; Garfinkel and





Oman, 2018; Jackson et al., 2018; Hindley et al., 2019) have revealed intense gravity wave activity at a range of scale sizes in the troposphere and middle atmosphere.

Here, a dedicated high-resolution local area configuration of the UK Met Office Unified Model (1.5 km grid, 118 vertical levels) is run for the region around South Georgia. No gravity wave parameterisations are applied in this local area model, so the dynamics are essentially free-running. In this study use data from two runs: one during July 2013 and another during June-July 2015 using model configurations as described by Vosper (2015) and Jackson et al. (2018) respectively. Gravity waves in these simulations are compared to co-located 3-D satellite observations from NASA AIRS/Aqua. A specialised 3-D temperature

retrieval for AIRS is used that has superior spatial resolution over the standard AIRS product. After validating the model wind with co-located radiosonde observations, the observational filter of the AIRS retrieval is applied to the model output so that we can compare the observed and simulated wave fields fairly. This novel approach allows us to address the question of how realistic the simulated gravity waves in the high-resolution model are when compared to observations.

     In Sect. 2 we describe the model, satellite and radiosonde datasets that we use in this study. We also validate the model winds

with the co-located radiosonde observations here to ensure they are realistic. In Sect. 3 we apply the AIRS observational filter to the model output fields so that we can compare gravity waves in the two data sets fairly and we apply a 3-D $S$-transform analysis method for measuring gravity wave properties. In Sect. 4 we present simulated and observed wave amplitudes, wavelengths and directional momentum fluxes over South Georgia during July 2013 and June-July 2015. We investigate the distribution and intermittency of these gravity wave properties with respect to time, altitude, distance from the island and spectral properties.

These results are then discussed in Sect. 5, and our conclusions are presented in Sect. 6.

## 2   Data

Three atmospheric datasets over South Georgia are analysed in this study: 1) 3-D satellite observations from AIRS/Aqua; 2) modelling simulations in a local-area domain centred on the island; and 3) radiosonde observations launched from King Edward Point (KEP) during June-July 2015. Comparisons of the satellite observations and modelling simulations make up the

main results of this study, while the radiosonde observations are used to validate the modelling simulations.

     The geographical locations and spatial extent of the three datasets are shown in Fig. 1. South Georgia is located near the centre of Fig. 1a, lying around 2000 km east of South America and the Antarctic Peninsula in the Southern Ocean. The 1200 km × 900 km local-area domain of the modelling simulations is shown by the light blue box, while the two red-and-white dashed boxes show two overpasses of the AIRS instrument (one during ascending and one during a descending node) over the

area. The exact location of each of the overpasses varies with each orbit, as discussed below. Figures 1b and 1c show 3-D views of these domains. Also shown are the trajectories of radiosonde balloons launched from KEP on South Georgia during January (green) and June-July (orange) 2015. Note that the June-July radiosondes travelled much further downwind due to stronger stratospheric zonal winds during austral winter, and many of these travelled so far east that they exited the local area model domain.





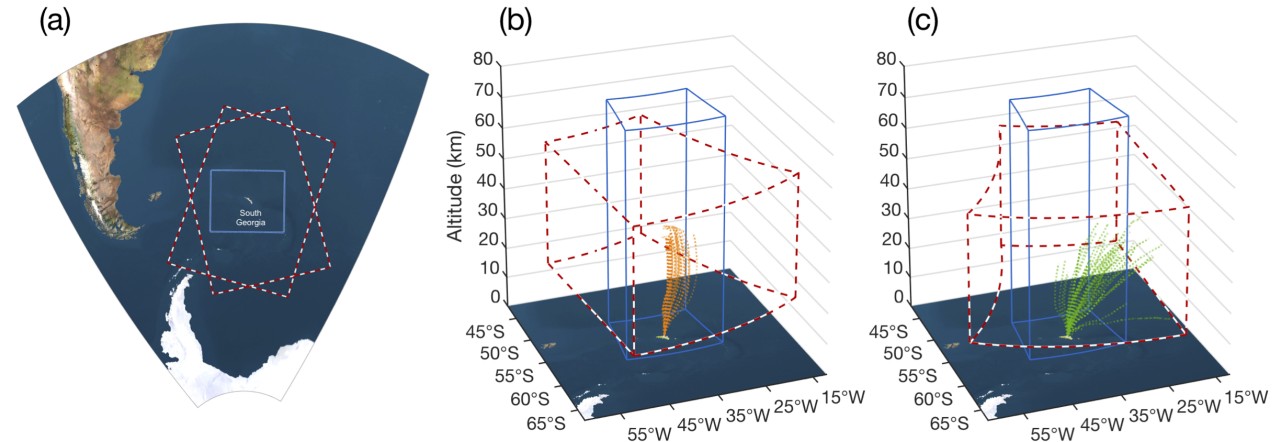

**Figure 1.** Maps showing the horizontal and vertical extent of the local area model (blue lines) around the island of South Georgia and two typical examples of satellite observations from AIRS (red and white dashed lines). Panel (a) shows a map of the local region around South Georgia, plotted on a regular distance grid. Panels (b) and (c) show the vertical extent of the model on a latitude-longitude grid. The vertical extent usable temperature data from the 3-D AIRS retrieval scheme of Hoffmann and Alexander (2009) is shown in red dashed lines for both an ascending (b) and descending (b) overpass. Orange (green) lines show the trajectories of radiosondes launched from the island during a summer (winter) campaign in January (June-July) 2015.

## 2.1 AIRS satellite observations

We use satellite data from the Atmospheric Infrared Sounder (AIRS) on NASA/Aqua (Aumann et al., 2003). Aqua has a $\sim$100-minute near-polar sun-synchronous orbit, with an ascending-node equator-crossing local solar time of 1:30pm. It has been collecting data since August 2002, with only minor interruptions since that date.

AIRS is a nadir-sensing instrument that makes measurements in the across-track direction at scan angles between $\pm 49.5°$ from the nadir. Radiances in 2378 spectral channels are measured in a continuous 90-element, $\sim 1800$ km-wide swath along the scan track. The horizontal spacing of these elements varying from around $13.5\,\mathrm{km} \times 13.5\,\mathrm{km}$ at nadir to $41\,\mathrm{km} \times 21.4\,\mathrm{km}$ at track-edge. In the along-track direction, the scan track is split into arbitrary 135-element along-track sections, referred to as granules, which correspond to 6 minutes of data collection. The spatial extent of these 135-element granules is shown in Fig. 1a.

In this study we use 3-D temperature measurements from AIRS observations derived using the retrieval scheme described by Hoffmann and Alexander (2009). This retrieval uses multiple 4.3 and $15\,\mu$m $CO_2$ spectral channels to produce estimates of stratospheric temperature that have a significantly higher vertical resolution than can be achieved using single channel radiances. Retrievals are carried out for each individual satellite footprint independently, improving the horizontal resolution of this retrieval by a factor of 3 in the along and across-track directions compared with AIRS operational data. Temperatures



are retrieved on a 3 km vertical grid at a vertical resolution that varies between $7 - 14$ km in the stratosphere (Hindley et al., 2019, their Fig. 2), compared to only $\gtrsim 14$ km for single-channel operational AIRS data at 41 km altitude. This makes the 3-D temperature retrieval well suited to the study of small-scale processes such as gravity waves. At the latitudes and altitudes studied here, uncertainty in temperature measurement is typically $\lesssim 1.5$ K (Hoffmann and Alexander, 2009; Hindley et al., 2019). Validation of the retrievals is described by Hoffmann and Alexander (2009) and Meyer and Hoffmann (2014).

To extract gravity-wave temperature perturbations, a 4th-order polynomial fit is subtracted from each across-track scan to remove slowly-varying background signals due to large-scale temperature gradients or planetary wave activity. This approach has been widely used in previous work (e.g. Wu, 2004; Alexander and Barnet, 2007; Hoffmann et al., 2014; Wright et al., 2017; Hindley et al., 2019), although we acknowledge that some artefacts may remain, its use here ensures consistency with previous studies.

The sensitivity of our final AIRS temperature perturbations product to GWs is thus defined by the combination of both the retrieval averaging kernels and the detrending method. The full processing chain results in data that are sensitive to GWs with wavelengths longer than about 8 to 9 km in the vertical. In the horizontal, the sensitivity cutoff for short horizontal wavelengths dependent on the footprint size, which varies between roughly 15 km at nadir and 40 km for the outermost tracks. For longer horizontal wavelengths, sensitivity drops below 90% at horizontal wavelengths of 730 km and below 10% at 1400 km; thus,
longer horizontal wavelengths will be more strongly attenuated than short ones, modifying the observed spectrum. The vertical and horizontal resolutions of the 3-D AIRS retrieval for different atmospheric conditions and sensitivity to gravity waves of varying scales can be seen in Fig. 2 of Hindley et al. (2019), Fig. 5 of Hoffmann et al. (2014) and the supplementary material of Ern et al. (2017).

There are typically two AIRS/Aqua overpasses per day over South Georgia. However, due to the precession of the Aqua
orbit relative to the Earth's surface, the extent of the AIRS overpass swaths do not always cover the same regions every day. For comparison with the model domain around South Georgia, we only select AIRS overpasses where at least three out of four of the corners of the model domain are contained within the AIRS measurement swath during a given overpass, as shown in Fig. 1a. Due to the high inclination of the Aqua orbit, this usually results in 80 to 90% of the model domain being covered, and the area directly over the island itself is usually measured twice per day.

The two daily overpasses that meet these criteria usually occur at around 0300 UTC and 1700 UTC with the offset in timing usually less than 20 minutes from these times. However, there were several occasions where the AIRS swath at these times did not meet the three-corner criterion for the model domain that we set above, so we do not include these overpasses in our analysis.

## 2.2 Numerical modelling: local-area simulations over South Georgia

In this study we use model output from specialised high-resolution runs of the UK Met Office Unified Model using the Even Newer Dynamics for General Atmospheric Modelling of the Environment (ENDGame) dynamical core (Davies et al., 2005; Wood et al., 2014). The model is configured in a high-resolution local-area domain 1200 km × 900 km around the island of South Georgia, with lateral boundary conditions supplied by a lower-resolution global forecast described below.





The nested local-area domain simulation consists of an $800\times600$-pixel latitude-longitude grid centred at 54.5°S, 37.1°W,
with 118 vertical levels from the surface to altitudes near 80 km. The simulations are run in a rotated-pole coordinate frame
in order to provide latitude-longitude spacing that is close to Cartesian. This grid gives a horizontal spacing of roughly
1.5 km × 1.5 km, which is significantly finer than the operational resolution of the Met Office global model. In the vertical,
a damping layer is applied above 58.5 km altitude to suppress reflection effects.

Grid spacing between the model vertical levels increases from around 10 m near the surface to 3 km at 75 km altitude. There
are 34 vertical model levels between altitudes of 20 to 60 km, where the AIRS measurements are most reliable. In this altitude
range the model grid spacing increases from 0.6 km at 20 km altitude to 2.1 km at 60 km altitude. This spacing is much finer
than the 3 km vertical grid of the AIRS retrieval, so we expect that gravity waves at vertical scales that are visible to AIRS to
be well simulated in the model.

Meteorological boundary conditions for the local-area domain are provided by a global N512 simulation with 70 vertical
levels from the surface to altitudes near 80 km. At latitudes near South Georgia, this global model has a horizontal grid spacing
of $\Delta x \approx 46$ km. This simulation is provided by Met Office operational analyses and re-initialised every 24 hours, providing
hourly forecasts that supply lateral boundary conditions for the local-area configuration over South Georgia. At the edges
of the local-area domain, these hourly forecasts are linearly interpolated in time to each model time step. No gravity wave
parameterisations were included in the local-area simulations, so the resolved wind and temperature fields are effectively free-
running. Output fields were archived hourly. More information on the configuration of these simulations is described in detail
in Vosper (2015); Vosper et al. (2016).

In this study we analyse data from two runs during austral winter, one covering the period 1st to 31st July 2013 and another
for the period 11th June to 8th July 2015. A third model run for January 2015 was also analysed but, due to the weaker
stratospheric winds during austral summer, very few gravity waves with vertical wavelengths long enough to be resolved by
AIRS are found. Because of this, a comparison is not included here. Both model simulations during 2015 were designed to
coincide with a summer and winter radiosonde campaigns on South Georgia (Moffat-Griffin et al., 2017; Jackson et al., 2018)
described below.

The high spatial and temporal resolution of the local-area domain, coupled with the application of daily boundary conditions
from Met Office operational forecasts should result in simulated conditions over South Georgia that are very close to reality
for the given time periods. This presents an ideal opportunity to compare the simulated wave fields with co-located satellite
measurements from AIRS.

To extract gravity wave temperature perturbations from the simulations, two runs were performed in parallel for each time
period. In one run, the high-resolution topography for the island is included as normal (we refer to this as the SG configuration),
while in the other run this topography is flattened to mean sea level (referred to as the nSG configuration). In previous studies,
such as Vosper (2015); Vosper et al. (2016), output temperature fields from two runs such as these are differenced (SG−nSG)
in order to reveal orographic gravity wave perturbations as a result of flow over the mountains. However, this approach will
only reveal orographic gravity wave activity. This is because non-orographic wave activity will exist in both the SG and nSG





runs and will thus be removed when the difference is taken. The AIRS observations contain gravity wave perturbations from both orographic and non-orographic waves, so this is not desirable for our comparison.

Therefore, in order to compare simulated gravity wave perturbations from both orographic and non-orographic waves in the model, non-orographic gravity wave activity in the nSG configuration must be removed before the difference is taken. To do this, we apply a 2-D horizontal smoothing filter to the output temperature fields of the nSG configuration. We found that a Gaussian smoothing filter with full-width-at-half-maximum (FWHM) equal to $300 \, \text{km} \times 225 \, \text{km}$ (one quarter of the model domain in each dimension) was sufficient to produce a broad, featureless background temperature field. This smoothed version

of the nSG configuration is then subtracted from the SG configuration to reveal temperature perturbations from both orographic and non-orographic wave activity. This process was only applied to altitudes above 15 km because it was found that artefacts were introduced in temperature perturbations due to synoptic systems in the troposphere. Since we only compare the model to AIRS observations above altitudes of 20 km, this does not affect our results significantly.

### 2.3   Radiosondes

In this study we also make use of wind measurements from a radiosonde campaign on South Georgia during June-July (austral winter) 2015. The details of this campaign and further analysis of these data are described by Moffat-Griffin et al. (2017), together with data from a campaign in austral summer during January 2015, the trajectories of which are shown by the orange and green lines in Fig. 1b and 1c. Although both campaigns were analysed in this study, only data from the wintertime (June-July) campaign is shown due to the much lower levels of stratospheric gravity wave activity observed during austral summer.

Balloons were launched twice-daily from the British Antarctic Survey base at King Edward Point (54.3°S, 37.5°W), equipped with Vaisala RS92-SGP radiosondes, with additional launches timed to coincide with AIRS overpasses or when forecasts predicted strong winds suitable for GW generation. Meteorological and geolocation parameters are recorded at 2-second intervals during the flight.

    54 balloons were successfully launched during the wintertime period 13th June to 6th July. Due to challenging local envi-

ronmental conditions, 10 launches failed to reach the tropopause and only 20 reached altitudes of 25 km or above. It can be seen in Fig. 1 that the balloons travelled much further downwind to the east during winter due to the much stronger prevailing winds, with several balloons exiting the eastern boundary of model domain before bursting.

### 2.4   Validation of modelled winds using co-located radiosonde measurements

Before we compare our simulated gravity-wave fields to satellite observations, it is first prudent to validate the model wind

fields against the co-located radiosonde observations. Since wind flow over orography is likely to be the dominant driver of gravity-wave activity directly over the island (e.g Alexander and Grimsdell, 2013; Vosper, 2015; Moffat-Griffin et al., 2017; Jackson et al., 2018), model winds should first be tested to ensure they are a fair representation of reality.

    Although the boundary conditions of the local-area model are initialised daily by Met Office operational analyses, these winds are poorly constrained by conventional observations over the Southern Ocean, relying largely on temperatures nudged

by assimilated satellite radiances. Wright and Hindley (2018) showed that a lack of observations can result in significant strato-

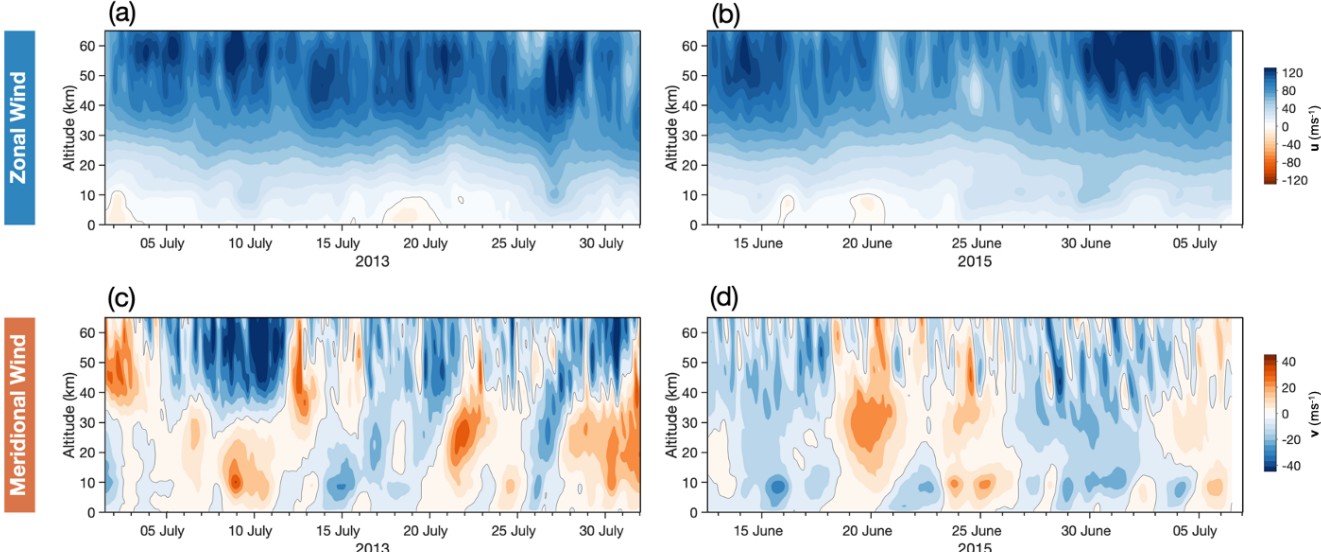

**Figure 2.** Hourly zonal and meridional wind speeds against altitude in the local area model over South Georgia during July 2013 (a,c) and June-July 2015 (b,d). Positive (negative) values indicate eastward (westward) and northward (southward) directions in the zonal and meridional winds respectively.

spheric biases in this region in global models. Thus, these radiosonde observations represent the only coincident measurements available with which to accurately assess the low-altitude wind fields in the model over the island during our period of study.

Figure 2 shows hourly zonal and meridional wind against height for the two model runs during July 2013 and June-July 2015. These values are horizontally averaged over the whole model domain, so are representative of the large-scale background flow.

As would be expected for a wintertime study at these latitudes, wind speeds in the zonal direction are eastward and generally increase strongly with height, with values reaching $120\,\mathrm{ms^{-1}}$ above $50\,\mathrm{km}$ altitude. Variability in zonal wind speed in the stratosphere is closely related to the changing latitude of the centre of the stratospheric polar vortex during winter. In the meridional direction, frequent changes between northward and southward flow are observed, with speeds reaching values near $\pm40\,\mathrm{ms^{-1}}$ above $40\,\mathrm{km}$ altitude. In one case near 10th July 2013, a wind reversal with height is seen at around $30\,\mathrm{km}$.

To compare these wind fields to radiosonde observations, each radiosonde trajectory is traced through the hourly model winds fields. The modelled wind speed along this trajectory is compared to the observed radiosonde winds. To do this, all model timesteps are loaded for the duration of each radiosonde flight, including one timestep before and after, and 4-dimensional linear interpolants $(x,y,z,t)$ of zonal $u$ and meridional $v$ wind fields are constructed. These interpolants are then evaluated for each point along the radiosonde's trajectory using the measured time, height and location information. This approach allows us to 250 compensate for any time-varying effects in the model wind speeds during the radiosonde flights.

It is important to mention at this point that caution should be taken when measuring gravity wave momentum fluxes from slanted vertical profiles through mountain wave fields (such as radiosonde measurements here). As discussed by Vosper

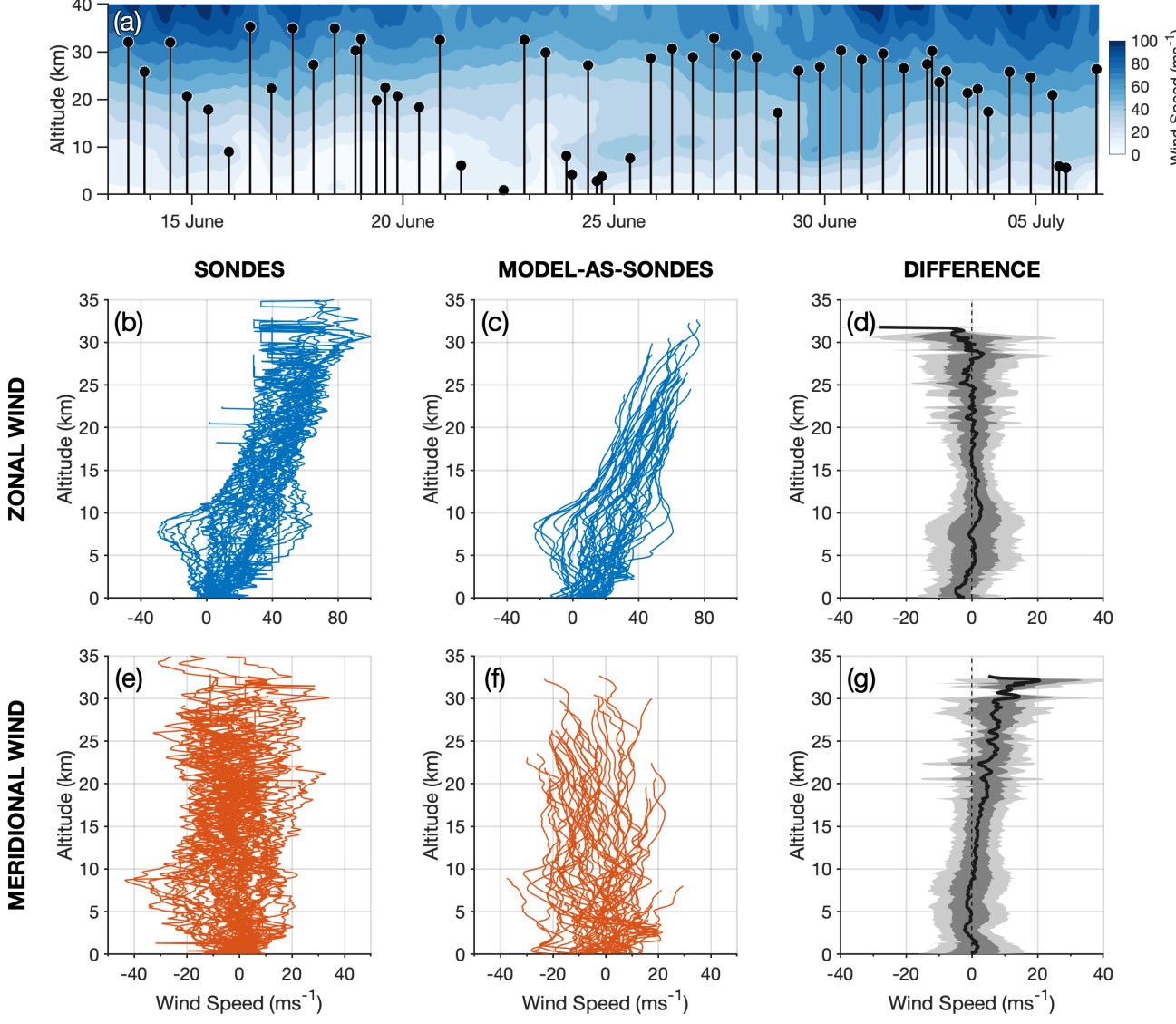

**Figure 3.** Comparison of observed and simulated wind speeds during June-July 2015 from radiosonde observations and the local area model over South Georgia. Panel (a) shows the absolute model wind speed against height, with launch times and maximum altitudes of the radiosonde observations overlaid in black. Profiles of zonal (blue) and meridional (orange) wind against height for the radiosonde measurements and the model wind evaluated along each sonde trajectory are shown in panels (b,e) and (c,f) respectively. Thick black lines in panels (d,g) show the mean difference (Sondes − Model-as-Sondes) between the observed and modelled wind speeds for each height, with dark and light grey shading indicating one and two standard deviations respectively.





and Ross (2020), the usual assumptions required for the measurement of vertically-integrated momentum fluxes of planar monochromatic waves do not hold true for mountain waves sampled with a slanted vertical profile. For this reason we do
not conduct a gravity wave comparison between the model and the radiosonde measurements here and instead only use the radiosonde measurements to validate the model winds.

Figure 3a shows the results of our wind comparison. Radiosonde launch times (UTC) and maximum recorded altitudes during the winter campaign are shown by the black lines and circles in Fig. 3a. Also shown in this panel is mean zonal wind speed over the modelling domain against altitude, which gives us an indication of the local wind conditions through which the
balloons travelled.

As can be seen in Fig. 3a, several of the radiosonde balloons did not reach their desired altitudes near 30 km, instead bursting soon after launch. This was usually due to the extreme weather conditions at low altitudes during the fieldwork campaign, as reported by the radiosonde launch team. In some cases, surface winds were so strong that radiosonde balloons did not ascend fast enough to exit the bay around the launch site, colliding instead with the slopes of nearby mountains.

Panels (b-g) in Fig. 3 show the results of this analysis, where "Sondes" indicates the measured radiosonde wind speed and "Model-as-Sondes" indicates the modelled wind speed evaluated along the radiosondes path. The two data sets are in good general agreement, with measured and simulated zonal winds in Figs. 3(b-c) increasing from a few metres per second near the surface to around $60\,\text{ms}^{-1}$ around 30 km altitude. In the meridional direction, both data sets show wind speeds between around $\pm 15\,\text{ms}^{-1}$ with little variation in altitude in Figs. 3(e-f). In both directions, the radiosonde measurements are found to exhibit
more small-scale variability than the model fields, likely due to small-scale wave or turbulence features and measurement errors which are not present in the model. Measurement artefacts from balloon bursts are also visible, in addition to instances where sonde measurements are shown but no model-as-sonde data is available due the balloons exiting the model domain.

To further compare the simulated and measured wind speeds, the difference between the sonde and the model-as-sonde winds (sondes minus model-as-sondes) against altitude is shown in Figs. 3(d) and (g) for the zonal and meridional directions
respectively. Dark and light grey shaded regions show one and two standard deviations of all differences respectively, while the thick black line shows the mean difference for the June-July 2015 run.

In the zonal direction, the campaign-mean difference in wind speed is reasonably close to zero for most altitudes, and lowest in the low to mid-stratosphere. Largest differences between the sonde and model-as-sonde winds are seen between altitudes of 3 to 10 km in Fig. 3d. This is near the tropopause region, and could suggest significant short-timescale variability
of the tropospheric jet observed over the island which is not well represented in the model. This may influence the upward propagation of mountain waves. Near the surface, a slight bias towards stronger zonal winds in the model in observed, which could be due to local topographic effects around King Edward Point.

In the meridional direction, wind speed differences between the sondes and model-as-sondes are generally less than $10\,\text{ms}^{-1}$. However, a clear positive difference is observed above around 15 km altitude which increases to near $10\,\text{ms}^{-1}$ around 30 km
altitude. This positive difference corresponds to slight southward directional bias in the simulated model winds. That is, the meridional components of the sonde-measured winds are greater (more northward) than the meridional components of the model-as-sonde winds. This could be indicative of a small directional bias in the global forecast that provides the boundary



conditions for the local area model over South Georgia. At these high southern latitudes, global models are poorly constrained by ground-based conventional measurements, so a small bias is perhaps could be expected. While we do not expect this bias
to affect our results significantly, we acknowledge that a difference in the rotation of the simulated wind vector with increasing altitude compared to reality could have an effect on the propagation and measured structure of any mountain wave field that forms over the island.

It should be mentioned that some of the differences between the model and model-as-sonde winds could be due to timing or lag issues in the model, such as in the arrival of synoptic systems. Anecdotal reports from the radiosonde launch team on South
Georgia suggested that the arrival of synoptic systems such as fronts and weather systems could differ from the Unified Model forecast by several hours. Although these phenomena are generally located in the troposphere, they may have a stratospheric response which may be earlier or later than predicted. These would manifest as pseudo-random errors in our analysis, which could explain some of the spread in the wind speed differences. Further, positional errors in the radiosonde measurements could lead to further spread, but these factors are unlikely to lead to the systematic biases reported here.

Aside from these minor differences we conclude that, on a climatological level, the model wind speed and direction over the island is reasonably well simulated during the June-July 2015 campaign. Large differences in surface wind speed, for example, may also be indicative of difficulty in simulating the local topographic environment of the launch site rather than simulating the island as a whole. The British Antarctic Survey base at King Edward Point, from which the balloons were launched, is located in a sheltered bay 2 km east of the main mountain ridge of the Thatcher Peninsula, which is nearly 2 km high. At the
1.5 km model horizontal resolution used in this study, this mountain ridge will be at most one model grid cell away from the launch site. Thus, accurately simulating surface winds at this site will be quite challenging.

## 3   Comparing gravity wave characteristics in the model to 3-D AIRS measurements

In order to make a fair comparison between gravity waves in the model and in the 3-D satellite observations, we must first ensure that the spectral range of gravity waves in each data sets is comparable. This involves re-gridding the model and observations
onto a common regular grid and applying the AIRS observational filter to the model output.

The high resolution model outputs hourly temperature fields on the model grid for the July 2013 and June-July 2015 modelling campaigns. During these campaigns, 39 and 48 3-D AIRS measurements over the island region for 2013 and 2015 respectively were found to meet our selection criteria in Sect. 2.1, giving 87 co-incident AIRS measurements in total.

In this section, we first describe how the AIRS measurements and the model output are re-gridded onto a common regular
grid. The AIRS observational filter is then applied to the model to compare the simulated wave field as if it were observed by AIRS. We then apply a 3-D Stockwell transform (3DST) to the AIRS measurements, the model output and the model at the AIRS resolution to measure gravity wave amplitudes, wavelengths, directions and momentum fluxes in each dataset.





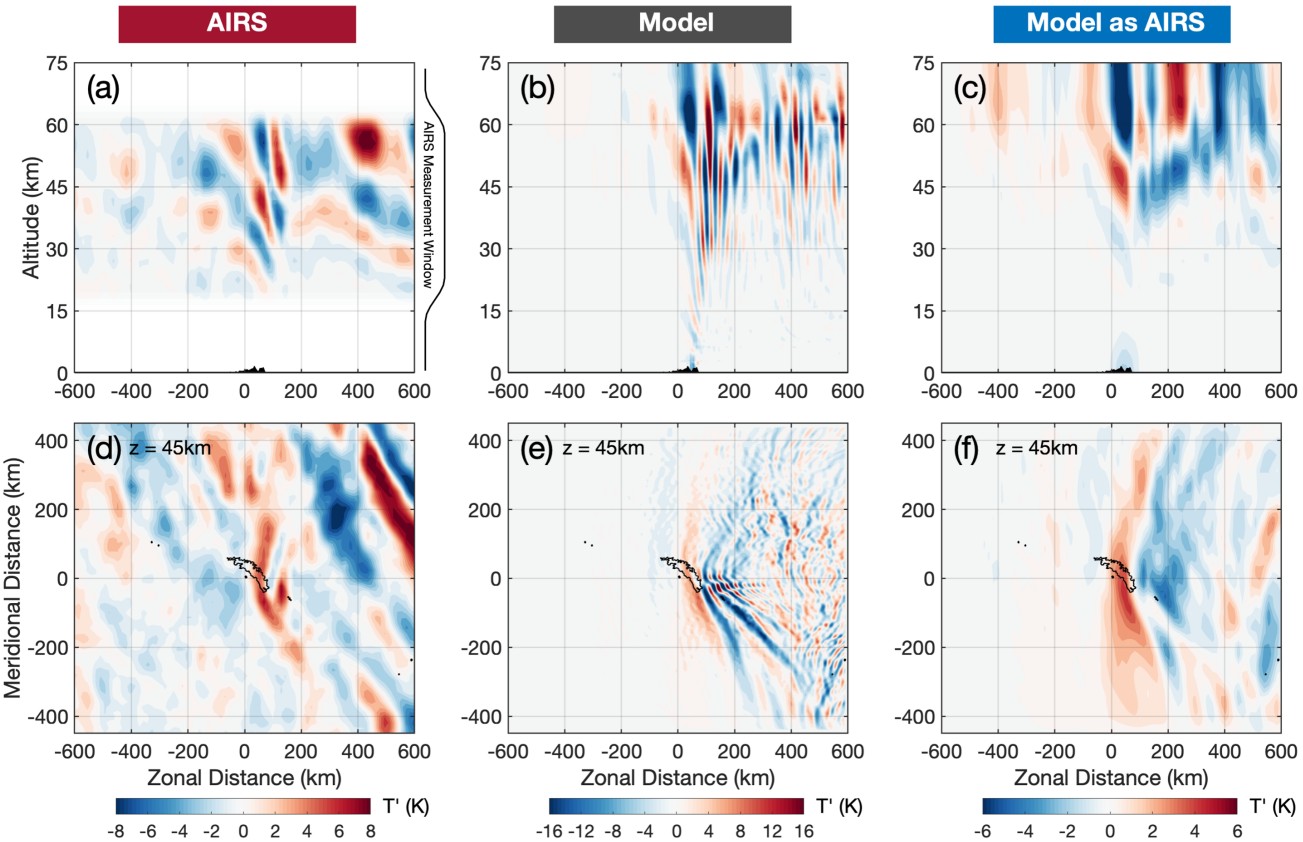

**Figure 4.** Vertical (top row) and horizontal (bottom row) cross-sections of observed and modelled temperature perturbations over South Georgia at 0300 UTC on the 5th of July 2015 for the regridded AIRS measurements, the model and the model-as-AIRS. Vertical cross-sections are taken at a meridional distance of $y = 0$ km and horizontal cross-sections are shown at an altitude of 45 km. Here, model-as-AIRS refers to the model temperature perturbations with the AIRS observational filter applied. Different colour scales are used for the model, AIRS and model-as-AIRS so that wave structure can be seen clearly in each data set. See text in Sect. 3.1 for details.

## 3.1 Applying the observational filter of AIRS to the model output

A key concept in the study of GWs is the observational filter. That is, no single instrument can measure the full gravity wave
spectrum (e.g. Preusse et al., 2002; Alexander and Barnet, 2007). For example, a nadir-sounding instrument such as AIRS will have, in general, relatively low vertical resolution but relatively high horizontal resolution. In contrast, limb-sounding instruments and techniques such as HIRDLS (e.g. Gille et al., 2003) or GPS radio occultation (e.g. Kursinski et al., 1997) will have relatively high vertical resolution but relative low horizontal resolution.

     Thus, in the present study, a simple direct comparison of gravity wave properties between the full resolution model and the
AIRS measurements is not particularly meaningful due to their different observational filters. To obtain a fair and meaningful





comparison, we can apply the observational filter of AIRS to the model to make a "model-as-AIRS" dataset that is more spectrally comparable to AIRS measurements. Although the AIRS observational filter is sensitive to waves with larger horizontal and vertical scales than the model is capable of simulating, this approach is still useful. This is because, especially for the case of mountain waves over South Georgia, the accurate simulation of the large scale waves visible to AIRS is critically dependent on the accurate simulation of wave generation mechanisms over the island, which can occur on the very smallest scales.

To produce the model-as-AIRS dataset, we first regrid the model and AIRS data onto a common spatial grid, and then convolve the model data with a height-varying 3D Gaussian filter approximating the true observational filter of AIRS.

For ease of explanation, we refer to these three datasets as

- **AIRS**: the AIRS measurements described in Sect. 2.1,

- the **model**: the high resolution model output described in Sect. 2.2, and

- the **model-as-AIRS**: the model output fields as if they were observed by AIRS, described in this section.

The data processing steps used to produce these three comparable datasets are described below. First, a common horizontal and vertical grid is specified for all three datasets. This grid must be regularly spaced and Cartesian in order for spectral analysis techniques to produce meaningful results. The chosen grid is centred at 54.5°S, 37.1°W (like the model grid) and is $1200 \times 900 \times 75$ km in the zonal, meridional and vertical directions, with a grid spacing of 15 km, 15 km and 1.5 km respectively. This is generally finer than the resolution of the AIRS measurements but much coarser than the model. Since we wish to apply the AIRS horizontal resolution to the model, it is not necessary to have much finer resolution than AIRS.

Each granule or time step of the AIRS measurements and model output is linearly interpolated onto this regular grid. This produces the AIRS and model datasets used throughout this study. All analyses and figures use these regridded data. In order to achieve a consistent horizontal resolution across the AIRS measurements, a horizontal Gaussian smoothing filter with a FWHM equal to 40 km in both the $x$ and $y$ directions is applied to each vertical layer. This also helps to reduce the impact of unwanted pixel-scale noise in AIRS, as discussed in previous studies (Hindley et al., 2016, 2019), and is consistent with our model-as-AIRS processing steps below.

Since the AIRS temperature retrieval has reduced vertical resolution and accuracy outside the height range 20 to 60 km altitude (Hoffmann and Alexander, 2009), we set AIRS data outside this range to zero and apply a half-bell tapering window to the upper and lower boundaries (see Fig. 4, discussed below). This minimises any impact of edge effects in subsequent spectral analysis.

The average vertical resolution of the 3-D AIRS temperature measurements varies between around 7 to 14 km between altitudes of 20 to 60 km (Hoffmann and Alexander, 2009). The full variation of vertical wavelength with altitude can be seen inFig. 5 of Hoffmann and Alexander (2009) and Fig. 2 of Hindley et al. (2019). The horizontal resolution of the AIRS measurements is taken to be 15 km, which is derived from the average horizontal spacing of the measurement grid.

To apply the AIRS resolution to the model output, we consider each vertical layer separately. For each vertical layer, the regridded model temperature perturbations are convolved with a 3-D Gaussian smoothing kernel with a FWHM equal to 40 km





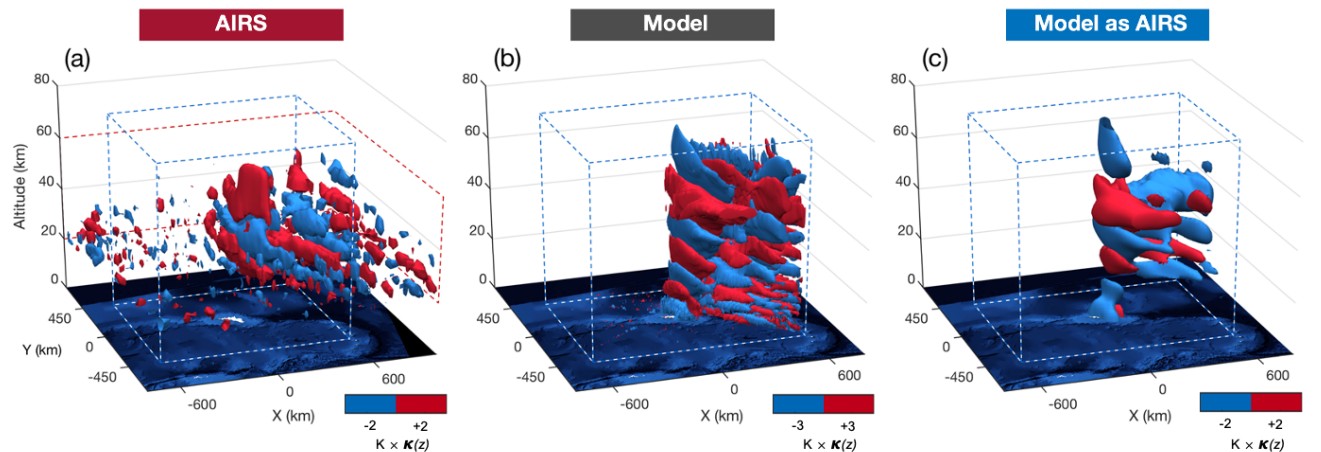

**Figure 5.** Gravity wave temperature perturbations over South Georgia at 1700 UTC on 5th July 2015 for (a) AIRS, (b) model and (c) model-as-AIRS. Here, values are shown as red and blue isosurfaces after a factor of $\kappa(z) = \exp\left(\frac{z - z_{\text{ref}}}{2H}\right)$, where $z_{\text{ref}} = 40$ km is a reference altitude and $H = 7$ km is the scale height of the atmosphere, has been applied in order to see the vertical structure of the wave field clearly. Blue and red dashed lines in each panel indicate the boundaries of the model domain and the AIRS measurements respectively.

in both horizontal directions. This is approximately the Nyquist limit of the AIRS measurements, that is, twice the average

AIRS horizontal spacing or around 20 km. The FWHM in the vertical is taken to be the vertical resolution for that vertical layer (see Fig. 2b of Hindley et al. (2019)). This vertical layer is extracted from the smoothed model to produce one layer of the model-as-AIRS dataset. The unfiltered model is then convolved again in 3-D with the next vertical resolution to produce the next vertical layer of the model-as-AIRS dataset. The rest of the vertical layers are then built up one by one using this method, which is then applied to each hourly model timestep for both the July 2013 and the June-July 2015 modelling campaigns. This

Gaussian-filtering approach is less accurate than using the the full retrieval algorithm of (Hoffmann and Alexander, 2009) or the fine-resampling algorithm of (Wright and Hindley, 2018), however it allows us to produce the observationally-filtered data for this study at much lower computational cost over applying the full 3-D AIRS retrieval scheme on the high-resolution model data.

Because the horizontal and vertical grid spacing of the local-area model is much finer than retrieval grid of the AIRS

measurements, the sensitivity to gravity waves of the AIRS and model-as-AIRS should be dominated by the AIRS observational filter and regridding approaches that we have applied here. Since these approaches are the same for both data sets, we expect the resulting portions of the gravity wave spectrum visible in the AIRS and model-as-AIRS data to be comparable.

## 3.2 Results of applying the AIRS observational filter to the model

The effect of applying the AIRS observational filter to the model is illustrated in Figure 4. For an AIRS overpass at 03:11 UTC

5th July 2015, the regridded temperature perturbations of the AIRS, model and model-as-AIRS data for are plotted as vertical (top row) and horizontal (bottom) cross-sections at $y = 0$ and 45 km altitude in our regular grid. The model timestep shown is





03:00 UTC on the same day, so the time separation is very small and the local wind conditions can be expected to be very close to reality.

The effect of restricting the vertical extent of the AIRS data (as mentioned above) is illustrated in Fig. 4a, where low-quality
measurements outside of altitudes between around 20 and 60 km are removed. The black line to the side of the panel shows the extent of the usable vertical measurement window.

Clear features consistent with gravity wave temperature perturbations are visible in each dataset in Fig. 4. In AIRS measurements and the full resolution model in Figs. 4a and 4b, westward-sloping phase fronts characteristic of a mountain wave field are observed over the island. Longer vertical wavelengths are observed directly over the island itself. The topography of South
Georgia is shown in black at the base of the figure, and is to scale.

As might be expected, the modelled gravity wave field exhibits much more fine horizontal structure than the AIRS measurements due to the higher resolution. These fine horizontal scale structures in the model also exhibit much higher gravity wave amplitudes than those observed by AIRS. The spectral regimes observed by the two datasets also are different.

Once the AIRS observational filter is applied to the model in Fig. 4c, much of the fine horizontal structure in the modelled
gravity wave field is removed and the wave amplitudes are more comparable to those observed in AIRS.

The horizontal cross-sections at an altitude of 45 km in Figs. 4(d-f) also show the same effect. A characteristic "bow wave" pattern centred over the south-eastern tip of the island is observed. This horizontal structure is typical of orographic "mountain waves" generated by small isolated islands (e.g. Alexander and Grimsdell, 2013). Once again, the model exhibits much more fine horizontal structure than is present in the AIRS measurements. Once the AIRS observational filter is applied in Fig. 4f,
this structure is removed and the wave field more closely resembles the AIRS measurements in Fig. 4d.

Interestingly, it can be seen in both cross-sections that wave amplitudes in the model-as-AIRS data are found to be around 20% lower than the corresponding amplitudes observed by AIRS. This could occur if our application of the AIRS observational filter is too harsh, which suppresses these wavelengths more than in the real AIRS measurements, or it could be that the modelled waves at these wavelengths have amplitudes that are too low compared to reality.

An interesting additional feature is seen at in the AIRS measurements in Figs 4(a,d). Between altitudes of around 30 to 60 km at a location of around $x, y = 500, 300$ km relative to the island, a large wave with phase fronts aligned at $+30°$ to the north is observed. By examining this structure in 3-D (not shown) we determined that it is very likely to be a gravity wave. However this wave is not seen in the either the model or model-as-AIRS data in Figs 4(b-c,e-f). The absence of this wave in the model, in addition to its size and orientation, suggests that it is not a mountain wave generated by wind flow over South
Georgia and may instead be a non-orographic gravity wave propagating into the region from outside, or it may be generated in situ by processes not present in the model. More examples of wave structures like this one can be seen in Fig.11, with further discussion in Sect. 4.2.

A clearer impression of the effect of the AIRS observational filter can be gained by visualising the gravity wave structure in 3-D. Figure 5 shows temperature perturbations for AIRS, the model and the model-as-AIRS at 17:00 UTC on 5th July 2015.
The AIRS overpass occurred at 16:41 UTC and the model timestep shown is at 17:00 UTC. This is 14 hours later than the example shown in Fig. 4.





In order to visually inspect the full vertical structure, the temperature perturbations in Fig. 5 are multiplied by a factor of $\kappa(z) = \exp\left(\frac{z - z_{\text{ref}}}{2H}\right)$, where $z_{\text{ref}} = 40\,\text{km}$ is a reference altitude and $H = 7\,\text{km}$ is the scale height of the atmosphere, following the approach of (Sato et al., 2012; Wright et al., 2017). This effectively removes the exponential increase (decrease) in wave amplitude above (below) the reference altitude $z_{\text{ref}} = 40\,\text{km}$. At the reference altitude $z = z_{\text{ref}}$, which is near the centre of the usable vertical range of AIRS measurements, the factor $\kappa(z) = 1$ and the isosurfaces plotted represent the true measured temperature perturbation values. This approach allows us to visually inspect the full vertical extent of the gravity wave structure clearly.

With the scale factor $\kappa(z)$ applied, red and blue isosurfaces are plotted for temperature perturbations at $\pm 2\,\text{K}$, $\pm 3\,\text{K}$ and $\pm 2\,\text{K}$ for the AIRS observations, the model and the model-as-AIRS respectively. Blue and red dashed lines in Fig. 5 show the spatial extent of the model and AIRS measurements respectively. A characteristic mountain wave field is observed in all three datasets. In the AIRS measurements in Fig. 5a, between altitudes of 20 to 40 km, an extended leeward wake structure is observed extending to the east beyond the eastern boundary of the model domain. The wave field directly over the island forms a series of near-vertical phase fronts, while the downstream wake pattern becomes increasingly poorly-defined with increasing altitude. Small-scale measurement noise is observed in a speckled pattern at lower altitudes near 20 to 25 km. The model shows a more clearly defined wake pattern relative to AIRS, which extends downstream from the island at all altitudes without any significant loss of definition. Once the AIRS observational filter is applied in Figure 5c, only the large-scale wave structure remains and the observed and modelled wave fields become visually similar. Once again, wave amplitudes are somewhat lower in the model-as-AIRS data than are observed in the AIRS measurements, but the horizontal structure within the AIRS vertical measurement window of 20 to 40 km altitude is quite similar.

It is interesting to note that the additional wave feature seen north east of the South Georgia in Figure 4(a,d) is now, 14 hours later in Fig. 5, no longer observed. This further suggests that this wave structure may be a transitory wave packet from a source outside the model domain.

### 3.3 Measuring 3-D gravity wave parameters with a 3-D $S$-transform

We identified 87 co-located 3-D AIRS overpasses that coincided with the time periods our local-area modelling simulations over South Georgia during winter July 2013 and June-July 2015. By applying the AIRS observational filter to the model output, we are able to make a fair comparison between the observed and simulated gravity wave fields in 3-D.

In order to investigate the properties of the gravity waves in our AIRS, model and model-as-AIRS datasets, spectral analysis is needed. In particular, we need to be able to measure wave amplitudes, wavelengths and directions but also localise these quantities spatially.

To do this, we use the $S$-transform (also known as the Stockwell transform). Developed by Stockwell et al. (1996), the $S$-transform is a widely-used spectral analysis technique that can localise and measure the amplitudes of individual frequencies (or wavenumbers) in a timeseries or distance profile. The $S$-transform has been applied for gravity-wave analysis in a variety of geophysical datasets (e.g. Fritts et al., 1998; Stockwell and Lowe, 2001; Alexander and Barnet, 2007; Alexander et al., 2008; Stockwell et al., 2011; McDonald, 2012; Wright and Gille, 2013; Alexander, 2015; Sato et al., 2016; Hindley et al., 2016;



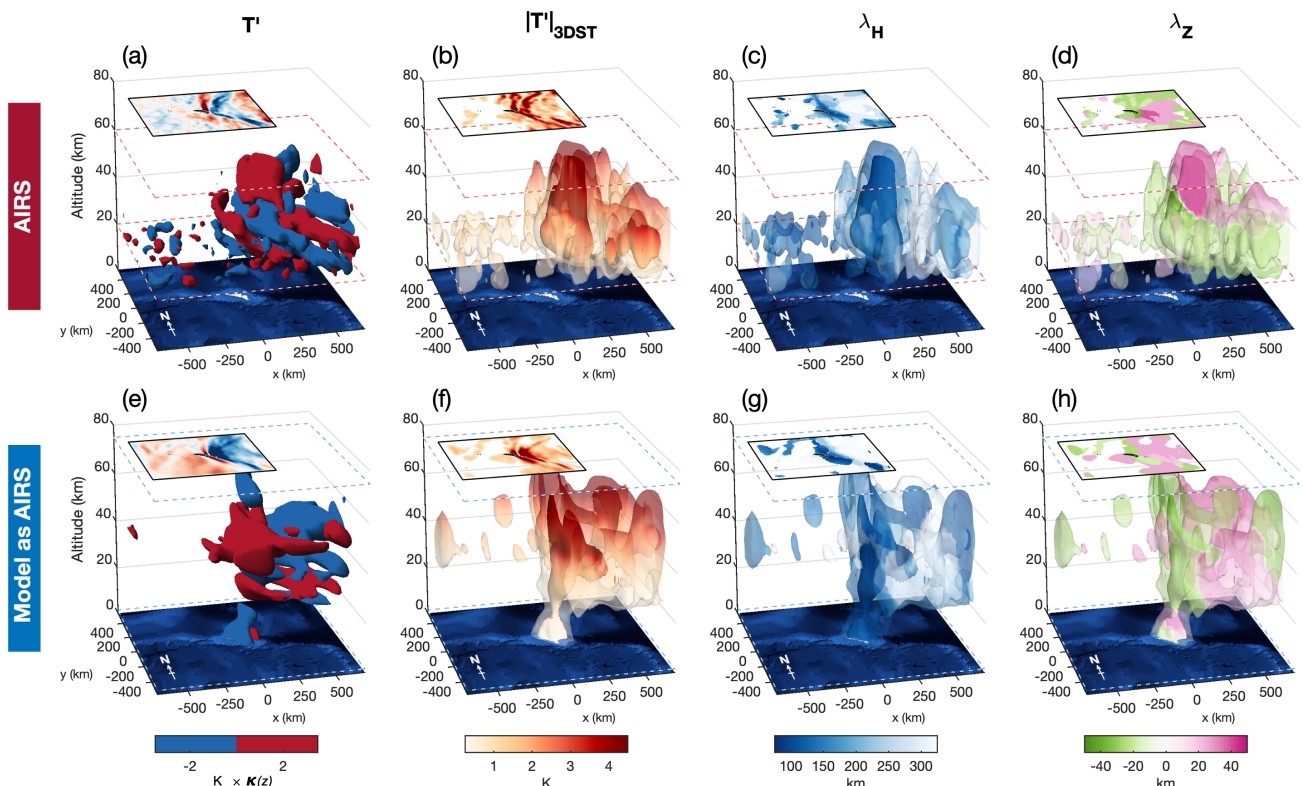

**Figure 6.** 3-D $S$-transform (3DST) analysis of temperature perturbations from AIRS satellite observations (top row) and the model-as-AIRS over South Georgia (bottom row) for 1700 UTC on 5th July 2015. Coloured isosurfaces in panels (a,e) show the AIRS and model-as-AIRS temperature perturbations $T'$, while panels (b,f), (c,g) and (d,h) show 3DST-measured absolute wave amplitude $|T'|_{3DST}$, horizontal wavelength $\lambda_H$ and vertical wavelength $\lambda_Z$ respectively. Blue dashed and red dashed lines denote the upper and lower boundaries of the model domain and AIRS measurements. Horizontal cross-sections through the data at 40 km altitude are shown in the top left hand corners of each panel, sharing a colour scale with the isosurfaces. As in Fig. 5, temperature perturbations in panels (a) and (e) are scaled by the factor $\kappa(z)$ in order to see the vertical wave structure clearly.

Wright et al., 2017; Hindley et al., 2019; Hu et al., 2019a, b) and has also been applied in a variety of other fields, such as the planetary (Wright, 2012), engineering (Kuyuk, 2015) and biomedical sciences (e.g. Goodyear et al., 2004; Brown et al., 2010; Yan et al., 2015).

Here we apply the 3-D $S$-transform (3DST) to our AIRS, model and model-as-AIRS temperature perturbations. We use the 450 $N$-dimensional $S$-transform (NDST) software package developed by Hindley et al. (2019). This version builds on the work of previous multi-dimensional $S$-transform analysis by Hindley et al. (2016) and Wright et al. (2017), but applies a superior wave amplitude measurement technique and features a much faster computational methodology which reduces computation time by around a factor of 10 over previous versions for AIRS analysis. A step-by-step guide describing how the 3DST method



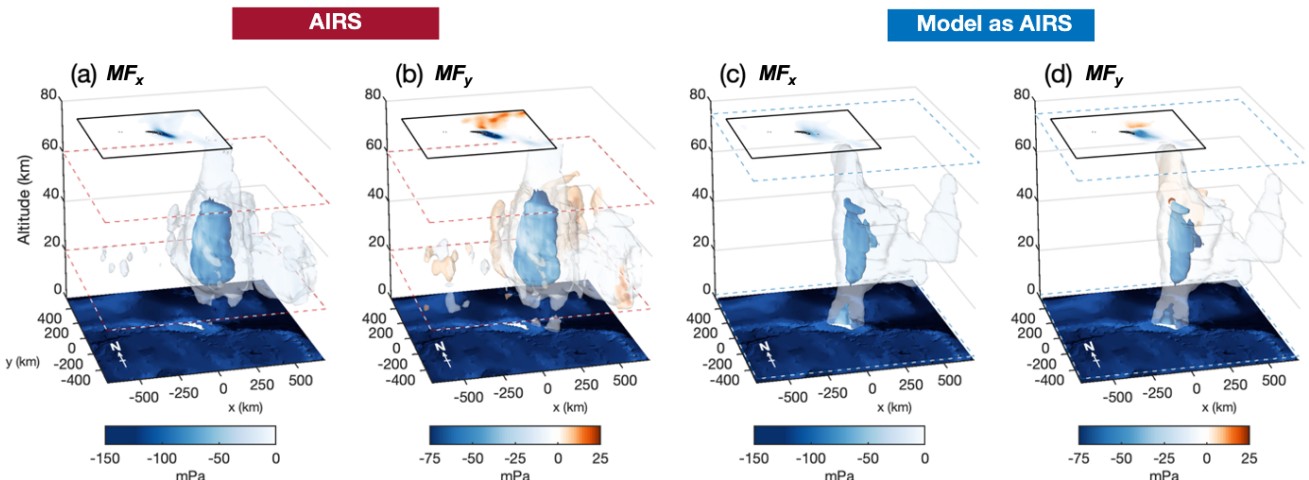

**Figure 7.** As Fig. 6, but for the zonal and meridional components of gravity wave momentum flux $MF_x$ and $MF_y$ for the AIRS and model-as-AIRS data at 1700 UTC on 5th June 2015.

is applied to 3-D Airs measurements is described in Hindley et al. (2019, their Sect. 3). The same process is followed here for
all three datasets.

The 3DST is applied to the regridded temperature perturbations of the 87 3-D AIRS measurements and every hourly time step
for the model and model-as-AIRS output during July 2013 and June-July 2015. The results are spatially localised amplitudes,
wavelengths and directions for all three datasets. Because all three datasets are on the same grid, the same frequencies can be
analysed for all three datasets. Following the approach of Hindley et al. (2019), we set the 3DST scaling parameter $c_x = c_y =$
$c_z = 0.25$ and analyse for the 1000 largest-amplitude wave signals with wavelengths greater than 30 km, 30 km and 3 km in
the $x$, $y$ and $z$ directions respectively.

Figure 6 shows the results of our 3DST analysis for an example of AIRS measurements (top row) and model-as-AIRS
(bottom row) at 1700 UTC on 5th July 2015. These are the examples shown in Fig. 5(a,c). Input and absolute 3DST-measured
wave amplitudes are shown in Figs. 6(a,e) and Figs. 6(b,f) respectively. As in Fig. 5, the factor $\kappa(z)$ has been applied the the
temperature perturbations in panels (a) and (e) in order to show the vertical structure clearly. Panels (b) and (f) show the true
measured values. Horizontal and vertical wavelengths $\lambda_H$ and $\lambda_Z$ are shown in Figs. 6(c,g) and Figs. 6(d,h) respectively. In
each panel, a horizontal cross-section through the data at an altitude of 40 km is overlaid in the top left hand corner, and the
extent of the AIRS and model data are shown by red and blue dashed line respectively.

A clear bow-wave pattern is observed in the AIRS measurements and the model-as-AIRS in Figs. 6(a-b) and 6(e-f). The 3-D
structure of the mountain wave field can be clearly seen in both datasets, and there is reasonable apparent similarity between
the two. The largest wave amplitudes are localised over the island in both datasets and are observed to increase with altitude,
exceeding 5 K at an altitude of 40 km directly over the island in both AIRS and the model-as-AIRS. In general however, the
model-as-AIRS exhibits slightly lower wave amplitudes further away from the island, and the large leeward wave structures




east of the island are have significantly lower amplitudes than their counterparts in the AIRS measurements. This effect, which
is seen throughout this study, suggests that the model has a tendency to generate tightly localised waves in a focused column
directly over the island, while observations show that broader bow-wave shaped wake regions are more enhanced that in the
simulated model-as-AIRS, particularly at lower altitudes.

Measured horizontal wavelength $\lambda_h = \left(\lambda_x^{-2} + \lambda_y^{-2}\right)^{-1/2}$ and vertical wavelength $\lambda_z$ for this example are shown in Figs.
6(c,g) and 6(d,h) for the AIRS and model-as-AIRS respectively. In both datasets, the mountain wave pattern is revealed as a
tight column of short horizontal wavelengths with values 100 km and shorter that is located directly over the island. Longer hor-
izontal wavelengths are observed with increasing horizontal distance from the island. Again, the short horizontal wavelengths
measured in the model-as-AIRS are more tightly localised over the island. In regions with very little wave activity, very long
horizontal wavelengths are measured, although the colour map of Figs. 6(c,g) saturates at 350 km. This is to be expected, as a
relatively featureless horizontal region will be measured in the 3DST analysis as having infinitely long horizontal wavelength.
In the AIRS measurements westward of this island, small isolated regions of short horizontal wavelengths are observed. These
correspond to small-scale uncorrelated noise features in AIRS measurements. Since their measured amplitude is low as seen
in Fig. 6b adjacent, we do not expect these remaining features to significantly affect our results.

### 3.3.1 Regions of downward-propagating wave structure

Figures 6(d,h) show an increase in absolute vertical wavelengths with altitude from about 10 km at altitudes near 20 km to
around 30 to 40 km above altitudes near 45 km. This is consistent with the refraction of a stationary mountain wave field to
longer vertical wavelengths with increasing zonal wind speeds with height. The longest vertical wavelengths are between 30
to 40 km and are observed directly over the island, with vertical wavelength decreasing slightly with increasing horizontal
distance from the island. This is consistent with the phase fronts seen in Fig. 4, 14 hours earlier.

Since the AIRS observations are, on this scale, a pseudo-instantaneous snapshot in time, an ambiguity arises between whether
an observed wave is travelling "upwards and forwards" or "downwards and backwards". This is a common ambiguity in gravity
wave observations. Various approaches have been used to constrain it in recent studies, such as the use of supplementary wind
fields or the assumption of upward propagation (e.g. Alexander et al., 2009; Alexander and Grimsdell, 2013; Wright et al.,
2016a; Hindley et al., 2019). To break the ambiguity for the example in Fig. 6, we assume that these measured waves are
westward propagating mountain waves, whose horizontal phase speeds and wave vector directions are equal and opposite to
the wind speed and direction. This results in wave field that appears stationary with respect to the ground. Simulated zonal
winds in Fig. 2 during this period are almost entirely eastward, so the assumption of a mountain wave field with a westward
orientated wavevector is reasonable.

When this assumption is applied, we are able to constrain the sign, and therefore the direction, of the vertical wavenumber
$m$ from the orientation of the measured wave. In Figs. 6(d) and 6h, positive (negative) vertical wavelength $\lambda_z$ values imply
downward (upward) wave propagation. This sign convention follows the derivation of Fritts and Alexander (2003), where
$m < 0$ for an upwardly propagating wave.





Interestingly, several large pink regions in Figs. 6(d,f) indicate possible regions of downward wave propagation in AIRS and the model-as-AIRS. This seems counter-intuitive given the nature of the probable gravity wave source, and so to investigate this further, we animated a vertical cross-section through the model-as-AIRS wave field as shown in Fig. 4c for each hourly model time step between 0000 UTC on 5th July to 1200 UTC on 6th July 2015. We found that eastward sloping phase fronts leeward of the island persisted strongly from 0200 UTC on the 5th to 2300 UTC on the 6th, dispersing thereafter. Indeed, the feature can be seen in Fig. 4c and was particularly strong at 1700 UTC on the 5th July, when the AIRS overpass occurred. If these wave structures are quasi-stationary with respect to the ground, which would be true for a mountain wave pattern, then this implies westward wave propagation into the prevailing wind. This would in turn imply that these regions do indeed contain downward propagating waves.

In the AIRS measurements in Fig. 6d, large negative vertical wavelengths around -30 to -40 km (implying upward wave propagation) are observed over the island and throughout most of the leeward wake pattern up to altitudes around 40 km. Above this point however, large positive vertical wavelengths around 30 to 40 km are also observed. This abrupt change in the sign of the vertical wavenumber, without any apparent change in the horizontal wavenumber in this area, implies that waves directly over the island have phase fronts aligned near-vertically, such that only a small change in inclination is required to change the sign of the vertical wavenumber. The example in Fig. 4 helps us to interpret this. In the AIRS measurements in Fig. 4a, phase fronts above altitudes of 40 km directly over the island are aligned near vertically. For this reason, we suspect that this change in the sign of the vertical wavenumber is, in this example in AIRS, more likely to be due to measurement error as the phase fronts become near-vertical. However, given the apparent downward propagating wave structure in the model-as-AIRS, we cannot rule out the possibility of a downward propagating structure in the AIRS measurements too, given the striking similarity between the two. Indeed, between around 150 and 300 km east of the island in Fig. 4a, a very small region containing apparently eastward sloping phase fronts is observed.

Hence it is possible that we are indeed observing downwardly propagating waves in the lee of South Georgia. This result would be consistent with the gravity wave analysis of radiosonde observations in Moffat-Griffin et al. (2017), who used data from the same radiosonde campaign as shown in Sect. 2.4. They found that, downwind of the island, 66% of observed gravity wave activity was downward propagating. However, gravity waves in radiosonde measurements correspond to much shorter vertical and horizontal wavelengths than considered here in AIRS measurements.

One possibility is that these downward regions in the model-as-AIRS results are evidence of secondary gravity waves generated by mountain wave breaking or other wave-wave processes. Previous work (e.g. Vadas and Fritts, 2002; Bossert et al., 2017; Becker and Vadas, 2018; Vadas and Becker, 2018; Liu et al., 2019) has suggested that such secondary waves may be generated by large orographic sources, acting as a mechanism to "convert" geographically stationary mountain waves into waves with overall-conserved but non-zero propagation characteristics, including the possibility of generating downward-propagating waves. While such work to date has focused on major gravity wave hot spots such as the southern Andes, South Georgia may also be a sufficiently intense mountain wave source to exhibit such effects. It could also be the case that such downward-propagating regions of the field are generally a normal feature of the gravity wave field over small-island sources, even in the absence of secondary wave generation, but their effects are usually dominated in high-resolution models by the





much more intense fine model structure (see Fig. 4b). Given that a model damping layer above altitudes of 58.5 km was applied in the simulations, reflections from critical layers near the model top are unlikely. More investigation of wave features such as these in future studies is needed.

### 3.3.2 Zonal and meridional momentum fluxes

A key quantity in gravity wave research is the vertical flux of horizontal pseudo-momentum, generally referred to as momentum flux. This property helps to quantify the transfer of momentum carried by gravity waves, and the constrains the drag or driving effect on the mean flow that will arise when the wave eventually breaks or is absorbed. Quantifying the momentum flux budget of mountain wave sources from isolated small islands is a key current area of research (McLandress et al., 2012; Alexander and Grimsdell, 2013; Garfinkel and Oman, 2018; Jackson et al., 2018).

The zonal and meridional components of gravity-wave momentum flux $MF_x$ and $MF_y$ can be estimated from our 3DST measurements of wave amplitude, horizontal wavelength and vertical wavelength via the relation in Ern et al. (2004):

$$(F_x, F_y) = \frac{\rho}{2} \left(\frac{g}{N}\right)^2 \left(\frac{|T'|}{\bar{T}}\right)^2 \left(\frac{\lambda_z}{\lambda_x}, \frac{\lambda_z}{\lambda_y}\right) \tag{1}$$

where $\rho$ is atmospheric density, $g$ is the acceleration due to gravity, $N$ is the buoyancy frequency, $|T'|$ is absolute wave amplitude, $\bar{T}$ is the background temperature, and $\lambda_x$, $\lambda_y$ and $\lambda_z$ are zonal, meridional and vertical wavelengths respectively. Wright et al. (2016a) and Ern et al. (2017) showed that this relation, based upon the mid-frequency approximation, is valid for the waves to which the 3-D AIRS retrieval is sensitive.

Fig. 7 shows zonal and meridional momentum fluxes $F_x$ and $F_y$ calculated via Eqn. 1 for the AIRS and model-as-AIRS measured gravity wave quantities shown in Fig. 6. For each panel, isosurfaces are drawn where the absolute momentum flux $|F_{x,y}| = \sqrt{F_x^2 + F_y^2} = 10$ and 100 mPa. These isosurfaces are then coloured using the values of zonal and meridional flux at each location. As in Fig. 6 horizontal cross section at an altitude of 40 km is overlaid in the top left hand corner of each panel.

Highest momentum fluxes values in Fig. 7 are observed in a vertical column directly over the island in both the AIRS and model-as-AIRS wave fields. These regions coincide with the largest wave amplitudes, shortest horizontal wavelengths and longest vertical wavelengths in Fig. 6, so this is to be expected. Zonal momentum fluxes are almost entirely directed westward (negative), with values that are significantly higher than the meridional components, exceeding -150 mPa in both the AIRS and model-as-AIRS data directly over the island. In the meridional direction, regions of northward (positive) and southward (negative) flux are observed to the north and south of the island respectively in both datasets in a characteristic bow-wave pattern. This is an encouraging result that suggests that our 3DST technique is correctly localising the opposing meridional components of the bow-wave pattern. Southward fluxes are around 3 times larger than their northward counterparts, exceeding 75 mPa in both the AIRS and model-as-AIRS data.

The AIRS measurements for this example in Fig. 7 exhibit higher momentum fluxes than the coincident model-as-AIRS time step at 1700 UTC on 5th July 2015. Inspection of Fig. 6 suggests this is primarily due to the generally increased wave amplitudes measured by AIRS as discussed above. Once again, it is apparent that the AIRS measurements exhibit a much





**Figure 8.** Time series of median gravity-wave amplitudes (coloured lines) from AIRS observations (a,b), the full-resolution model (c,d) and the model-as-AIRS (e,f) over a horizontal region of 250 km radius centred 100 km east of the island and a vertical region between altitudes of 25 and 45 km for July 2013 (left) and June-July 2015 (right). Shaded regions in panels (a-f) show the 25th and 75th, 15th and 85th, and 5th and 95th percentiles of measured wave amplitudes over the same region. Red circles in (a,b) show the overpass times of the AIRS measurements. Panels (g) and (h) show the magnitude of the wind speed from the model for the 2013 and 2015 modelling campaigns respectively.

more enhanced leeward bow-wave pattern than is observed in the model-as-AIRS, whose flux is more concentrated into a tight

column directly over the island.



**Figure 9.** Time series of zonal gravity-wave momentum flux $MF_x$ for the AIRS observations (a,b), the full-resolution model (c,d) and the model-as-AIRS (e,f) over South Georgia during June-July 2013 and 2015 for the same region used in Fig. 8. Here, coloured lines show the average net zonal momentum flux, while the grey shaded areas show the average eastward and westward fluxes over the region. Positive (negative) values indicate an eastward (westward) direction. Panels (g) and (h) show horizontally-averaged zonal wind speed from the model for the 2013 and 2015 modelling campaigns respectively.

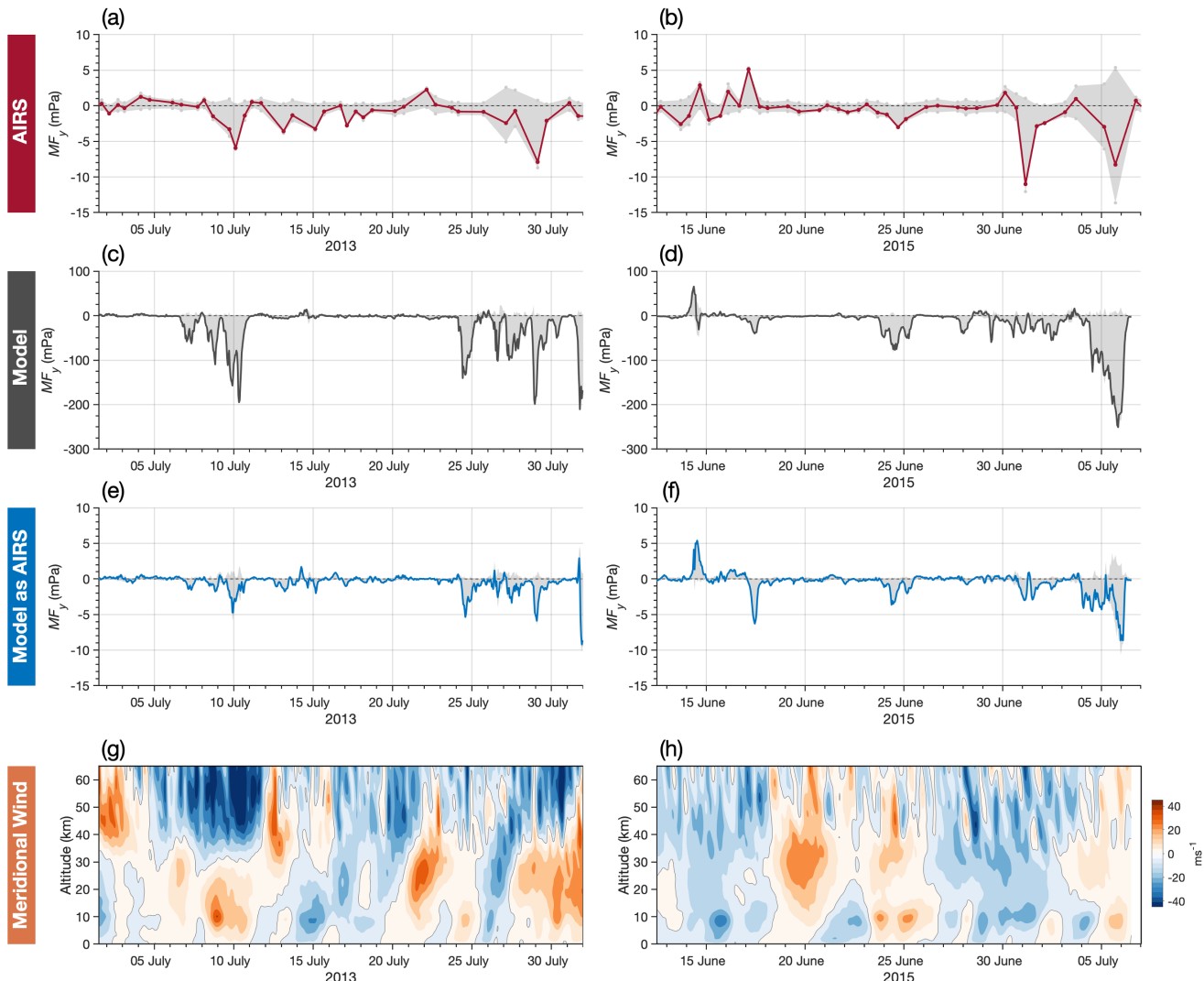

**Figure 10.** As Fig. 9, but for the meridional component of gravity-wave momentum flux $MF_y$ in panels (a) to (f) and meridional wind in panels (g) an (h). Coloured lines show the average net meridional flux, while the grey shaded areas show the average northward and southward fluxes. Here, positive (negative) values indicate an northward (southward) direction.

## 4 Results

### 4.1 Time series of wave amplitudes and directional momentum fluxes in the AIRS, model and model-as-AIRS

The examples shown in Sect. 3.3 demonstrate the effect of applying the AIRS observational filter to the model output and that wave amplitudes, wavelengths and directional momentum fluxes can be measured in each dataset using the 3DST method.





In this section we investigate how these properties vary temporally during the two modelling campaigns in July 2013 and June-July 2015.

Figures 8, 9 and 10 show time series of wave amplitude, zonal momentum flux $MF_x$ and meridional momentum flux $MF_y$ respectively. For each AIRS overpass and time step of the model and model-as-AIRS, the median wave amplitude and mean directional momentum fluxes are found within a horizontal region of radius $r = 250$ km centred 100 km east of the island 585 between altitudes of 25 and 45 km. This horizontal region was chosen so as to capture the most intense region of the gravity wave fields just to the east of the island (this is best illustrated in Figs. 12 and 14, which are discussed later). The height region was chosen in order to take advantage of the best AIRS vertical resolution at lower altitudes and to avoid any effects from the model damping layer at high altitudes. Red markers in panels (a) and (b) of each figure indicate the overpass times of the AIRS instrument. Model wind speeds, averaged horizontally over the model domain, are shown in panels (g) and (h) of each figure.

In Fig. 8, coloured lines show the median wave amplitude in the region. The area within the shaded regions show the 5th and 95th, 15th and 85th and 25th and 75th percentiles of all measurements within the region. The 5th and 95th percentile shading is the lightest, while the 25th and 75th percentile shading is the darkest. This approach is useful for two reasons. First, gravity wave amplitudes are likely to exhibit a log-normal distribution, so the median is more appropriate than the mean. Second, this approach can give us information about the spatial distribution of wave amplitudes within the region.

Figure 8 reveals that gravity wave activity over the island is highly intermittent. Several time periods of enhanced gravity activity are seen in all three datasets, such as the periods 7th - 11th July and 24th - 31th July during 2013, and the periods 14th - 16th June, 24th - 26th June and 29th June - 6th July during 2015. These periods of increased stratospheric wave activity generally coincide with high wind speeds that extend down into the troposphere. This gives rise to a vertical conduit of high wind speeds that allows mountain waves from South Georgia to propagate vertically without encountering critical levels.

During these periods, median gravity wave amplitudes between 1 and 2 K are observed in AIRS measurements, with higher values of around 2 to 6 K observed in the model. Somewhat lower median values typically less than around 1 K are observed in the model-as-AIRS. The shaded percentile regions in Fig. 8 yield information within this distribution and reveal high wave amplitudes exceeding 16 K at the 95th percentile in the model on 17th June and 5th July 2015. The latter example is the one shown in Fig. 6. The fact that the median values are quite low for the model suggests that these high amplitudes are confined 605 to a small region immediately downwind of the island, as we will show later in Fig. 12.

Wave amplitudes do not reach such high values in the AIRS and model-as-AIRS, whose 95th percentile values typically range between 5 and 6 K. Interestingly, the AIRS measurements exhibit a somewhat narrower distribution of wave amplitudes during periods of high wave activity, since the percentile boundaries are quite close together. This suggests that wave amplitudes are more broadly distributed over the horizontal region rather than tightly localised, which could be indicative of large-scale 610 non-orographic wave activity or simply longer horizontal wavelengths with large amplitudes in AIRS measurements. The impact of measurement noise in AIRS is also apparent in the measured wave amplitudes, with median values almost never falling below 0.5 K, even during periods of low wave activity. This suggests a usable noise threshold of around 0.5 K for AIRS amplitudes. It is very difficult to determine whether this noise is due to instrument measurement noise, 3-D retrieval noise, artefacts arising from unresolved waves or true anisotropy in the real atmosphere, but it is likely that all of these sources of





error play a part. In the model and model-as-AIRS, median amplitudes fall to near zero due to the lack of this measurement
       noise.

       The median model-as-AIRS amplitudes over this region rarely exceed those in the AIRS measurements, however there are
       some notable exceptions. One such example is on 17th June 2015, where a brief spike of increased wave amplitude is observed
       with median wave amplitudes nearly reaching 3 K in the model-as-AIRS and the 95th percentile exceeding 5 K. Inspection of
the model wave field during this period revealed a series of large and likely non-orographic wave fronts that formed over most
       of the model domain. These phase fronts did not appear to move with time, but also did not appear to be orographic (that is, no
       clear bow-wave structure), suggesting that they may have originated in the global forecast that supplied the model boundary
       conditions.

       Time series of the zonal and meridional components of gravity-wave momentum flux in Figs. 9 and 10 show a similar
pattern. Here, coloured lines indicate the average (net) zonal and meridional flux over the same horizontal region as used in
       Fig. 8 between altitudes of 25 and 45 km. Shaded regions indicate the average positive and negative flux values within the
       region, with positive (negative) values indicating eastward and northward (westward and southward) directions in Figs. 9 and
       10 respectively. These shaded values are found by taking the average of all the positive (negative) fluxes with all the negative
       (positive) fluxes set to zero. This information is important since, for a typical mountain wave field such as might be observed
over South Georgia, significant northward and southward flux could be generated but the net meridional flux could be very low
       since the two directions may cancel out, suggesting incorrectly that there is little gravity wave activity. Zonal and meridional
       wind speeds against altitude from the model, averaged horizontally over the model domain, are shown in panels (g,h) in both
       figures.

       Net zonal fluxes in Fig. 9 are largely westward in all three data sets. As discussed above, an assumption of upward wave
propagation is used to break the directional ambiguity in the data. The fact that the zonal flux is overwhelmingly westward,
       which is what we would expect given the eastward wind conditions, after this assumption has been made is a good indication
       that the assumption is reasonable for the majority of wave measurements here.

       A similar pattern of gravity wave activity with time is observed compared to Fig. 8, with wave activity occurring in bursts that
       last several days at a time. In terms of the timing of these bursts of momentum flux, the model and model-as-AIRS reproduce
the AIRS observations remarkably well. Average values in the model over the region peak around 200 to 300 mPa during these
       wave bursts, falling to near zero during periods of low wave activity. Zonal flux values are more comparable in AIRS and the
       model-as-AIRS, peaking at around 10 to 20 mPa and 5 to 15 mPa respectively. As with the wave amplitudes, average zonal
       fluxes in the model-as-AIRS can be around 25% lower than the AIRS fluxes during the same periods. This suggests more wave
       activity in the model may be occurring outside of the AIRS observational window, most likely at very short horizontal scales
below around 50 km.

       Although the direction of the zonal fluxes is overwhelmingly westward, particularly in the model, small periods of eastward
       flux are observed, for example during 9th-10th July 2013 in the model-as-AIRS and 4th-7th July 2015 in both AIRS and the
       model-as-AIRS. Although much of this eastward flux could be real, if we recall that in Fig. 6(d,h) we identified regions where
       the gravity wave phase fronts were aligned so close to the vertical that they were measured as being eastward sloping with





height, which could suggest downward wave propagation. If we always assume upward wave propagation, as we have done here, then the direction of flux from downward propagating waves would be reversed. Here, such downward flux would be incorrectly measured as eastward, so we may be underestimating the westward zonal flux in Fig. 9 during these time periods.

Meridional fluxes in Fig. 10 are largely southward, particularly in the model and model-as-AIRS data. As before, the variability of meridional flux shows a similar pattern with time in all three data sets, where intermittent bursts of gravity wave activity appear to last a few days at a time. Interestingly, it can be seen in panels (g) and (h) that these periods of enhanced flux coincide with periods of northward winds below altitudes of around 20 km down to the surface. This makes sense, since the topography of South Georgia presents a larger cross-section to the prevailing surface wind when the wind vector is directed northward, resulting in a larger southward orographic wave forcing. This feature has been observed in previous studies (e.g. Alexander et al., 2009; Alexander and Grimsdell, 2013; Hindley et al., 2016), where peak fluxes are often found over the south eastern tip of the island. Larger wave amplitudes are generally observed in the southern section of the mountain wave pattern, usually corresponding to a net southward momentum flux. This highlights the importance of resolution in models, since this behaviour would not be replicated if South Georgia was modelled simply as a single grid cell point source in a low-resolution simulation.

The AIRS measurements exhibit a slightly lower tendency towards southward fluxes, with significant northward flux observed in Figs. 10(a,b). A small fraction of this distribution is likely to be measurement error, but the results may still be significant. The increased northward flux in the AIRS observations is likely to be due to either the presence of more non-orographic waves in the observations, which could be expected to have a more random distribution of wavevector directions, or it may be indicative of a southward bias in the modelled fluxes. This bias could be caused by a northward wind bias in the model winds. Our comparison of model wind speeds to radiosondes observations in Sect. 2.4 revealed a small northward bias in the model between altitudes of 5 and 10 km, but a more significant southward bias was observed above altitudes around 15 km, so it is not clear if this could be a significant factor.

Only one significant period of northward flux is observed in all three data sets at the same time, during 14th June 2015. This coincided with a period of enhanced northward flux in the AIRS observations and a period of southward winds of 10 to 30 ms$^{-1}$ at altitudes from the surface to 60 km in the model.

## 4.2 Examples of simulated and observed gravity wave structures in 3-D

To make a more in-depth comparison of individual cases, we inspect the wave temperature perturbations in 3-D. Fig. 11 shows AIRS and model-as-AIRS gravity wave temperature perturbations for ten selected AIRS overpasses and their corresponding model time steps. Isosurfaces are drawn at $\pm 2$ K for AIRS data and $\pm 1$ K for the model-as-AIRS. As in Fig. 5, the factor $\kappa(z)$ has been applied to the temperature perturbations in order to see the vertical wave structure clearly. The selected events are arranged in panels in chronological order.

For these select examples, the observed and simulated wave fields are quite similar. Mountain wave structures are clearly visible in both the AIRS and model-as-AIRS data on the 9th, 10th, 27th and 29th July 2013 and 14th June 2015 in Fig. 11, not including the examples on 5th July 2015 shown in Figs. 4 and 5. The directionality of the mountain wave field can also





**Figure 11.** Selected examples of temperature perturbations from AIRS measurements and the model-as-AIRS during the July 2013 (top) and June-July 2015 (bottom) modelling campaigns. Temperature perturbations are shown as coloured isosurfaces at $\pm 2$ K for AIRS data and $\pm 1$ K for model-as-AIRS, with red indicating positive values and blue negative. As in Fig. 5, the factor $\kappa(z)$ has been applied to the temperature perturbations in all examples in order to see the vertical wave structure clearly. Red and blue dashed lines illustrate the spatial extent of the AIRS measurements and model domain.

be inspected. In Figs. 11(a,b,e) the mountain wave field in AIRS exhibits a slightly southward orientation on 9th, 10th and

29th July 2013, while in Figs. 11(d,f) more northward orientations are observed on 27th July 2013 and 14th June 2015. These





examples correspond to periods of net southward and northward momentum fluxes respectively in the time series in Fig. 8(a,b). This provides confidence in the direction-finding of the 3DST analysis. In these examples in early July 2013 however, very little meridional flux is observed in the model-as-AIRS. Since these are clear mountain wave structures, it suggests that this could be due to errors in the speed and direction of the background wind in the model.

One other interesting event occurred at 0300 UTC on 1st July 2015, shown in Fig. 8. Here, AIRS shows the largest measured wave amplitudes for the whole 2013 and 2015 periods, with 95th percentile values exceeding 8 K. This event corresponded to large westward and southward momentum fluxes of around 10 and 12 mPa respectively in Figs. 9 and 10. This event on the 1st July 2015 is shown in Fig. 11j. Here, a large wave structure is observed in AIRS measurements occupying almost the entirely domain. Dominant wave vectors for this wave structure are orientated in a southward direction. In the model-as-AIRS,

a similar structure is observed but wave amplitudes are significantly lower and the wave is orientated further southwards. Significant wave amplitudes are observed both upwind and downwind of the island, particularly in AIRS, where phase fronts aligned in the opposite direction to what we would expect for a mountain wave pattern. Our interpretation is that this is a clear example of a large-scale non-orographic wave in the region. Other examples of potential non-orographic waves are shown in panels (c), (g), (h) and (i).

The origin of these large-scale non-orographic waves is not clear, but it is clear that their amplitudes appear to be significantly under-estimated in the model-as-AIRS in these examples. These waves may originate from in situ non-orographic processes such as jet adjustment around the edge of the polar vortex, or from intense storms and fronts in the Drake Passage region. Hindley et al. (2019) reported that large-scale gravity waves with very similar characteristics to the example shown in Fig. 11j were commonly observed in AIRS measurements over the Southern Ocean during winter. The geographic location of South

Georgia may also be significant. The island lies only 2000 km east of the southern tip of South America, a region associated with the largest stratospheric mountain wave activity observed anywhere in the world (e.g. Hoffmann et al., 2013, 2016). One other possibility therefore is that these waves are in fact mountain wave structures that formed over the southern Andes and Antarctic Peninsula but have since become detached due to changing wind conditions, as discussed by Sato et al. (2012); Garfinkel and Oman (2018); Hindley et al. (2019) and others. These waves could also be secondary waves generated as a

result of intense primary mountain wave breaking over the southern Andes near the stratopause region (Woods and Smith, 2010; Bossert et al., 2017; Vadas et al., 2018; Becker and Vadas, 2018). These secondary waves would have non-zero phase speeds so could be observed far downwind of their source regions. Further investigation is needed to quantify the relative contributions of these different wave sources observed in the AIRS measurements so that we can determine the reasons for the under-representation of these waves in the model.

### 4.3   Horizontal distributions and intermittency of gravity waves around the island

In Figs. 8-10 we found gravity wave activity over the model domain during June-July 2013 and 2015 often occurred in intermittent bursts. As discussed above, orographic waves from South Georgia may only make up one part of this wave activity, so in order to assess this here we investigate of the horizontal distribution of wave activity around the island in AIRS, the model and the model-as-AIRS.





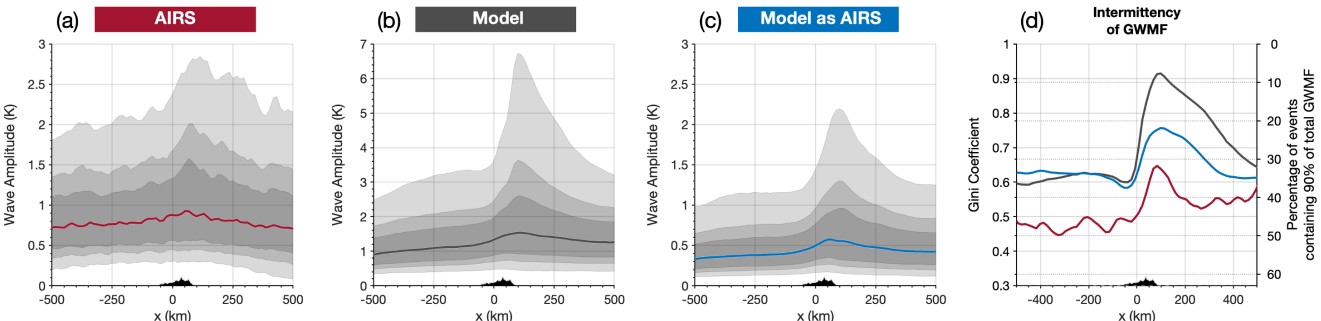

**Figure 12.** Measured gravity wave amplitudes as a function of distance east (positive $x$) and west (negative $x$) from the model centre for (a) AIRS, (b) the model and (c) the model-as-AIRS during June-July 2013 and 2015. Median wave amplitudes are shown by red, black and blue coloured lines in (a-c) for a meridional region $\pm250$ km and a vertical region between altitudes of 25 and 45 km, while the shaded regions show the 5th and 95th, 15th and 85th, and 25th and 75th percentiles. Panel (d) shows the Gini coefficient of the average absolute gravity wave momentum flux (GWMF) over the same region for all AIRS overpasses and model time steps, which is related to the intermittency of GWMF over time. Red, black and blue lines in (d) correspond to AIRS, the model and the model-as-AIRS respectively. The right hand axis of panel (d) also shows the corresponding percentage of wave events that carried 90% of the total GWMF during the 2013 and 2015 campaigns. The topography of South Georgia is shown in black at the bottom of each panel for illustration.

Figure 12 shows the distributions of measured wave amplitudes as a function of horizontal distance east and west from the model centre. A region is selected that is $y = \pm250$ km in the meridional direction and between 25 to 45 km in the vertical. Median wave amplitudes are found in this region for all AIRS measurements and model time steps during June-July 2013 and 2015. As in Fig. 8, coloured lines in Figs. 12(a-c) show the median wave amplitude as a function of horizontal distance east and west of the island. The shaded areas show the extent of the 5th and 95th, 15th and 85th and 25th and 75th percentiles of

the wave amplitude distribution. The island topography is shown in the bottom centre of each panel to guide the eye.

In all three datasets, largest amplitudes are measured just to the east of the island, around 100 km away from the model centre. Median values of around 1 K are observed at this location in AIRS, but in the model-as-AIRS median values peak at around to 0.6 K. In the model results in Fig. 12b, largest median amplitudes reach 1.6 K just to the east of the island, but the 95th percentile reaches nearly 7 K in the model compared to nearly 3 K for AIRS and just over 2 K for the model-as-AIRS.

This indicates that tightly localised wave amplitudes over the island in the model were significantly higher than in AIRS measurements, as we would expect. However, the model-as-AIRS values did not reach similar values.

The distribution of wave amplitudes with zonal distance from the island in Fig. 12 is somewhat different too. In the model and model-as-AIRS, low amplitude values are found upwind (to the west) and a large spike is seen immediately downwind of the island, with values decreasing sharply with eastward distance. In the AIRS results in panel (a), the largest amplitudes are

also found over and immediately downwind of the island. However, a larger relative distribution of increased wave amplitudes is seen at large horizontal distances from the island than is observed in the model-as-AIRS. This is indicated by the amplitude





percentile regions being spaced somewhat further apart than in the model-as-AIRS. Although wave amplitudes do decrease somewhat with increasing distance from the island (including slightly larger values to the east than the west, as is seen in the model) this tendency towards larger wave amplitudes away from the island is strongly suggestive of relatively large-amplitude

non-orographic wave activity measured by AIRS. Although there is significant non-orographic wave activity in the model, the results of Fig. 12 suggest that their amplitudes are much lower in the model-as-AIRS than is observed in reality. Although measurement noise in AIRS is likely to play a part in these results, the increased 85th and 95th percentile values in Fig. 12 are suggestive of large amplitude wave events that would be unlikely to be produced solely by retrieval noise.

Figure 12d shows the intermittency of absolute gravity wave momentum flux throughout June-July 2013 and 2015 as a

function of zonal distance from the island, using the same region as used in panels (a) to (c). To quantify intermittency, we use the Gini coefficient (Gini, 1912). The Gini coefficient quantifies the unevenness of a distribution using a scalar value between zero and one, and has been used in numerous gravity wave studies in recent years (e.g. Hertzog et al., 2012; Plougonven et al., 2013; Wright et al., 2013, 2017; Hindley et al., 2019). Here, a high value for the Gini coefficient implies that momentum fluxes are unevenly distributed into a few large events during the 2013 and 2015 campaigns (high intermittency). A low value for the

Gini coefficient implies that fluxes are more evenly distributed into more frequently occurring events of comparable intensity (low intermittency). Generally, orographic wave sources have been found to exhibit higher intermittency than non-orographic sources over long timescales in previous studies (Hertzog et al., 2012; Plougonven et al., 2013; Wright et al., 2013; Hindley et al., 2019).

All three data sets exhibit the highest intermittency immediately to the east of the island, with Gini coefficient values near

0.65, 0.92 and 0.75 for the AIRS, model and model-as-AIRS respectively. This is consistent with the results of previous studies that found orographic gravity wave sources to exhibit higher intermittency than non-orographic sources. Gini coefficient values decrease with increasing distance from the island, but values are higher east than to the west, as a result of the leeward mountain wave field from South Georgia which is more intermittent than non-orographic waves measured upwind.

Gini coefficient values in the AIRS measurements are significantly lower than the model and model-as-AIRS at all horizontal

distances from the island, implying lower intermittency. One reason for this is likely to be measurement noise, which would reduce the Gini coefficient value. This is because, during periods of very little wave activity, measurement noise in AIRS would still exhibit some small amplitudes rather than falling to near zero as seen in the model and model-as-AIRS in Fig. 8. However, given the increased wave amplitudes away from the island in Fig. 12a, it could also be indicative of non-orographic activity in the AIRS measurements that is under-represented in the model.

It is useful to relate these intermittency values to a more meaningful quantity. Following the approach of Hindley et al. (2019), we can use the Gini coefficient to find the percentage of gravity wave events that contributed to 90% of the total momentum flux during both time periods. This is shown on the right hand axis of Fig. 12d. For a perfectly even distribution, 90% of the total flux would be carried by 90% of the wave events. For an uneven distribution, such as we observe here, this percentage is significantly lower.

We find that in the model, 90% of the total momentum flux over the island was carried by less than 10% of wave events, compared to around 22% and 32% in the model-as-AIRS and AIRS measurements respectively. This shows that, despite the





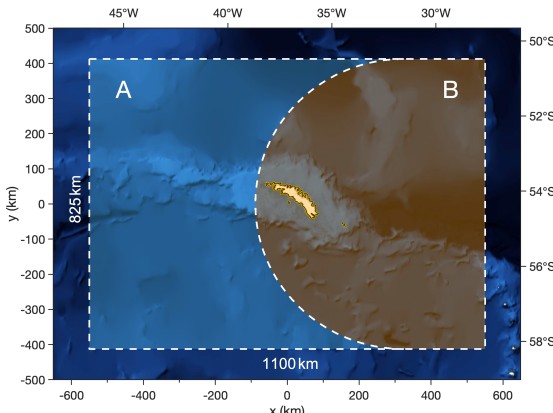

**Figure 13.** Illustration of the two regions to the east and west of South Georgia used to produce the values in Table 1. Region A is upwind of the island and Region B is over and downwind of the island. The two regions have equal area.

| | | Wave Amplitude | Zonal MF (mPa) | | Meridional MF (mPa) | | % of total |
|---|---|---|---|---|---|---|---|
| | | (K) | Eastward | Westward | Northward | Southward | absolute MF |
| AIRS | A | 1.02 | 0.55 | -2.28 | 0.85 | -1.49 | 43% |
| | B | 1.11 | 0.60 | -3.57 | 1.06 | -1.94 | 57% |
| Model | A | 1.56 | 0.01 | -3.14 | 0.35 | -2.69 | 15% |
| | B | 2.16 | 0.49 | -16.83 | 1.75 | -12.90 | 85% |
| Model-as-AIRS | A | 0.61 | 0.01 | -1.18 | 0.14 | -0.68 | 28% |
| | B | 0.81 | 0.17 | -2.24 | 0.47 | -1.07 | 72% |

**Table 1.** Average gravity wave amplitudes and directional momentum fluxes over South Georgia during June-July 2013 and 2015 for the AIRS observations, the model and the model-as-AIRS. Values are shown for two geographical regions A and B as shown in Fig. 13 for altitudes between 25 to 55 km. The two regions have equal area, and the rightmost column shows the percentage of the total absolute momentum flux in each data set that was contained in each region A and B.

effects of retrieval noise, the same fraction of the total momentum flux over the island in AIRS measurements was carried by significantly more wave events than in the model or model-as-AIRS. Because the distribution of wave amplitudes with horizontal distance in Figs. 8a implies increased wave activity away from the island, and particularly upwind, this result could

further support the hypothesis that the model is under-representing the contribution of large-scale non-orographic wave events to the total momentum flux over the time period of the campaign.





### 4.4 Gravity wave properties upwind and downwind of South Georgia

To further explore the question of orographic versus non-orographic wave activity in the region, we divide the model domain into two regions A and B, as shown in Fig. 13. These two regions have equal area of $453750 \, \mathrm{km}^2$, and together they form a
rectangular region that is $1100 \, \mathrm{km} \times 825 \, \mathrm{km}$ as shown. Region A is designed to capture non-orographic wave activity upwind (to the west) of the island, while region B is designed to capture both orographic and non-orographic wave activity over and downwind (to the east) of the island. The shape of region B consists of a rectangle and half circle of diameter $825 \, \mathrm{km}$. This shape was chosen to be a simple representation of the mountain wave field region (see Fig. 4(d-f)) with a horizontal area that is straightforward to calculate. In the vertical, these regions extend between altitudes of 25 and 45 km, which is the same height
range as used in Figs. 8 to 12.

Table 1 shows average wave amplitudes, directional momentum fluxes and the percentage of the total momentum flux in each region A and B during June-July 2013 and 2015. As in Figs. 9 and 10, campaign-average directional flux values are found by taking the average of all the positive flux values with the negative values set to zero and vice versa to give the true area average. The larger area of these regions mean that this approach yields values that are somewhat lower than those shown
in previous figures. Because these two regions have equal area, the average directional flux over both regions is simply the average of the two values shown for regions A and B in Table 1. The net flux over both regions is then the sum of the average eastward and westward (or northward and southward) values.

Average wave amplitudes and fluxes are significantly larger downwind of the island in region B than upwind in region A in all three data sets, but values also differ significantly between the data sets. For example, in AIRS measurements the net zonal
momentum flux in region B is -2.97 mPa, but in the model-as-AIRS it is -2.07 mPa, some 31% lower.

Further contrast is found when we compare the fraction of the total momentum flux in each dataset contained within region A and region B. Interestingly, 85% of the total absolute momentum flux in the model and 72% in the model-as-AIRS was found in region B, but in the AIRS measurements only 57% of the total flux was found in region B compared to 43% in region A. If retrieval noise in AIRS is assumed to be uniformly distributed both upwind and downwind of the island, then such
noise will not affect this value. This suggests that gravity wave momentum flux is significantly more evenly distributed over the whole region in AIRS measurements than was simulated in the model. When the AIRS observational filter is applied, the model-as-AIRS percentage becomes closer to the AIRS value for region B, but is still 15% higher.

This result reinforces some of the earlier findings of this study. Namely, the model seems to underestimate the momentum flux associated with non-orographic waves, or perhaps overestimates the flux associated with orographic waves from South
Georgia, when compared to AIRS measurements. Direct inspection of the gravity wave fields in the model do reveal significant non-orographic wave features but, as shown above, wave amplitudes tend to be around 20 to 25% lower in the model-as-AIRS compared to observations. Since this is also true for mountain waves in the model-as-AIRS, we suspect therefore that it is indeed an underestimation of non-orographic wave activity in the model rather than an overestimation of orographic wave activity that leads to this discrepancy.





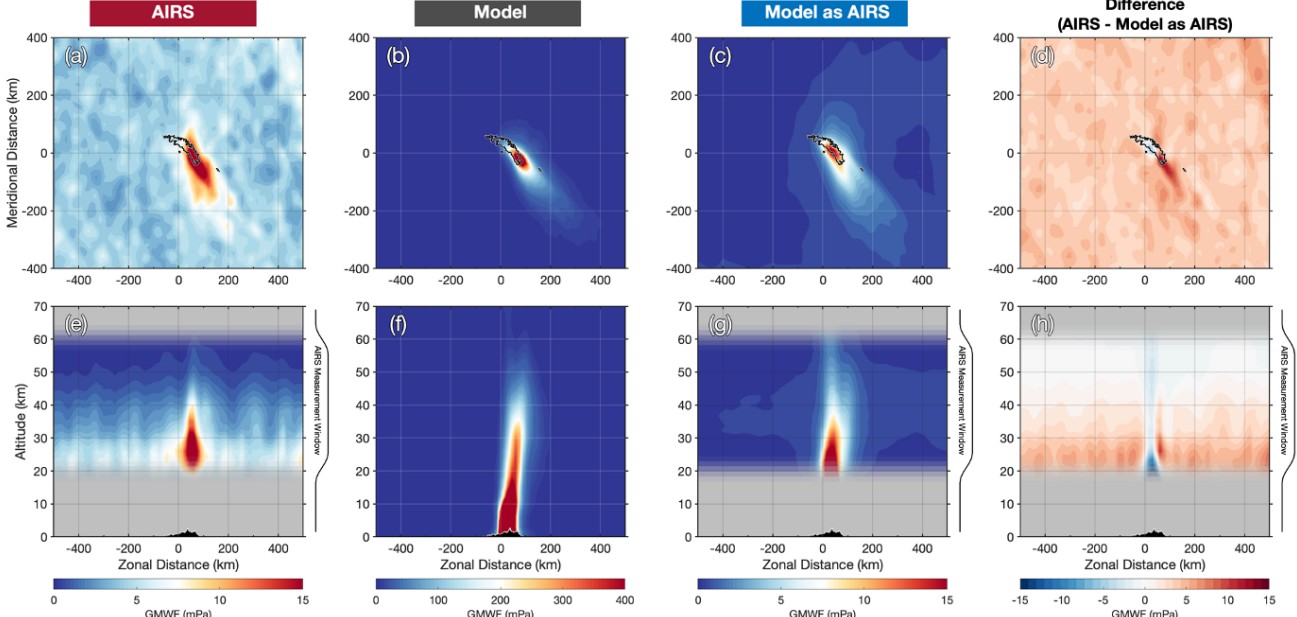

**Figure 14.** Cross-sections of the mean absolute gravity-wave momentum flux (GWMF) from the AIRS observations, the full-resolution model and the model-as-AIRS for both campaigns during July 2013 and June-July 2015. The top row shows a horizontal cross-section through the domain at 30 km altitude, while the bottom row shows a vertical cross-section at $y = 0$ km, where $y$ is the meridional direction. Panels (d) and (h) show the differences between mean GWMF from the AIRS observations and the model-as-AIRS for the horizontal and vertical cross-sections respectively. Both panels in each column share the same colour scale below. The black lines to the side of panels (e), (g) and (h) illustrate the AIRS vertical measurement window. The topography of South Georgia is shown in black at the bottom of panels (e-h).

## 4.5 Campaign-mean momentum fluxes

A useful quantity to constrain is the long-timescale mean of stratospheric momentum flux from South Georgia. These values are useful for simplified parameterisation schemes and model tuning for climate simulations.

Figure 14 shows horizontal and vertical cross-sections through the campaign-mean absolute momentum flux derived from the AIRS, full-resolution model and model-as-AIRS data for both June-July 2013 and 2015. The horizontal cross-sections reveal, as expected, localised maxima in momentum flux over the island extending eastwards and to the south in all three data sets, with values in AIRS and model-as-AIRS peaking exceeding 15 mPa. The model-as-AIRS flux in Fig. 14c is however more tightly localised over the south-eastern part of the island than is observed in AIRS in Fig. 14a. As a result, the area-average momentum flux values in the model-as-AIRS are lower. The AIRS flux is centred on the same location, but large flux values are more spatially distributed into the beginnings of a characteristic bow-wave shape. This reduced area-average flux in the model-as-AIRS can also be seen in Fig. 9 and 10 in Sect. 4.1, where model-as-AIRS fluxes are generally lower than AIRS measurements during bursts of coincident wave activity. As we would expect, both the AIRS observations and the model-as-





AIRS exhibit flux values that are more than an order of magnitude lower than in the model, whose values exceed 600 mPa in Fig. 14b, and are tightly localised over the same region as the model-as-AIRS (the colour scale is saturated to show the spatial distribution clearly).

These results suggest that the downwind bow wave pattern formed to the east of the island observed in AIRS measurements occurs is under-represented by the model simulation, since the downwind mountain wave pattern is far less apparent in the model-as-AIRS. Fig. 14d highlights this difference between the campaign-average AIRS and model-as-AIRS fluxes. The AIRS observations exhibit around 10 mPa more flux away from the island to the south east than in the model-as-AIRS, but around 5 mPa less over the island itself.

The vertical cross-sections in Figs. 14(e,f,g) reveal a similar picture, with a localised maximum of momentum flux directly over the south-eastern part of the island. With the AIRS vertical measurement window applied, the AIRS and model-as-AIRS fluxes maximise between altitudes of around 20 to 30 km before slowly decreasing with increasing altitude, while the full-resolution model shows an intense vertical column of flux from the surface to around 40 km altitude before doing the same.

The vertical difference plot in Fig. 14h shows that between altitudes of 20 to 30 km AIRS measurements exhibit slightly 835 more flux just to the east of the island but significantly less directly over the island compared to the model-as-AIRS. The model-as-AIRS also exhibits fluxes around 2 to 10 mPa larger than AIRS in a localised vertical column directly over the island at all altitudes, but smaller fluxes over a horizontal area average. Again, the high resolution model fluxes are at least an order of magnitude larger than observed in AIRS and the model-as-AIRS, with average values exceeding 300 mPa at altitudes around 30 km and more than 1000 mPa below 10 km (once again the colour scale is saturated to show the spatial distribution clearly). 840 This is due to large wave amplitudes at very short horizontal wavelengths in the model that are not visible to AIRS.

### 4.6   Wave amplitude and momentum flux growth with altitude

Figure 15 shows campaign-mean measured wave amplitudes and absolute momentum fluxes against height for a vertical column with horizontal radius $r = 50$ km centred on a location 50 km east of the model centre. This region was selected so as to capture the peak of the flux distributions based on the results of the cross-sections in Fig. 14.

Campaign-mean fluxes in Fig. 15a show reasonable agreement in AIRS and model-as-AIRS derived fluxes between altitudes of 25 and 45 km, although the AIRS fluxes are slightly higher, consistent with what we have seen in previous sections. Above altitudes of 45 km however, the AIRS fluxes are lower than the model-as-AIRS values. The model exhibits flux values just over an order of magnitude higher than AIRS and the model-as-AIRS. This is due to the fine horizontal scale structure in the wave field, as illustrated in Fig. 4b. All three data sets show a general decrease in gravity-wave momentum flux with increasing 850 altitude, except for the lowest region of the AIRS fluxes where an increase is observed between altitudes of 20 and 25 km. This is due to the half-bell tapering functions that we applied to the upper and lower boundaries of usable AIRS measurements (shown on the right hand side of panel (c)), which slowly reduced the perturbations to zero below these altitudes as illustrated. This reduction is not physical but an artefact of our method, and we would expect that in reality the red line denoting AIRS flux in Fig. 15a is likely to continue to increase below 25 km altitude following the blue model-as-AIRS line.





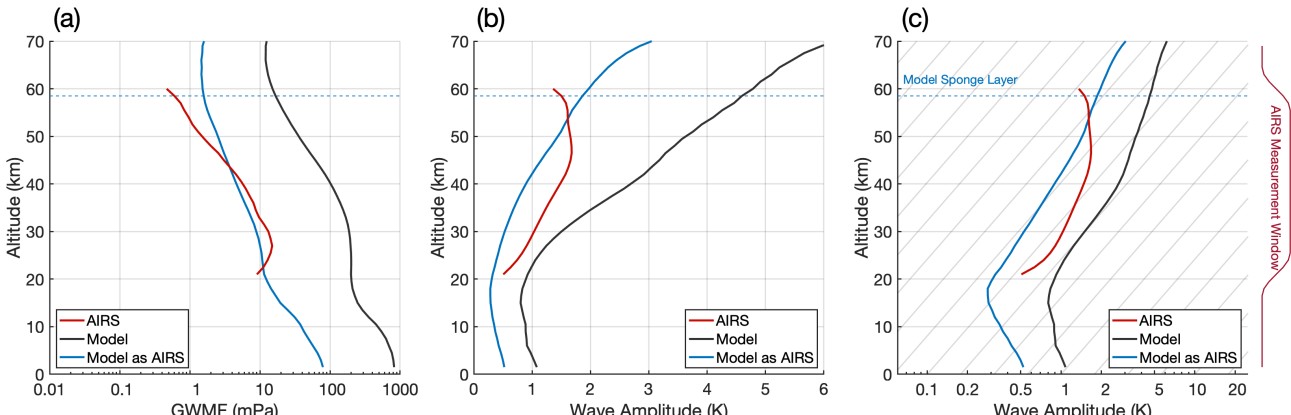

**Figure 15.** Vertical profiles of the campaign-mean absolute gravity-wave momentum flux (a) and measured wave amplitudes (b,c) against height for the AIRS observations (red), full-resolution model (black) and model-as-AIRS (blue) for June-July 2013 and 2015. Panel (c) is as panel (b) but with a logarithmic x-axis. Grey diagonal lines in (c) show the exponential adiabatic growth of gravity-wave amplitudes with altitude $e^{\frac{z}{2H}}$ expected from theory for an atmospheric scale height of $H = 7$ km. The dashed blue line indicates the start of the model damping layer which begins at $z = 58.5$ km and extends to the model top.

Figures 15b and 15c show campaign-mean measured wave amplitude against height in the same vertical column over South Georgia. In the model and model-as-AIRS, wave amplitudes are found to decrease slightly with increasing height from the surface to the tropopause, then increase steadily through the stratosphere up to altitudes around 40 km, as expected from exponential amplitude growth of $e^{\frac{z}{2*H}}$ under dry conditions, where $H = 7$ km is the average scale height for the atmosphere. Following the approach of Wright et al. (2016b), thin grey diagonal lines in Fig. 15c show this theoretical growth.

Wave amplitudes in the model and model-as-AIRS in Fig. 15c closely follow lines of exponential growth between altitudes of 20 to 40 km. Above 40 km altitude however, exponential amplitude growth in the model declines sharply, along with a smaller decline in the model-as-AIRS. This is likely to be related to the wave field in the model reaching saturation above these altitudes as the wave fields experience super-adiabatic conditions. The reduced measured amplitudes in the spectral range of the model-as-AIRS data as shown in Fig. 15b would make this condition less likely, so the reduction is less abrupt. Above an

altitude of 58.5 km, the model damping layer is applied as described in Vosper (2015), so results for measured wave amplitudes are not likely to be physical at these heights. Note also that the reduction in wave amplitude below 25 km altitude is due to our usable vertical measurement window and not likely physical as mentioned above.

An interesting observation is that between altitudes of 25 to 45 km, average wave amplitudes in AIRS measurements increase slower with increasing altitude than both the model and the model-as-AIRS in Fig. 15c, and significantly slower than expected

from idealised conditions using a scale height of $H = 7$ km. This could be indicative of increased wave breaking, saturation or dissipation effects that occur in the real atmosphere that are not accurately simulated in the model, or it could suggest that the atmospheric scale height in this region was greater than 7 km during this period. It is also possible that this could be a result of





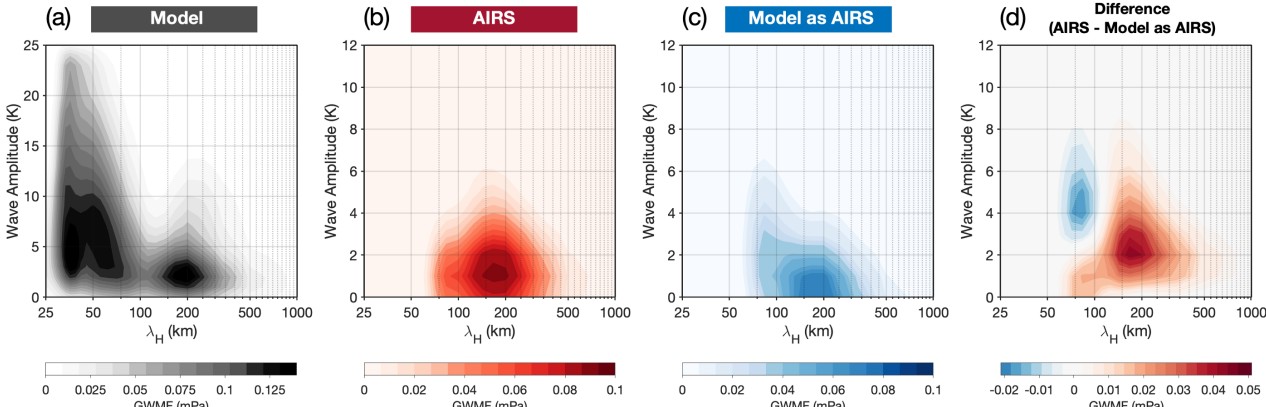

**Figure 16.** Average absolute gravity-wave momentum flux as a function of wave amplitude and horizontal wavelength $\lambda_H$ over the whole model domain between altitudes of 25 to 55 km for (a) the AIRS measurements, (b) the model and (c) the model-as-AIRS during June-July 2013 and 2015. Panel (d) shows the difference between the AIRS measurements and the model-as-AIRS.

changes in the AIRS vertical resolution or noise levels at these altitudes, but the AIRS measurement noise actually increases significantly above altitudes of 50 km (Hoffmann and Alexander, 2009; Hindley et al., 2019) so this is unlikely to be the cause

of this reduction in wave amplitudes. If it were due to reduced AIRS vertical resolution with altitude, we would expect to see a similar sharp reduction in the model-as-AIRS measured amplitudes, which we do not. We therefore suspect this change may be due to a physical process.

Given the results of the previous sections, and in particular taking into account the variability in the radiosonde-observed winds, we propose that this sharper reduction in AIRS amplitudes above 45 km altitude is due to a reduced likelihood of

mountain wave structures being stable up to altitudes above 45 km in the real atmosphere due to variability in surface and stratospheric winds.

### 4.7 Distribution of GWMF with wave amplitude and horizontal wavelength

Our final analysis concerns the spectral properties of the observed and simulated wave fields. In the previous sections we have investigated the distribution of gravity wave amplitudes and momentum fluxes in the geographical region around South

Georgia. Here we investigate how the campaign-average momentum fluxes are distributed as a function of wave amplitude and horizontal wavelength. This analysis will help us to understand what features of the wave field may give rise to the discrepancies reported earlier in the study.

For a range of wave amplitudes between 0 and 25 K and a range of horizontal wavelengths between 25 and 1000 km, the average absolute momentum flux is found for each amplitude and wavelength combination during all overpasses and timesteps

in June-July 2013 and 2015. Average values are taken over the same geographical region as used in Figs. 8 to 10 and between altitudes of 25 to 45 km. As before, regions that do not have a particular amplitude and wavelength combination are set to zero





before the average is taken to give the true area average. This yields values that are quite low compared to values shown in previous figures. The results of this analysis for AIRS measurements, the full-resolution model and model-as-AIRS are shown in Fig. 16.

In each panel in Fig. 16, colours indicate the area-averaged momentum flux for measured waves with a specific amplitude and wavelength. In panel (a), two distributions of waves become clear in the model: one group with large amplitudes up to nearly 25 K and horizontal wavelengths between 30 and 75 km and a second group with amplitudes below around 5 K and horizontal wavelengths between 100 and 300 km. Given the horizontal scale of the first group, and the results in Fig. 14, this is likely to correspond to the very short horizontal scale waves directly over the island, while the second group likely corresponds

to the mountain wave wake region and non-orographic wave activity in the model.

        In the AIRS and the model-as-AIRS distributions in panels (b) and (c), only the second group of waves is apparent, where largest average momentum fluxes occur for horizontal wavelengths between 150 and 200 km. This peak occurs at amplitudes between around 0.5 and 2 K in AIRS measurements, but typically below 1 K in the model-as-AIRS. Panel (d) shows the difference between the AIRS and model-as-AIRS flux distributions, where up to 50% more average flux is observed in AIRS

for amplitudes larger than around 1 K and horizontal wavelengths longer than around 100 km. The only part of the distribution where slightly more average flux is found in the model-as-AIRS occurs for amplitudes between around 3 and 6 K and horizontal wavelengths near 75 km. This increased flux towards larger amplitudes and shorter horizontal wavelengths is apparent in the model-as-AIRS distribution in panel (c), which is of course simply a subset of the model distribution, and is similar to that of the model before the observational filter of AIRS removes flux from short horizontal scale waves to near zero.

These distributions suggest an interesting conclusion. The AIRS distribution in Fig. 16b does not show the same tendency towards larger amplitudes at horizontal wavelengths shorter than 100 km in the same way that the model and model-as-AIRS does. In the model and model-as-AIRS, these shorter horizontal wavelength waves with large amplitudes correspond to the mountain wave field directly over and immediately downwind of the island itself, as shown in Figs. 6 and 12. The fact that the AIRS measurements do not appear to follow this distribution could suggest that such short horizontal wavelength waves

over the island, which carry the largest momentum fluxes individually via Eqn. 1, are not as observed so commonly in the real atmosphere as they are in the model.

        This is not likely to be an AIRS horizontal resolution issue, since the island is usually located near to the centre of the AIRS swath for the overpasses we have selected in Sect. 2.1, where the spacing between AIRS pixels is as low as 15 to 20 km, yielding a Nyquist resolution of around 40 km. It is interesting therefore that in the AIRS distribution we do not see the beginning of

distribution of the first group of short-wavelength waves as seen in the model in panel (a) or the model-as-AIRS in panel (c). This is is also unlikely to be an artefact introduced by the application of the AIRS observational filter to the model to generate the model-as-AIRS, as described in Sect. 3.1. This is because the same method is followed for both data sets: measurements are gridded onto our 15 km × 15 km regular horizontal grid then convolved with a Gaussian with a horizontal FWHM equal to 40 km × 40 km. The model is downsampled onto this grid, whereas the AIRS measurements are slightly upsampled, so if any

artefacts were introduced by this process we would expect to see an underestimation of the flux from short horizontal scale





waves in the model-as-AIRS due to the downsampling. We do not see evidence of this, so it is likely that this increased in flux at short horizontal scales in the model-as-AIRS compared to AIRS is physical.

One possible reason for this could be due to variability in the background wind. Mountain wave fields require a stable vertical column of strong tropospheric and stratospheric winds in order to propagate vertically into the stratosphere without encountering critical levels. Here, hourly boundary conditions for the local-area domain are provided by a global forecast which is initialised at midnight each day and integrated forward in time in hourly steps. These hourly boundary conditions are then linearly interpolated in time to the time step of the local-area simulation. One issue with this approach is that variability of the background winds on timescales shorter than one hour will not be present in the local-area model. Further, the reduced vertical grid of the global forecast, which has only 70 vertical levels, will mean that the boundary conditions will exhibit reduced wind variability in the vertical.

These factors could result in reduced variability of the background winds in the local-area model. This could potentially provide more favourable conditions for the generation of mountain waves with short horizontal scales, since it is these short horizontal scale waves that would the be first to be disrupted by sudden changes in the wind vector. This could explain the discrepancy in Fig. 16d, where the model-as-AIRS exhibited more momentum flux at short horizontal wavelengths and less at long horizontal wavelengths than observed in AIRS measurements.

## 5    Discussion

In this study we used a range of different analysis techniques to investigate the similarity between gravity waves in a high resolution local area model over South Georgia and coincident satellite observations. The results of these analyses, as listed in their respective sections above, are quite varied. Here we combine the key findings of this study to discuss them in a broader context.

Our interpretation of the results in this study is that, when allowed to run at a high spatial resolution, a mountain wave field over the island tends to form in our local area model configuration that is somewhat over-idealised compared to observations. Specifically, this idealised wave field in the model consists of a central region of large amplitude, short horizontal wavelength waves that is tightly located over the island. Our results in Figs. 12, 14 and 16 indicate that this feature is not so commonly seen in observations.

This over-localisation of flux over the island in the model could arise due to reduced short-timescale variability in the background wind. As discussed earlier, the lateral boundary conditions of the local-area model are supplied hourly from an N512 global forecast with reduced vertical resolution. Variability in the background wind over timescales shorter than one hour, or over short vertical scales, might not be accurately simulated in the local-area model. This could provide more favourable conditions for the generation of short horizontal scale mountain waves than is observed in the real atmosphere. Although our radiosonde comparison in Sect. 2.4 showed reasonable agreement between observed and simulated winds over monthly time scales, it was found that significant discrepancies could occur for individual measurements.





We also observe a greater distribution of gravity wave activity at larger horizontal distances from South Georgia in Sect. 4.4, particularly upwind of the island to the west. This is a strong indication of an under-representation of the amplitudes and fluxes

of non-orographic waves in the model. These waves, whose physical scales can be very large, as seen in Fig. 11j or in Fig. 1c of Hindley et al. (2019), were found to have around at least 50% larger amplitudes in the AIRS observations than in the model-as-AIRS. The lack of these large scale non-orographic waves could also contribute to the overly-stable wind vector that allows fine-scale horizontal mountain wave structure to form in the model. This is because a large amplitude non-orographic wave passing through the region in the real atmosphere could disrupt the background horizontal wind vector, resulting in non-linear

interaction and dissipation of the fine horizontal scale mountain wave field.

We also showed in Fig. 12 that the short-timescale variability, or intermittency, of gravity wave momentum flux is significantly lower in AIRS observations than in either the high resolution model or the model-as-AIRS. Although retrieval noise in the AIRS measurements may play a part in these results, this result is consistent with a more uniform distribution of gravity wave activity over time in observations. In the model, mountain waves may be somewhat over represented, leading to

discrete bursts of gravity waves activity and higher intermittency during our period of study. An under-representation of large-scale waves from non-orographic sources in the model, which have been found to exhibit lower intermittency than orographic sources, would also be consistent with our results. One possible cause for reduced from transitory non-orographic wave activity in the model could be that these waves are present in the hourly global forecast, but they do not survive the transmission from the forecast grid to the local-area domain through the model boundary conditions.

Thus, our answer to the question posed in Sect. 1 is two-fold. Generally speaking, the high-resolution model over South Georgia used here is found to produce a gravity wave field over South Georgia that is in good agreement with coincident observations. The timing and directionality of gravity wave momentum fluxes closely matches observations. Although area-averaged momentum fluxes are around 25% lower than observed, the agreement is significantly better than would be expected in a coarser resolution free-running GCM. This indicates that increasing model resolution can improve the representation of

gravity waves from small islands through the accurate determination of grid cells containing mostly land or sea, as discussed by Vosper et al. (2016). This leads to further improvements such as the accurate direction of the meridional momentum flux from South Georgia, shown in Sect. 4.1, which is observed to be overwhelmingly southward in both the model and observations. This is important because, if the island orography was underrepresented in a lower resolution GCM such that it was effectively a point source, the net meridional flux for a westerly wind vector would be close to zero. The high-resolution simulations allow

the orientation of the island's topography to be properly represented, with a larger cross-section to southwesterly winds. This results in a significantly larger southward component of meridional flux, which is in good agreement with the observations without the need for model parameterisations.

However, we can also conclude from our results that, if the background wind vector is not accurately simulated, increasing the model spatial resolution can lead to inaccurate gravity wave generation at different horizontal scales. Further, for a local

region considered in this study, accurate simulation of transitory non-orographic gravity waves propagating in and out of the region is also needed in order to produce a realistic local field of gravity wave activity. It is also important to note that it is not just model resolution which is important for accurate gravity wave simulations. Model numerics can also be significant.



The Unified Model, for example, uses semi-implicit time integration for operational efficiency, but choosing too large a time step can make the model dissipative to gravity waves (e.g. Shutts and Vosper, 2011; Vosper, 2015). This could lead to an
underestimation of gravity wave amplitudes in the model, which could partly explain the reduced amplitudes and momentum fluxes in the model-as-AIRS compared to AIRS measurements. However, we find that wave amplitudes appear to decay with height at a consistent rate in Fig. 15 in the model and model-as-AIRS, which could suggest that this effect may be small. As mentioned above, it is interesting to note that the AIRS amplitudes appear to decay faster with height than the model-as-AIRS in Fig. 15, suggesting that wave amplitudes may, in general, dissipate more quickly with altitude in the real atmosphere than in
the model.

## 6 Conclusions

In this study, we posed the fundamental question: when a numerical model is allowed to run at very high spatial resolution over a small mountainous island like South Georgia, how realistic are the simulated gravity waves compared to observations? The answer to this question is key to the development of the future generations of global climate models. As spatial resolution is
increased, a larger portion of the gravity wave spectrum can be resolved, thus reducing the reliance on model parameterisations which are poorly constrained by observations. The increased gravity wave fluxes from these resolved waves may in turn lead to significant reductions in model biases, such as the cold-pole problem. In the shorter term, high resolution modelling experiments are being used as reference simulations to test gravity wave parameterisations (both orographic and non-orographic) so it is important to understand their validity (Vosper, 2015; Vosper et al., 2016).
To answer this question, we compared simulated gravity waves in a high resolution local area model over South Georgia to coincident 3-D satellite observations from AIRS/Aqua. We applied the AIRS observational filter to the model output to produce a model-as-AIRS dataset that could be compared the observations directly. Our results show that:

1. Overall, the timing and magnitude of gravity wave activity in the model-as-AIRS is in reasonable agreement with observations, but area-averaged momentum fluxes can be up to around 25% lower than observed in AIRS measurements.

2. In both the model and observations, meridional momentum fluxes over the island are overwhelmingly southward, something that would not be accurately simulated if the island was under-resolved in a GCM.

3. Directly over the island, the model-as-AIRS exhibits higher individual flux measurements but is more intermittent than the observations, with 90% of the total flux carried by just 22% of wave events, compared to 32% for AIRS.

4. 72% of the total flux is located downwind of the island in the model-as-AIRS, compared to only 57% in the AIRS
measurements. This suggests that non-orographic wave activity observed in AIRS is under-represented in the model, which is supported by directed inspection of the wave fields.

5. Gravity wave momentum fluxes in AIRS appear to dissipate more quickly with increasing height than in the model-as-AIRS, which could suggest that the model may under-represent wave-mean flow interactions.



6. Spectral analysis results, combined with spatial distributions, indicate that our high resolution model runs over-estimate the fluxes of large amplitude, short horizontal scale mountain waves directly over the island, but under-estimate the fluxes of large amplitude, large horizontal scale waves in the surrounding region. This could be due to reduced short timescale variability of the background wind in the model.

7. Model winds exhibit a slight southward bias of around 5 to $10\,\mathrm{ms^{-1}}$ with increasing altitude compared to coincident radiosonde observations.

In conclusion, we find that although increasing the horizontal resolution of models can improve the representations of small islands such as South Georgia, this can also lead to an over-representation of small-scale mountain wave activity and an under-representation of large-scale non-orographic waves. Our results suggest that, in order to produce a realistic simulation of wintertime stratospheric gravity waves over small islands in the Southern Ocean in future GCMs, a holistic approach is needed. This approach should include: (a) sufficient spatial resolution to resolve the island topography; (b) accurate wind speed, direction and short-timescale variability over the island and the wider geographical region; and (c) accurate simulation of large-scale non-orographic waves from sources such as geostrophic adjustment processes around the vortex edge that may be found in the stratosphere over small islands during winter. As the spatial resolution of future GCMs is increased ever further in the coming years, we must ensure that each of these factors is implemented correctly in order to produce realistic gravity wave characteristics over small islands in the Southern Ocean.

*Author contributions.* The investigation was carried out as part of the South Georgia Wave Experiment (SG-WEX) project, designed by NJM, AMG, DRJ, JCK, TMG and ANR. The modelling simulations were designed and carried out by SBV, ANR, and JKH. The acquisition and analysis of these data was performed by NPH, with assistance from ANR and SBV. The specialised 3-D temperature data from the AIRS satellite was provided by LH. The radiosonde field campaign on South Georgia was conducted by CJW and ACM with logistical support from TMG and JCK. The scientific investigations, computational data analysis and publication figures for this study were developed and produced by NPH. The written manuscript was prepared by NPH, CJW and NJM with contributions from all authors.

*Competing interests.* The authors declare that they have no competing interests

*Acknowledgements.* The SG-WEX project is supported by the UK Natural Environment Research Council (NERC) under grant numbers NE/K015117/1, NE/K012584/1, NE/K012614/1, NE/R001391/1 and NE/R001235/1 and a Royal Society University Research Fellowship UF160545 supporting CJW. The authors would like to acknowledge the use of supercomputing and data archival systems at several institutions, with a special thanks to their support staff. These are: the Met Office and NERC Supercomputing Node (MONSooN), the JASMIN data analysis platform at the UK Centre for Environmental Data Archival (CEDA), the Managed Archive Storage System (MASS) at the UK Met Office, the BALENA high-performance computing service at the University of Bath and the JURECA high-performance computing service





at Forschungszentrum Jülich. We would also like to acknowledge logistical staff at British Antarctic Survey involved in the radiosonde campaign and the cooperation of the government of South Georgia and the South Sandwich Islands. Finally, we would like to thank the NASA AIRS instrument team and the many teams involved in developing the Met Office Unified Model.

1055



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
