# Peer review of "Stratospheric gravity waves over the mountainous island of South Georgia: testing a high-resolution dynamical model with 3-D satellite observations and radiosondes"

_Atmospheric Chemistry and Physics, 2020_

## Referee Comment (RC1) · Anonymous Referee #1 · 6 Jul 2020

General comments

In this manuscript the authors address an important issue for the adequate representation of gravity waves (GWs) in the numerical modelling of the atmosphere. They study the possible impact of the use of a high horizontal resolution grid in the stratospheric simulations around a mountainous island. The overall presentation is well structured and clear with nice figures. However, some of them have been placed uncomfortably far from their first citation. Some arguments in the text (as explained below) remain too hypothetical without further deepening and need to be more profoundly substanti-

ated to support some interpretations and conclusions, e.g. the absence in AIRS data of short horizontal wavelength GWs shown by the numerical model or vice versa with non-orographic waves. There are also some uncertain definitions or descriptions that need a clarification. Finally, I believe that the authors acknowledged only a fraction of the relevant work related to the study of GWs through satellite data and numerical modelling over the last decade in the nearby well-known Southern hotspot.

Specific comments

Introduction: As stated above, I believe that some significant work in relation to GWs studied with satellite data and numerical modelling in the Southern GWs hotspot should be included in the context of this section.

l.134-135 and l.147 There is a need for clarification about some definitions. If vertical resolution is at best 7 km (uncorrelated or independent successive data), then the 3 km sampling is just some kind of interpolation and no wavelengths shorter than 14 km may be detected, which would have a disastrous effect on the following results. Also l.353-355.

l.138 If temperature uncertainty may be up to 1.5 K you need to justify how you rely on results of 1 K amplitude or even less.

l.143 Please mention the artefacts that may remain.

Section 2.2: More details of the simulation characteristics are needed. You should mention the type of sponge and its intensity and the timestep that was used. Did numerical instabilities arise during the initial steps? If so, how did you handle them? Did you assess the model spin up? Did simulations exhibit alterations with slightly earlier or later initial time? In addition, your operational analyses have a 46 km resolution, whereas your simulation domain has a 1.5 km grid. Have you evaluated the possible effect of this factor of 30? Wouldn't it be advisable to use a smaller ratio? May this fact be responsible for the model not being able to adequately represent the non-orographic

GWs (l.403-406, l.972-974)? How reliable are simulations if such a large structure is not "transmitted" from the forecast to the local area model?

L.236-237 You should also compare the numerical model with radiosonde temperature (not only wind validation) as you have it at disposal, but you should not use it for GWs as you clearly stated in l.254-256.

l.298 There may be significant positional errors? How large can they be?

l.449-453 The redundancy of the method should be shortly discussed or cited as it is strongly related to the reliability of the calculated GWs amplitudes. This is especially important in the context of some notable amplitude discrepancies below among AIRS and the model.

l.490-491 You should check your hourly output for stationary phases and increasing wind speed with height.

l.643-645 You should use your model simulations to test this argument.

l.650-651 The model simulations should give you a clue for upward or downward phase propagation.

l.678 Does this imply that AIRS amplitudes are typically the double of the model? If so, explain.

l.701-702 The presence of the jet, the polar vortex, storms and fronts can all be probably checked from your operational analyses in order to verify the support to your argument.

l.706-714 Please use your hourly simulations to verify at least partially in the mentioned geographical domain your detachment or moving secondary waves argument.

Table 1: To check if differences are significant it is necessary to include uncertainties with the averages.

Figure 14d, h: Please discuss the parts where the difference is larger than the absolute fluxes.

Figure 16: How can you define a unique amplitude if it can change by a factor over 4 from 25 to 55 km?

l.896-897, 913-916 and 919-920 This could indicate that AIRS may be omitting an essential contribution to GWs momentum flux and its later parametrization in global models. This fact merits a quantification of the above effect due to the possible discrepancies of simulations or observations in this work with the real atmosphere.

l.936-940 Can you give a reference where this effect has been quantified? How likely is it that this high frequency wind variability exists in that zone? Can you draw conclusions from the individual radiosonde profiles?

l.951-957 Again, another possibility is that AIRS is missing these GWs.

l.993-1000 What was the expectation for GWs amplitudes in your simulations according to the timestep you have chosen? Was it in agreement with your results?

l.1024-1027 Please check if further analysis in the previous sections produces any modifications.

Minor technical corrections or comments

l.28 waves play a

l.42 applied to

l.92 we use

l.251-254: The problem of inferring certain GWs parameters from slanted radiosonde profiles has already been addressed more than 20 years ago (see e.g. doi: 10.1175/1520-0450(1995)034<2747:TIOWAP>2.0.CO;2).

l.281 in -> is

l.284 "."

l.289 "is" or "could be"

l.309 set

l.403 in either

l.435 of our

l.474 are have

l.476 than

l.499 westward in the intrinsic system

l.500 in a wave

l.547-548 Rephrase

l.648-650 Rephrase

l.681 selected

l.700 clear clear

l.718 of

l.826 occurs

l.862-863 Please include reference.

l.921 is is

l.968 results result

l.975-976 over South Georgia. . . over South Georgia

l.976 was found

l.977 match

l.1012 compared to

---

## Referee Comment (RC2) · Anonymous Referee #2 · 14 Jul 2020

**Stratospheric gravity-waves over the mountainous island of SouthGeorgia: testing a high-resolution dynamical model with 3-D satellite observations and radiosondes, Hindley et al**

This paper presents an extensive comparison of wintertime gravity wave activity and gravity wave properties in mesoscale model simulations and AIRS satellite observations above and around South Georgia. The authors have analyzed numerous aspects of the gravity wave field using state of the art methods and visualization. In the end, the comparison is summarized in 7 key findings addressing among other things the amount and distribution of momentum flux in the model. Overall, the paper presents interesting results that can help gain a better understanding of gravity waves in mesoscale model simulations without gravity wave parametrization. However, methods, results, and discussion are not strictly separated in the paper and, due to the length of the paper, this makes it hard to stay focused during reading. Results are often directly followed by 1-2 sentences of discussion (sometimes speculation). I recommend to address the three major remarks mentioned below to improve the paper (mainly readability) before final publication.

**Major remarks**

I) Why don't you use a similar 4th-order polynomial fit as for the AIRS data to determine temperature perturbations that contain orographic and non-orographic GWs? Using SG and smoothed nSG to determine perturbations seems nice way to get the contribution of the orographic GWs in the model but doesn't immediately sound like the best choice for a comparison AIRS (unless you can show that 4th-order polynomial leads to similar perturbations as the procedure described here and/or your results are not very sensitive to the background removal). [Moreover, I realized that contributions of non-orographic waves seem to be important in several sub-sections later in the manuscript. SG and nSG could be used to separate and quantify some of the non-orographic contributions in the model simulations (not all the analyses need to be done but for some quantities it could strengthen the findings and conclusions with respect to the non-orographic GWs).]

II) Structure of the paper:
- strictly separate "data and methods" and "results"
  This would also mean help the reader to already know by the end of Sec. 2 what to expect in the result section of the paper.
  - First part of Sec 3 and 3.1 describing the data processing is better moved to Sec 2 (which could then be called Data and Methods)
  - First part of Sec 3.3 Should be moved to Sec 2
  - First part of 3.3.2 should be moved to Sec 2.
  - Sec 4.3: Gini coefficient can be introduced in Sec. 2
  - First part of Sec 4.4. should be moved to Sec 2.
  - Sec. 2.4 already presents results and could be moved to Sec. 3 (or create new Sec 3 with only content of Sec. 2.4)
- separate results and discussion
  - L397: This sentence can be left for later discussion.
  - L403-407: This sentence can be left for later discussion.
  - …and so on

III) Subjective and expletive words like "overwhelmingly" or "very" (>20 occurrences) can be reduced without losing information. There is also a large amount of speculation (some contradictions, some repetitions) in the paper (>25 occurrences of likely and >30 could explanations) that lack quantification.
Just some examples: "…should result in simulated conditions over South Georgia that are very close to reality for the given time periods." "…time separation is very small and the local wind conditions

can be expected to be very close to reality." "A small fraction of this distribution is likely to be measurement error, but the results may still be significant.", "Since these are clear mountain wave structures, it suggests that this could be due to errors in the speed and direction of the background wind in the model.")

I recommend looking through the paper and deciding if such expressions/sentences are essential for the main content/message of the paper and if they can be justified or quantified. If not, they could be removed. Instead of listing every possible explanation for some of the observed differences between model and observations, the explanations could be limited to the one or two most relevant ones.

**Minor comments**

L9: "high" instead of „very high"; you may want to add "without gravity wave parametrization" over South Georgia.

L23: please specify which scales are meant by short and long

L40: not all but "a large amount of these short vertical and horizontal scales are too small to be resolved even in recent GCMs"; pls add more recent citation (e.g., Plougonven et al 2019, How does knowledge of atmospheric gravity waves guide their parametrizations?)

L43: In some cases? Isn't it rather the norm than the exception?

L84: GWs can propagate large horizontal distances, and from this point of view the Andes are not too far at all. (compare L705: The island lies only 2000 km east of the southern tip of South America, a region associated with the largest stratospheric mountain wave activity observed anywhere in the world)

L88: range of scale sizes? Please clarify.

Fig. 1: Why are the soundings of January 2015 shown here? They are not relevant for the content manuscript. (see also comment on L190)

L137: "…to study gravity waves." Not the whole spectrum of gravity waves is small scale.

L140: Is the fit applied horizontally or vertically?

L176: …much finer than the 3 km vertical grid of the AIRS retrieval: this is kind of a change in the objective of the paper. "when a model is allowed to run at very high spatial resolution over South Georgia, how realistic are the simulated gravity waves compared to observations?" vs how realistic are simulated gravity waves in the observational window of AIRS?

Moreover, can the vertical grid spacing of the model be directly compared to the vertical grid spacing of the retrieval? At least in the horizontal effective resolution is more like 5-10 times the grid spacing.

L183: Does "no gravity wave parametrization" also mean no non-orographic parametrization?

L190: I would expect that there wasn't much mountain wave activity at all in the stratosphere in summer, so I think January 2015 can be omitted.

L194: Can you revise this sentence being more specific and naming the simulated conditions you are interested in, i.e. gravity waves. Then a large part of the wave spectrum can be expected to be close to reality but not the small scales.

L196: This sentence can be omitted.

L213: How can this have an effect at all on the data above 20 km? Is this due to the analysis performed later on?

L235: Was the radiosonde data assimilated in the operational analyses? This should be mentioned here.

L244: Do you mean a wind reversal in the meridional wind? Meridional wind direction is also changing at 30 km on 21$^{st}$ of July and end of July 2013.

L271: Measurement errors and artifacts should be removed from the measurement data before doing the comparison. They are not physically meaningful and are too obvious in the profiles (especially in Fig. 3b, d but also in Fig. 3g above 15 km). Moreover, it would probably help to filter the small scale fluctuations in the sounding data that are well below the vertical resolution of the model data. Fig 3b, e would then look smoother and easier to compare to 3c, d.

L286-290: "slight southward directional bias", "more northward": please revise this paragraph. The wording is very circuitous. It's easier to just say that the model tend to slightly overestimate (underestimate) the southward (northward) winds in the mid-stratosphere. Because the mean difference is zero for the zonal component, this then not only tends in a small directional bias but also in a bias in the horizontal wind speed.

L287: the initial and boundary conditions

L293-L299: In my view, this paragraph is too speculative and can be omitted. Moreover, real time forecast of one to multiple days is different from short-term forecasts of up to 6h used here. Positional errors larger than then the horizontal grid-spacing of the model (everything smaller than that does not really an influence) are hopefully not contributing to the spread because they do not occur (or rarely occur and should then be removed from the sounding data before doing the comparison).

L301: Comparison is concluded and then starts again with discussing the surface winds. They are already included in L281 and local topographic effects are mentioned as possible reason. So L301-L306 can be removed. Detailed discussion of topographic differences between model and reality would include a comparison of the model topography to high-quality elevation data of the island. I don't think it's relevant for the rest of the paper.

L227: Can you specify what scales are meant? Vertically it's clear to me from Sec. 2 (8-9km) but not horizontally (3 times footprint size, e.g. > approx. 80km?).

L330: I cannot follow the reasoning of this sentence. Is this because the model runs without GW parametrization or why/how does the generation of long scale waves depend on the smallest scales?

Figure 4: It is probably better not to show the model data above 58 km where the damping layer is located. With the saturated amplitudes and vertical phaselines, it distorts the visual perception.

L464: applied to the

L470ff: Can you provide some values for a more quantitative comparison? For example, max. amplitude (and later on horizontal and vertical wavelength) at 20 and 40 km above the island and the downstream values you are referring to for both AIRS and model.

L523: Really "measurement error of AIRS" or rather an uncertainty in the analysis and determination of the sign of m?

Sec 4 "results": Section 3 contains already plenty of results. 4.1 could be just labelled 3.4 and so on

L643: I cannot follow. Isn't this a conclusion resulting from comparing model to model as AIRS? There is clearly more MF in the model outside the observational window of AIRS.

L979: It would be interesting to repeat the analysis with the output of the UKMO global configuration in the near future. Or was something similar already done in the past? If yes, you could add the reference here.

---

## Referee Comment (RC3) · Anonymous Referee #3 · 31 Jul 2020

This paper describes a comparison of various parameters of gravity waves (GWs) in the region of the South Georgia Island (South Atlantic) during wintertime as deduced from AIRS and real-date simulations using a regional circulation model. The papers is well organized and in most parts very well written. The figures are mostly of very high quality too. Furthermore, I believe that this is very worthwhile study showing the level of consistency that can be achieved in high-resolution models when compared to observational data. The reason for recommending major revision is due the fact that the authors do not sufficiently interpret the shortcomings of their model results. I believe

that the paper needs to be significantly improved regarding this aspect. The simulated very high GW amplitudes at small scales are suspicious. This may be related to a too coarse vertical level spacing in relation to the very fine horizontal grid. My major and minor comments are listed below in chronological order.

Major comments:

L169-176: The vertical resolution applied in the model is extremely coarse related to the horizontal resolution. The vertical grid-spacing is 0.6-2 km in the stratosphere, versus a horizontal grid-spacing of 1.5 km. This vertical grid-spacing in the stratosphere in not even sufficient to simulate a self-induced QBO in GCMs. More importantly, GCMs with explicit simulation of GWs (e.g., Watanabe et al., 2008, JAS: General aspects of a T213L256 middle atmosphere general circulation model) employ a vertical level spacing of 300-600 m throughout the middle atmosphere while the resolvable horizontal wavelengths in these models are of the order of 200 km. The necessity for a small enough vertical grid-spacing derives from the fact that the GWs resolved by the horizontal grid must not be spectrally biased in the vertical to too large vertical wavelengths. Indeed a too coarse vertical resolution artificially prevents the GWs from reaching dynamic or convective instability and thus being dissipating by the model's turbulent diffusion scheme.

L176-178: I do not find this statement very conclusive. The grid-spacing of a model as such does not say anything about the scales that are reliably resolved. It is the dynamical core (spatial resolution, numerics) combined with the subgrid-scale diffusion (either explicit or implicit) that determines the reliable scales of a model.

L193-196: See my 2 previous major comments and consider reformulation.

L137-348: When the model data are interpolated to a 15 km grid, the Fourier components with horizontal wavelengths shorter than 30 km must be filtered out beforehand to avoid aliasing errors from the scales below the 15 km grid. Did the authors apply this spectral filtering before re-griding the model data (for model and model-as-AIRS)?

If yes, please mention this point in the text for the sake of clarity. If not, the resulting aliasing could be an explanation for the high power in the GW amplitudes and in the MF at horizontal wavelengths of 30-40 km (e.g. Fig. 16a). In that case you might consider a substantial revision and re-submission of the paper.

L399: The authors should not only mention that model-as-AIRS produces too small amplitudes compared to AIRS, but also that the GW-phases of the MWs over the Island differ significantly in the two data sets (Figs. 4 and 5). Moreover, the slopes of the phase lines from x=100 km to 600 km in Fig. 4 differ in sign(!); that is, these GWs must propagate in different directions when comparing model-as-AIRS to AIRS. Please mention and discuss these dissimilarities.

L429-430: See my comments above: The horizontal structures in model-as-AIRS and AIRS are at best qualitatively similar over the mountain; they are dissimilar farther downstream. Please describe your comparison of results from model-as-AIRS and AIRS consistently with your high-quality figures.

Fig. 7: How did you apply averaging over the GW scales when calculating the MF. Furthermore, the regions of phases going upward with increasing x in Fig. 4c and f should give rise to a reversal from westward to eastward MF in Fig. 7c. Please clarify.

L489-493: The wave refraction argument can be applied for either upward propagating GWs (negative vertical wavenumber) or downward propagating GWs (positive vertical wavenumber). Here you apply this argument even though the longer vertical wave-lengths that you expect for a westward MW in an increasing stratospheric eastward jet show up in your plot with reversed sign. How do you explain the reversal from neg-ative to positive vertical wavenumber at 20-30 km in Fig. 6d? Why is there a noisy mixture of positive and negative vertical wavenumbers in Fig.6h? These wavenumber (wavelength) results need to be revisited.

L510-515 and L528-L532: This discussion relates to my previous comment. Please give a hint on why you possibly have positive vertical wavennumbers in AIRS. One

possibility is that the background wind in the lower atmosphere shows accelera-tion/decelerations which can cause the phase lines of MWs sloping upward/downward in time-height cross-sections. Another possibility is the generation of secondary GWs from MW breaking causing downward propagating GWs (which are no longer MWs). See also Vadas and Becker (2018, JGR Atmos.: Numerical Modeling of the Excita-tion, Propagation, and Dissipation of Primary and Secondary Gravity Waves during Wintertime at McMurdo Station in the Antarctic), as well as Vadas et al. (2018).

L599: Note that the wind in the lower troposphere is crucial for MW generation, while the wind at higher altitudes facilitates propagation (strongly eastward) or dynamical instability (weakly eastward or westward). Again, it is unclear how dynamical instabil-ity (including critical levels) are handled by the model, given its coarse vertical level spacing in the stratosphere and the lack of information about subgrid-scale processes.

L619-623: This is another example of a very speculative discussion about suspicious features in the model data. Are stationary, non-orographic GWs indeed present around the island in the global model? Are these waves artificial? Please clarify.

L638-645: How is the simulated very large MF at scales close to the horizontal grid-scale possibly related to the coarse vertical level spacing and, in addition, to insufficient parameterization of dissipation processes in the stratosphere below the sponge layer? Your model results would imply that the vast majority of MW momentum flux resides at horizontal scales not even observable by AIRS. Hence, according to your model re-sults, observations from AIRS are essentially useless to estimate the orographic GW MF from small Islands that is missing in global models? Please clarify.

Fig. 15: This is a very nice figure (like most of the other figures)! I cannot see the grey lines mentioned in the caption. My comment is this: The AIRS curves nicely indicate wave dissipation from about 25 km on. This wave dissipation is not reflected by the model results. Therefore, this figure rsupports my major concerns about the model: Too large vertical level spacing combined with possible shortcomings in subgrid-scale

parameterization leads to insufficient dissipation.

L858-868: Ditto.

Fig. 16a: This figure suggests that you would get a reversed power spectrum of the wave amplitude with respect to the horizontal wavenumber, i.e., increasing (instead of decreasing) power with increasing wavenumber? Please check. If this is so, this would imply that the model results at these small scales are not reliable at all.

L927: Ditto

L978-981: As mentioned above, it is not just horizontal grid-spacing (and model numerics, as you mention in L992) that determines how well a model simulates GWs. You have to consider the vertical grid spacing as well. Most importantly, inviscid fluid dynamics cannot handle GW breakdown and wave-mean flow interaction. You need an explicit dissipative process for non-transient wave-mean flow interaction (see the non-acceleration theorem, Lindzen's GW saturation theory, or the classical McFarlane paper about orographic GW parmameterization). That is why the parameterization of subgrid-scale processes (turbulent diffusion) is very important in any GW-resolving circulation model (e.g., Becker and Vadas, 2018).

Minor Comments:

L73-75: I agree with this statement. However, the authors miss the opportunity to put the orographic GW momentum flux from South Georgia into the context of the general circulation in SH winter.

L92: Please point out that the model used in this study is a real-date regional model that is forced by a global forecast model via lateral boundary conditions. Therefore, this regional model is not "essentially free running".

L135: Please be specific whether the vertical resolution relates to wavelength or grid-spacing.

L179: The vertical resolution of the global model is presumably too coarse to represent inertia GWs in the stratosphere. This could be the reason why the regional model misses these waves when compared to the AIRS data.

L205-214: This paragraph is hard to follow and distracts a bit from the very good writing otherwise in the paper.

Figure 2: Please plot the zonal wind with the same color coding as the meridional wind (blue for minus, red for plus)? Can you use a nonlinear color scale to make the accelerations and deccelerations of the tropospheric wind visible? Note that the wind in the lower troposphere determines the forcing of orographic GWs.

L245: The radiosonde observations do not provide a horizontal average over the domain covered by the model. Please reformulate correspondingly.

L266: Figure 3 is very well composed. However, Fig. 3g illustrates that the simulated winds are not in good agreement with the radiosonde data. Rather, the agreement is only reasonable. The mean meridional wind in Fig. 2d is predominantly southward from 30 to 60 km and is of the order of a few -10 m/s. The corresponding wind in Fig. 3d shows a bias of about 10 m/s.

L279: Short-timescale variability would average out when comparing time-averaged wind profiles. I suggest to accept these discrepancies and to discuss the possible implications for orographic forcing and vertical propagation of GWs in the model.

L284-L290: See my comment with respect to L226 above.

L300-306: The differences between model and radiosonde data are not minor. Invoking the "climatological level" of simulated wind in case studies of orographic GWs, which are subject to extreme intermittency, does not sound conclusive.

L360-363: These sentences are hard to understand (e.g., "vertical resolution for that vertical layer"). Please reformulate.

L377: This statement is not conclusive. What about model errors?

L470-471: A "reasonable apparent similarity" is not observed when considering the dissimilarity of individual phase lines between the two data sets in Fig. 6a and e.

page 25: Why is this new section called "Results". The previous Section 3 contained plenty of results, not just methodology.

L535-538: This description of secondary GW generation from MW breaking does not seem consistent with the aforementioned papers by Vadas and coauthors.

L539-542: This sounds very vague. I recommend to simply discard speculations of this kind. Furthermore, if you want to discuss secondary GWs in your model, then you need to consider how the model simulates dynamical instability and dissipation of resolved GWs and, hence, the necessary body forces for secondary GW generation. As discussed earlier, the very coarse vertical resolution of the model combined with the lack of knowledge about the built-in (presumably implicitly numerical) dissipation casts doubts on whether the model reliably simulates body forces from GW dissipation in the stratosphere.

L553: Note that this equation holds strictly only for a monochromatic GW or, at best, for a narrow spectrum of GWs. As soon as you have a broad spectrum, the wavelengths to be used at the rhs become arbitrary. More importantly: I am missing the Reynolds-type average of $(T')^{**}2$ (see my comment on Fig. 7 above). Please clarify.

L582-585: This information clarifies my previous comment at least for Fig. 8-10. Given the size of the island relative to the model domain and the GW scales in AIRS and model-as-AIRS, you use the area-average to compute the MF. I think that is the right choice here. How would the resulting MF contribute to the zonal mean parameterized in global models?

L588: I can not see the red markers in Fig. 8-14.

L681: Again, I disagree that "observed and simulated wave fields are quite similar". As

mentioned earlier, there are even qualitative differences.

L700: What about spontaneous emission from the upper tropospheric jet stream? See Plougonven and Zhang, 2014, Rev. Geophys: Internal gravity waves from atmospheric jets and fronts.

L827-829: These differences could simply result from errors in the background wind (driven by the global model) in the lower troposphere, leading to errors in orographic forcing of MWs in the model. I believe the authors should discuss this role of the tropospheric winds somewhere in the paper.

L845-849: See my previous major and minor comments regarding the obvious and possible shortcomings of the model.

L928-935: It is hard to follow these arguments. Of course, MWs can be forced by non-stationary background winds. Furthermore hourly fluctuations of the background wind would correspond to non-orographic GWs that you force at the lateral boundaries. Your discussion of possible reasons for the model shortcomings (see also L936-940) do not mention the concerns that I raised above.

L946-947: This sentence seems not logical. Consider reformulation.

L949-950: "not so commonly"? Which observations are you aware of that show this feature of very large MW amplitudes in the stratosphere at very small horizontal scales?

L954-955: Why should intermittency of MW forcing give rise to shorter horizontal wave-lengths than stationary forcing? Usually, the structure of the topography determines the spectrum that can be forced.

L963-964: Now you argue that an "overly-stable wind vector" could give rise to the high power of MWs at very small scales in models.

L993-997: I think that here you reveal a misconception about semi-implicit time stepping in circulation models. Semi-implicit time stepping is applied to suppress the artificial generation of very fast anelastic waves and sound waves; otherwise, smaller time steps would be required for numerical stability. In any event, the time step is always small enough to properly resolve the time scales of anelastic GWs that are well described by the representation of the model equations in gridspace.

L999-1000: Here you finally come up with a critical comment about the lack of dissipation in the model stratosphere.

L1002: You did not run the model at very high spatial resolution. Your vertical resolution in the stratosphere is much coarser than even in GW-resolving global models run at moderate horizontal resolution (e.g., Sato et al. 2012., JAS: Gravity Wave Characteristics in the Southern Hemisphere Revealed by a High-Resolution Middle-Atmosphere General Circulation Model). Again, your coarse vertical level spacing is certainly not adequate to support your very high horizontal resolution.

L1005: As long as we do not solve the (viscid) Navier-Stokes equations with a resolution of 1 cm in the troposphere, the performance of our circulation models will always depend on how unresolved (subgrid-scale) dynamical processes are parameterized.

L1022-1023: Yes! See my comments above.

L1030: You did not perform sensitivity experiments using the same model with different horizontal resolutions.

Typos/suggestions:

L28: play a key role

L35: recognized that GCMs

L46: match measurements

Caption to Fig. 1: and descending (c) overpass

L227: stronger than what?

L228: exceeding

L244: a meridional wind reversal

L269: cancel "In both directions"

L301: in near surface winds

L359: AIRS measurements regarding the horizontal wavelength, that is

L420: observations and for the model and model-as-AIRS, respectively.

L464: applied to the

L497: whose intrinsic horizontal

L506: upward propagating

L526: discard "striking"

L547: and constrains

L548-549: momentum flux from mountain wave sources at isolated small islands is an important area of current research

---

## Author Comment (AC1) · 21 Dec 2020

**Authors' response to comments from Reviewer #1 on "Stratospheric gravity waves over the mountainous island of South Georgia: testing a high-resolution dynamical model with 3-D satellite observations and radiosondes"**

N. P. Hindley et al.

**General Comment for all Reviewers**

We would like to thank the reviewers for the hard work in preparing their reviews of our submission. Their helpful suggestions have significantly improved the study. Several main improvements are listed below:

- In response to the reviewers' comments, we have significantly improved the way the model is sampled to create the model-as-AIRS dataset in our study. We realised that it is not enough to simply apply the AIRS horizontal resolution to the model: the AIRS horizontal sampling must be considered too. By sampling the model on the AIRS horizontal grid and taking into account the different sampling locations of each overpass, we are able to remove the background temperatures in exactly the same way for the AIRS and model-as-AIRS temperatures (we no longer use the nSG model runs for this). This ensures that our analysis steps allow for the spectral range of GWs visible to the AIRS and model-as-AIRS to be consistent.

- We also apply specified AIRS retrieval noise to the model-as-AIRS, which is characterised from a realistic AIRS granule. By applying the noise to the model-as-AIRS, we can separate out the effects of retrieval noise. This is important for the area-averaged results upwind and downwind of South Georgia.

- We now keep GW results measured in the full-resolution model very separate from the comparison between the AIRS and the model-as-AIRS. GW momentum flux in the full-resolution model is now calculated using wind perturbations, rather than from down-sampled temperature perturbations as before, and no comparison is made between GWMF in the model and the model-as-AIRS. This an important distinction because it is not possible to apply consistent horizontal sampling and background removal methods to both datasets, so no fair comparison can be made.

- The above steps have greatly improved the agreement between the AIRS GW measurements and the model-as-AIRS. As a result, the paper has been substantially reduced in size from 16 figures to 11 with a $\sim$20% reduction in text. Inconclusive or superfluous results and discussions have been removed, and a new Fig. 11 showing a case study of a short-$\lambda_H$ GW event has been added.

**Response to Reviewer #1: Specific Comments**

- *General Comments: Some figures have been placed uncomfortably far from their first citation*

  Thanks, currently the figures are placed automatically, but we will address this in the pre-print stage once we have uploaded our latex and figure files to ACP.

- *Introduction: As stated above, I believe that some significant work in relation to GWs studied with satellite data and numerical modelling in the Southern GWs hotspot should be included in the context of this section.*

This study is not specifically focused on the southern GW hot spot, but we agree it can be a motivation. We have included an additional list of relevant GW studies. Such studies are useful for context so we are happy to include them. It is difficult to know which studies the reviewer would like us to include since they did not specify any further, but we hope that we have included the studies they had in mind. They are welcome to contact us about any relevant studies that we may not be aware of yet.

- *l.134-135 and l.147 There is a need for clarification about some definitions. If vertical resolution is at best 7 km (uncorrelated or independent successive data), then the 3 km sampling is just some kind of interpolation and no wavelengths shorter than 14 km may be detected, which would have a disastrous effect on the following results. Also l.353-355.*

  The AIRS vertical resolution is determined by the vertical weighting functions of the spectral channels used in the retrieval (at best $\sim$7 km). The 3-D AIRS retrieval of Hoffmann and Alexander (2009) is calculated in 3 km vertical steps. Apologies for the confusion. We have updated the text for clarity.

  Because the vertical resolution due to the kernel functions of the selected AIRS channels (7-14 km) is always larger than twice the vertical spacing of the vertical grid (Nyquist resolution, $2\times3=6$ km), the vertical resolution of the former dominates our sensitivity to GWs in the vertical. We have updated this description is the revised paper.

- *l.138 If temperature uncertainty may be up to 1.5 K you need to justify how you rely on results of 1K amplitude or even less.*

  The AIRS retrieval noise is assumed to be uncorrelated pixel-to-pixel variations. If there are coherent wave cycles of a 1 K amplitude wave over many pixels, then the 1 K will still be measurable beneath the 1.5 K noise (like a noisy sine wave). But the reviewer raises a good point. It is not perfect, but in practice we find that retrieval noise is often better than estimated in Hoffmann and Alexander (2009), and such waves are measurable using our 3DST method Hindley et al. (2019).

- *l.143 Please mention the artefacts that may remain.*

  We recently found that, in some situations, minor artefacts can arise in the use of the 4th-order polynomial method (which is the main method used in the community for extracting GWs from AIRS observations). These artefacts are very small, with amplitudes less than 0.1 K, manifesting in artificial phase fronts in the along-track direction. But these are only visible if the method is applied to data that contains no clear GW features or retrieval noise. We found no evidence of such artefacts in the AIRS or model data used in our study here, so further discussion is not relevant for this paper.

- *Section 2.2: More details of the simulation characteristics are needed. You should mention the type of sponge and its intensity and the timestep that was used. Did numerical instabilities arise during the initial steps? If so, how did you handle them? Did you assess the model spin up? Did simulations exhibit alterations with slightly earlier or later initial time? In addition, your operational analyses have a 46 km resolution, whereas your simulation domain has a 1.5 km grid. Have you evaluated the possible effect of this factor of 30? Wouldn't it be advisable to use a smaller ratio? May this fact be responsible for the model not being able to adequately represent the non-orographic GWs (l.403-406, l.972-974)? How reliable are simulations if such a large structure is not "transmitted" from the forecast to the local area model?*

  The description of the model set up has been revised in the resubmission. Details regarding the model set up, including of sensitivity tests that justify the model grid configurations can be found in Vosper (2015) and Jackson et al. (2018).

  The sponge layer damping increases exponentially with height above 58.5km to the upper boundary. No indications significant spurious wave reflection from the sponge layer was found. No model instabilities were encountered. The

timestep used was 30s. The results were not sensitive to this choice and this is expected because of the second-order accuracy of the Met Office Unified Model ENDGame time integration scheme.

Model spin up: the integrations presented are long simulations (1-31 July etc), in which the local-area domain is driven by lateral boundary conditions from a sequence of 24 hourly global forecasts, which are linearly interpolated onto timestep of the local-area domain. Spin-up issues in the high resolution domain are therefore eliminated after only a few hours of the experiment due to realistic boundary conditions. The boundary conditions are re-initialised every 24 hours. We did not encounter any discrepancies in the wind or GW field over the island that would relate to initialisation issues.

The reviewer is correct here, we do not expect any GWs in the global forecast to be realistically "transmitted" into the local-area model due to the global to regional grid spacing ratio. This is now discussed in the revised paper.

We did not observe numerical artefacts near the boundaries. This nested high-resolution setup is a common configuration of the Unified Model for providing local forecasts, and the model is specifically designed to be realistic in this case.

- *L.236-237 You should also compare the numerical model with radiosonde temperature (not only wind validation) as you have it at disposal, but you should not use it for GWs as you clearly stated in l.254-256.*

We did compare this, but we did not include it because it was not especially relevant to our comparison of stratospheric GWs.

The results of our comparison of model and radiosonde temperatures, performed in the same way as in the paper, is shown in Fig. R1 here. We find that, on average, the model and radiosonde temperatures agree quite well, although there appears to be a systematic positive bias of around 2 K in the radiosonde measurements at most altitudes. This could indicate a cold bias in the model, or it could indicate a systematic temperature bias in the radiosonde instrumentation itself. More investigation is needed to explore these issues, which is not the focus of the present study. For this reason, we did not include the temperature results in the paper since the wind comparison is more useful for GWs. The wind results are derived from the GPS position of the balloon, so systematic biases such as this are not likely to occur.

The radiosondes are planned to be used in a future study comparing various model/radiosonde parameters, such as comparing model vertical velocity to radiosonde ascent rate. Any systematic temperature biases may be discussed then.

- *l.298 There may be significant positional errors? How large can they be?*

In l.298 we said that the errors were not significant. We are referring here to geolocation errors in the radiosonde position during each balloon flight. However, since these positional errors are expected to be much smaller than the model grid (metres rather than kilometres), they are not expected to be significant. The sentence has been removed in the revised paper.

- *l.449-453 The redundancy of the method should be shortly discussed or cited as it is strongly related to the reliability of the calculated GWs amplitudes. This is especially important in the context of some notable amplitude discrepancies below among AIRS and the model.*

It is not completely clear what the reviewer means (in this context) by the redundancy of the $S$-transform method, but we suspect that they are referring to its robustness in response to unreliable, anomalous, featureless or noisy data, or systematic differences in how different datasets are analysed.

The 3DST method we apply here is very thoroughly tested and validated in Hindley et al. (2019). It is based upon the 2DST method of Hindley et al. (2016), who also tested it thoroughly. Synthetic wave fields with simulated noise,

in addition to AIRS measurements, are analysed in both studies for a complete evaluation. We are as confident as we can be that what comes out of the analysis is a fair representation of what went in, whatever the dataset.

Regarding the measured amplitude differences (which are smaller in the revised paper due to better sampling), the exact same 3DST analysis method is applied to AIRS and the model-as-AIRS. The software package we use, and the method that it follows, are entirely independent of what kind of data is being analysed. Thus, we are as confident as we can be that any amplitude discrepancies that may arise are inherent to the different datasets, not to the analysis method. We have updated the text to make this clearer.

- *l.490-491 You should check your hourly output for stationary phases and increasing wind speed with height.*

  The increasing model wind speed with height can be seen in the new Fig. 2. We have inspected the hourly output of the model temperature perturbations during both modelling periods as video animations. We find that the phase fronts forming over the island during this time are stationary with respect to the ground, highly indicative of mountain wave activity. We have updated the text to make this clearer.

- *l.643-645 You should use your model simulations to test this argument.*

  This sentence was not clear, so it has been removed. This section has also been significantly revised in the resubmission. It is expected that the full-resolution model simulates significant GWMF that is below the resolution and sampling limits of AIRS due to the fine horizontal grid of the simulation. The new Fig. 11 highlights this. The full-resolution model results are also not compared to the AIRS observations in the revised paper, because we realised that a fair comparison is not possible.

- *l.650-651 The model simulations should give you a clue for upward or downward phase propagation.*

  Because the same procedure must be applied to AIRS and the model-as-AIRS, we cannot use supplementary model information to constrain the upwards/downwards waves in the model-as-AIRS, because we cannot do the same for the AIRS measurements. Therefore, it would not be a fair comparison.

  There is no way to independently break the upwards/downwards ambiguity from the AIRS temperature measurements alone, so to make a fair comparison we must use the same approach for both the AIRS and model-as-AIRS. We accept any small directional errors in order to ensure consistency.

- *l.678 Does this imply that AIRS amplitudes are typically the double of the model? If so, explain.*

  Please see the new results in the revised paper. Area-averaged GW amplitudes in AIRS are typically larger than the model-as-AIRS, likely due to the presence of NGWs away from the island. As mentioned above, NGWs away from the island are unlikely to be well-simulated in the local-area model. This is discussed in the revised paper.

- *l.701-702 The presence of the jet, the polar vortex, storms and fronts can all be probably checked from your operational analyses in order to verify the support to your argument.*

  Indeed, this is a good suggestion. We have included these possibilities in the revised paper, but Fig. 11 has been removed for brevity because it did not provide a particularly quantitative comparison.

  The discussion of these issues has been revised in the resubmission. As mentioned above, it is not expected that NGWs would be realistically "transmitted" through the boundary conditions of the local-area model.

- *l.706-714 Please use your hourly simulations to verify at least partially in the mentioned geographical domain your detachment or moving secondary waves argument.*

  This argument has been removed from the revised paper, but it is a legitimate possibility.

  The detachment or 3-D advection theory for mountain waves is described and discussed in the modelling study by Sato et al. (2012). Further, Ehard et al. (2017) used forward ray tracing analysis of mountain waves in lidar data

over New Zealand to show that it was possible for such waves to propagate several 1000s of km horizontally in the stratosphere due to the strong meridional shear of the zonal wind. They found examples where the original mountain wave structure over the mountains had dissipated but the advected part of the wave structure still continued to propagate far away from its source. Their results were supported by reanalysis and AIRS observations.

Figure R2 shows four snapshots from an animation of gravity wave temperature perturbations at 30 km altitude in ERA5 reanalysis during July-August 2012 over the southern hemisphere. This is an animation we prepared previously a conference presentation. Gravity wave temperature perturbations are extracted from the background temperature via a zonal planetary wave fit.

Here we can see that mountain wave structures that form over the Andes are found to extend far out over the Southern Ocean to the east. In several cases, we found that these perturbations extended all the way to South Georgia. The size and orientation of these perturbations is consistent with some of the apparent non-orographic wave activity seen in AIRS observations in our study over the ocean around the island.

We appreciate the reviewer's point. These kinds of events not typically expected under classical mountain wave theory in the 2-D case, because the meridional gradient of the zonal wind speed is not considered. Further investigation into this aspect is expected in future studies, but it is not the focus of the present study.

- *Table 1: To check if differences are significant it is necessary to include uncertainties with the averages.*

  The reviewer has a good point, but uncertainties are not straightforward to calculate for these measurements. Further, we are confident that the sources of measurement error are the same for each data set. This is especially so in the revised paper, where specified AIRS retrieval noise is also applied to the model-as-AIRS for consistency.

  Regarding the values in Table 1, standard error values could be included but for these average values they are not helpful for the following two reasons:

  1) The average wave amplitudes and GWMF are all area averages over a large number of time steps. The estimated standard error of the mean of a distribution is given by $\sigma_{\bar{x}} = \frac{s}{\sqrt{n}}$, where $s$ is the sample standard deviation and $n$ is the number of sample points in the distribution. Our measurement grid between 25km and 45km altitude is 90x120x14 points. There are 87 AIRS overpasses and over 1300 model timesteps. For AIRS and the model-as-AIRS this yields $n > 10000$ sample points that go into the Table 1. This yields a standard error of order $\sim 10^{-3}$ for these average values, which is not helpful.

  2) The same is applied to AIRS and the model-as-AIRS. As a result, any systematic errors will propagate through the analysis in the same way for each data set. Thus, any relative differences in the area-averaged and time-averaged values shown in Table 1 are related to real differences between the data sets over the two regions.

  Errors in the measurement of individual wave amplitudes and wavelengths, from which these values are derived, are harder to define. For this, we would point towards the testing and validation of the 3DST measurement technique in Hindley et al. (2019). However, any systematic errors these measurements would be identical for each dataset (or random errors would average out in the large sample size). Therefore, any mean differences in our comparison are likely to be due to genuine differences in the datasets.

- *Figure 14d,h: Please discuss the parts where the difference is larger than the absolute fluxes.*

  We have gone back and checked our data, and there are no regions in Figs. 14(d,h) where the difference between the AIRS and model-as-AIRS shown (AIRS minus model-as-AIRS) is larger than the absolute AIRS fluxes. Note that these are differences in the absolute flux between the AIRS and model-as-AIRS data and do not imply a direction, and that panels (a,e) and (d,h) use different colour scales.

  In any case, Fig. 14 has been revised and simplified in the resubmission (new Fig. 9).

- *Figure 16: How can you define a unique amplitude if it can change by a factor over 4 from 25 to 55 km?*

  We do not define a unique amplitude, we simply show the average of all measurements between the two altitudes. Indeed, we are aware that the exponential increase of wave amplitudes with altitude will bias the average value to be more representative of values at higher altitudes, but the vertical resolution of the AIRS and the model-as-AIRS is actually significantly better at lower altitudes below around 40 km (see Fig. 5 of Hoffmann and Alexander (2009) or Fig. 2 of Hindley et al. (2019)), so there are a range of aspects involved here.

  In the revised paper, we now only consider averages between 25 and 45km to address this issue.

- *l.896-897, 913-916 and 919-920 This could indicate that AIRS may be omitting an essential contribution to GWs momentum flux and its later parametrization in global models. This fact merits a quantification of the above effect due to the possible discrepancies of simulations or observations in this work with the real atmosphere.*

  As mentioned above, Fig. 16 has been removed in the revised paper due to an error in the bin-width normalisation

  Of course, the observational filter of AIRS means that it can only observe GWs within a spectral range. We do not claim that AIRS can observe all the GWs required to constrain GW parameterisations in global models, but to our knowledge these 3-D observations are the only global dataset that that can independently measure directional GWMF, so they do have value.

  As the reviewer suspects, the sampling pattern of AIRS means that GWs at short-$\lambda_H$ directly over the island are not observed in AIRS unless the viewing geometry is favourable (see new Fig. 11 in the revised paper). This is discussed in the revised paper.

  A quantification of the GWMF that is not visible to AIRS can be found in Table 1, where GWMF in the full-resolution model is also included. It is shown in the revised paper however that when the AIRS sampling and resolution is correctly applied to the model, the agreement in GWMF between the observations and the model-as-AIRS is reasonably good.

- *l.936-940 Can you give a reference where this effect has been quantified? How likely is it that this high frequency wind variability exists in that zone? Can you draw conclusions from the individual radiosonde profiles?*

  This is a good suggestion, but this argument is not discussed so much in the revised manuscript due to the better agreement between AIRS and the model-as-AIRS, after the improved sampling and resolution methods are applied.

  For the sake of discussion, our radiosonde comparison in Fig. 3 supports this argument somewhat. The model winds evaluated along the radiosonde flight path exhibit less small-scale variability than the radiosonde measurements themselves. But this is no longer a significant consideration in the revised paper.

- *l.951-957 Again, another possibility is that AIRS is missing these GWs.*

  Yes, see our point above and the new Fig. 11 in the revised paper. Some of the characteristic mountain waves over South Georgia occur at scales around $\lambda_H \sim$30–40km, which are only visible to AIRS when the viewing geometry is favourable. This means that such waves may be underestimated in recent global climatologies such as Hindley et al. (2020). This is a key point in the revised paper.

- *l.993-1000 What was the expectation for GWs amplitudes in your simulations according to the timestep you have chosen? Was it in agreement with your results?*

  The timestep used was 30 seconds, and the GW amplitude results were not sensitive to this choice, provided that it was a sensible value (e.g. of order 30 seconds to several minutes etc). This was expected because of the second-order accuracy of the Unified Model ENDGame time integration scheme.

- *l.1024-1027 Please check if further analysis in the previous sections produces any modifications.*

  As discussed above, the paper has been substantially improved by the reviewers suggestions. The analysis has been re-formulated and the revised paper has been significantly improved.

**Minor Technical Corrections and Comments**

All corrections made, sentences rephrased/deleted and references added, thank you.

**References**

B. Ehard, B. Kaifler, A. Dörnbrack, P. Preusse, S. D. Eckermann, M. Bramberger, S. Gisinger, N. Kaifler, B. Liley, J. Wagner, and M. Rapp. Horizontal propagation of large-amplitude mountain waves into the polar night jet. *Journal of Geophysical Research: Atmospheres*, 122(3):1423–1436, 2017. doi: 10.1002/2016JD025621.

N. P. Hindley, C. J. Wright, N. D. Smith, L. Hoffmann, L. A. Holt, M. J. Alexander, T. Moffat-Griffin, and N. J. Mitchell. Gravity waves in the winter stratosphere over the southern ocean: high-resolution satellite observations and 3-d spectral analysis. *Atmospheric Chemistry and Physics*, 19(24):15377–15414, 2019. doi: 10.5194/acp-19-15377-2019.

N. P. Hindley, C. J. Wright, L. Hoffmann, T. Moffat-Griffin, and N. J. Mitchell. An 18-year climatology of directional stratospheric gravity wave momentum flux from 3-d satellite observations. *Geophysical Research Letters*, 47(22), November 2020. doi: 10.1029/2020gl089557.

L. Hoffmann and M. J. Alexander. Retrieval of stratospheric temperatures from Atmospheric Infrared Sounder radiance measurements for gravity wave studies. *J. Geophys. Res.*, 114:D07105, 2009. doi: 10.1029/2008JD011241.

D. R. Jackson, A. Gadian, N. P. Hindley, L. Hoffmann, J. Hughes, J. King, T. Moffat-Griffin, A. C. Moss, A. N. Ross, S. B. Vosper, C. J. Wright, and N. J. Mitchell. The south georgia wave experiment: A means for improved analysis of gravity waves and low-level wind impacts generated from mountainous islands. *Bulletin of the American Meteorological Society*, 99(5):1027–1040, 2018. doi: 10.1175/BAMS-D-16-0151.1.

K. Sato, S. Tateno, S. Watanabe, and Y. Kawatani. Gravity Wave Characteristics in the Southern Hemisphere Revealed by a High-Resolution Middle-Atmosphere General Circulation Model. *J. Atmos. Sci.*, 69:1378–1396, 2012. doi: 10.1175/JAS-D-11-0101.1.

S. B. Vosper. Mountain waves and wakes generated by south georgia: implications for drag parametrization. *QJRMS*, 141 (692):2813–2827, 2015. doi: 10.1002/qj.2566.

[Figure]

Figure R1: Temperature measurements in (a) the local-area model and (b) coincident radiosonde observations during June-July 2015. Temperature in the model are evaluated along the radiosonde flight path in 3 spatial dimensions and time. Panel (c) shows the difference (Sondes − Model) between them.

[Figure]

Figure R2: Selected snapshots from an animation of ERA5 reanalysis temperature perturbations over the southern hemisphere at 30 km altitude during July-August 2012.

---

## Author Comment (AC2) · 21 Dec 2020

**Authors' response to comments from Reviewer #2 on "Stratospheric gravity waves over the mountainous island of South Georgia: testing a high-resolution dynamical model with 3-D satellite observations and radiosondes"**
N. P. Hindley et al.

**General Comment for all Reviewers**

We would like to thank the reviewers for the hard work in preparing their reviews of our submission. Their helpful suggestions have significantly improved the study. Several main improvements are listed below:

- In response to the reviewers' comments, we have significantly improved the way the model is sampled to create the model-as-AIRS dataset in our study. We realised that it is not enough to simply apply the AIRS horizontal resolution to the model: the AIRS horizontal sampling must be considered too. By sampling the model on the AIRS horizontal grid and taking into account the different sampling locations of each overpass, we are able to remove the background temperatures in exactly the same way for the AIRS and model-as-AIRS temperatures (we no longer use the nSG model runs for this). This ensures that our analysis steps allow for the spectral range of GWs visible to the AIRS and model-as-AIRS to be consistent.

- We also apply specified AIRS retrieval noise to the model-as-AIRS, which is characterised from a realistic AIRS granule. By applying the noise to the model-as-AIRS, we can separate out the effects of retrieval noise. This is important for the area-averaged results upwind and downwind of South Georgia.

- We now keep GW results measured in the full-resolution model very separate from the comparison between the AIRS and the model-as-AIRS. GW momentum flux in the full-resolution model is now calculated using wind perturbations, rather than from down-sampled temperature perturbations as before, and no comparison is made between GWMF in the model and the model-as-AIRS. This an important distinction because it is not possible to apply consistent horizontal sampling and background removal methods to both datasets, so no fair comparison can be made.

- The above steps have greatly improved the agreement between the AIRS GW measurements and the model-as-AIRS. As a result, the paper has been substantially reduced in size from 16 figures to 11 with a $\sim$20% reduction in text. Inconclusive or superfluous results and discussions have been removed, and a new Fig. 11 showing a case study of a short-$\lambda_H$ GW event has been added.

**Response to Reviewer #2: Major Remarks**

1. *Why don't you use a similar 4th-order polynomial fit as for the AIRS data to determine temperature perturbations that contain orographic and non-orographic GWs? Using SG and smoothed nSG to determine perturbations seems nice way to get the contribution of the orographic GWs in the model but doesn't immediately sound like the best choice for a comparison AIRS (unless you can show that 4th order polynomial leads to similar perturbations as the procedure described here and/or your results are not very sensitive to the background removal) [Moreover, I realized that contributions of non-orographic waves seem to be important in several sub-sections later in the manuscript. SG and nSG could be used to separate and quantify some of the non-orographic contributions in the model simulations*

*(not all the analyses need to be done but for some quantities it could strengthen the findings and conclusions with respect to the non-orographic GWs).]*

We'd like to thank the reviewer for their helpful and perceptive comment. They are correct that the different background removal methods needed to be revised. As mentioned above, all model-as-AIRS data has now been re-processed in an improved way that takes into account the horizontal sampling of the AIRS measurements. Because we now sample the model-as-AIRS directly onto the grid of the closest AIRS overpass, the same 4th-order polynomial fit can be applied to extract GWs from both datasets. We also apply specified AIRS retrieval noise to the model-as-AIRS for comparability. This means that the background removal method is now thoroughly consistent between the two datasets. As a result, we find a significant improvement in the agreement between AIRS and the model-as-AIRS (see revised paper). The nSG model runs are no longer used. We'd like to thank the reviewer for prompting us to consider this issue. The study is now substantially improved.

Regarding the separation of OGWs from NGWs in the model, because we cannot do the same thing for AIRS measurements, we cannot make a fair and quantitative comparison of OGWs/NGWs in the model to GWs observed in AIRS. Therefore we are not sure how useful such a separation of waves in the model would be. In any case, upon further consideration, transitory NGWs from outside the local-area domain are not expected to be well-simulated around South Georgia. This is due to the coarse resolution of the global simulation and it is unclear how realistically GWs are "transmitted" into the local-area domain. Our current method of "upwind" and "downwind" boxes either side of the island provides a crude but consistent metric for assessing the relative quantities of OGWs and NGWs in the two datasets.

As mentioned above, GW results in full-resolution model are now well-separated from the comparison of GWs in AIRS and the model-as-AIRS. This is because we cannot extract GWs from the full-resolution model in the same way as the AIRS and model-as-AIRS using the 4th-order polynomial across track fit. Instead, a polynomial fit in the zonal direction is used to extract GWs in the full-resolution model for reasonable consistency, but we stress in the revised paper that the two approaches are not directly consistent. Also, the model GWMF is now calculated using wind perturbations. This further separates these results from the AIRS and the model-as-AIRS, which are now the focus of our study.

2. *Structure of the paper:*

   - *strictly separate "data and methods" and "results"*
     *This would also mean help the reader to already know by the end of Sec. 2 what to expect in the result section of the paper.*
     *- First part of Sec 3 and 3.1 describing the data processing is better moved to Sec 2 (which could then be called Data and Methods)*
     *- First part of Sec 3.3 Should be moved to Sec 2*
     *- First part of 3.3.2 should be moved to Sec 2.*
     *- Sec 4.3: Gini coefficient can be introduced in Sec. 2*
     *- First part of Sec 4.4. should be moved to Sec 2.*
     *- Sec. 2.4 already presents results and could be moved to Sec. 3 (or create new Sec 3 with only content of Sec. 2.4)*

   - *separate results and discussion*
     *- L397: This sentence can be left for later discussion.*
     *- L403-407: This sentence can be left for later discussion ...and so on*

Thank you for this suggestion. The revised paper has be significantly reformulated, and there is a much better split between data/methods/results/discussion. This has significantly improved the readability and removed repeated discussions.

3. *Subjective and expletive words like "overwhelmingly" or "very" (>20 occurrences) can be reduced without losing information. There is also a large amount of speculation (some contradictions, some repetitions) in the paper (>25 occurrences of likely and >30 could explanations) that lack quantification. Just some examples: "...should result in simulated conditions over South Georgia that are very close to reality for the given time periods." "...time separation is very small and the local wind conditions can be expected to be very close to reality." "A small fraction of this distribution is likely to be measurement error, but the results may still be significant.", "Since these are clear mountain wave structures, it suggests that this could be due to errors in the speed and direction of the background wind in the model.") I recommend looking through the paper and deciding if such expressions/sentences are essential for the main content/message of the paper and if they can be justified or quantified. If not, they could be removed. Instead of listing every possible explanation for some of the observed differences between model and observations, the explanations could be limited to the one or two most relevant ones.*

Thank you for this comment. The reviewer is absolutely correct that the original submission contained a lot of unnecessary and subjective discussions. The revised paper is more concise, with a ∼20% reduction in text. One reason for this reduction is that the revised methods mentioned above have greatly improved the consistency between AIRS and the model-as-AIRS results.

**Minor Comments**

- *L9: "high" instead of "very high"; you may want to add "without gravity wave parametrization" over South Georgia.*

  Added, thanks.

- *L23: please specify which scales are meant by short and long*

  Added, thanks.

- *L40: not all but "a large amount of these short vertical and horizontal scales are too small to be resolved even in recent GCMs"; pls add more recent citation (e.g., Plougonven et al 2019, How does knowledge of atmospheric gravity waves guide their parametrizations?)*

  Added, thanks. The Plougonven et al. (2020) paper is an excellent inclusion.

- *L43: In some cases? Isn't it rather the norm than the exception?*

  Agreed, sentence revised.

- *L84: GWs can propagate large horizontal distances, and from this point of view the Andes are not too far at all. (compare L705: The island lies only 2000 km east of the southern tip of South America, a region associated with the largest stratospheric mountain wave activity observed anywhere in the world)*

  Agreed, the sentence has been revised. In our response to Reviewer #1 we briefly discussed the study of Ehard et al. (2017), who showed that some mountain waves at these latitudes during winter could propagate several 1000s of km from their sources due to meridional gradients in the zonal wind.

  For discussion, we suspect that some of the large NGWs found in AIRS measurements near to South Georgia may have originated over the southern Andes and Antarctic Peninsula and could have propagated downwind via this mechanism. But we do not expect such waves to be well-simulated in the local-area model due to the coarse resolution of the

global forecast that provides the lateral boundary conditions (and the time interpolation applied between these hourly forecasts on to the local-area model timestep).

Discussion of these factors is minimised in the revised manuscript because more further observations and investigation are required (which may be the focus of a future study, but is certainly beyond the focus of the present study).

- *L88: range of scale sizes? Please clarify.*

  This phrasing is confusing and has been removed. We meant that because of the different measurements involved (i.e. the radiosondes, satellites, models) a range of GW scales have been studied.

- *Fig. 1: Why are the soundings of January 2015 shown here? They are not relevant for the content manuscript. (see also comment on L190)*

  They are included to put the wintertime radiosonde measurements in context. They highlight the strong wintertime winds in the troposphere and lower stratosphere. We think they are useful for context for readers new to the field. We want to mention that the data set exists in case these measurements are useful for future researchers.

- *L137: "…to study gravity waves." Not the whole spectrum of gravity waves is small scale.*

  Agreed, the sentence has been revised.

- *L140: Is the fit applied horizontally or vertically?*

  The fit is performed horizontally in the across-track direction. We have revised the sentence to make this clearer.

- *L176: …much finer than the 3 km vertical grid of the AIRS retrieval: this is kind of a change in the objective of the paper. "when a model is allowed to run at very high spatial resolution over South Georgia, how realistic are the simulated gravity waves compared to observations?" vs how realistic are simulated gravity waves in the observational window of AIRS? Moreover, can the vertical grid spacing of the model be directly compared to the vertical grid spacing of the retrieval? At least in the horizontal effective resolution is more like 5-10 times the grid spacing.*

  We agree. The discussion of this aspect has been significantly revised in the resubmission. In any model-observation comparison paper we can only compare gravity waves within the observational window of the measurements used, but this was not clear in the original abstract.

  We also agree about that we cannot infer model resolutions from model grid spacings alone. The text has been revised to reflect this. Vertical transport however is typically better represented in the Unified Model than in the horizontal, so we expect gravity wave perturbations to be better represented over a few vertical layers than the same number of horizontal grid cells. Even so, using the range of 5–10 times the grid spacing, this would lead to a model vertical resolution of 3–5.5 km at 20 km altitude and 6.5–15 km at 45 km altitude (25–45 km is the altitude range considered in the revised paper). These values are quite comparable to AIRS average vertical resolutions at these altitudes, so our comparison is still valid.

  Sensitivity tests for this model configuration with vertical grids with 70, 118 and 173 levels were performed by Vosper (2015). They found no significant differences in the resolved GWMF over South Georgia between 118 and 173 level simulations, suggesting that the vertical grid spacing used here is sufficient to resolve the dominant components of the mountain wave field.

- *L183: Does "no gravity wave parametrization" also mean no non-orographic parametrization?*

  Yes, we have revised the text.

- *L190: I would expect that there wasn't much mountain wave activity at all in the stratosphere in summer, so I think January 2015 can be omitted.*

  Indeed, very little GW activity was found in the AIRS measurements during January, so a comparison is not included here. The weaker stratospheric winds in summer do not typically refract mountain waves to long vertical wavelengths visible to AIRS. As with the summertime radiosonde measurements, we wanted to briefly mention that the summertime modelling data exists in case it is useful for future researchers.

- *L194: Can you revise this sentence being more specific and naming the simulated conditions you are interested in, i.e. gravity waves. Then a large part of the wave spectrum can be expected to be close to reality but not the small scales.*

  Agreed, sentence revised.

- *L196: This sentence can be omitted.*

  Agreed, the paragraph has been revised.

- *L213: How can this have an effect at all on the data above 20 km? Is this due to the analysis performed later on?*

  Sentence removed. The reviewer is right, it doesn't affect our results at all. We were finding that, in the lower troposphere, passing synoptic systems could sometimes manifest as temperature perturbations to our background fit, so we didn't want to include them. For the model, the spectral analysis method is applied to the whole vertical range, so we didn't want to risk any spectral contamination from these features.

  Fortunately however, these considerations are no longer important for the revised study due to the consistent sampling and background removal method used for AIRS and the model-as-AIRS (see revised paper).

- *L235: Was the radiosonde data assimilated in the operational analyses? This should be mentioned here.*

  No, we have now mentioned this in the text.

- *L244: Do you mean a wind reversal in the meridional wind? Meridional wind direction is also changing at 30 km on 21st of July and end of July 2013.*

  Yes. Sentence removed, it was unnecessary. We only mentioned the 10th July 2013 case because it was the most significant meridional wind reversal in the data.

- *L271: Measurement errors and artifacts should be removed from the measurement data before doing the comparison. They are not physically meaningful and are too obvious in the profiles (especially in Fig. 3b, d but also in Fig. 3g above 15 km). Moreover, it would probably help to filter the small scale fluctuations in the sounding data that are well below the vertical resolution of the model data. Fig 3b, e would then look smoother and easier to compare to 3c, d.*

  This is another very useful comment, thanks. Once we removed measurement errors from the radiosonde error by hand for each flight individually (there were more than we had realised), we found that this had a significant improvement on the resulting agreement between the zonal winds in the model and the sondes. This also made the southward wind bias in the model clearer, which has helped to reaffirm our results. The figure and associated text has been updated.

- *L286-290: "slight southward directional bias", "more northward": please revise this paragraph. The wording is very circuitous. It's easier to just say that the model tend to slightly overestimate (underestimate) the southward (northward) winds in the mid-stratosphere. Because the mean difference is zero for the zonal component, this then not only tends in a small directional bias but also in a bias in the horizontal wind speed.*

  Fixed, thanks. We have used the reviewer's wording.

- *L287: the initial and boundary conditions*

  Fixed, thanks.

- *L293-L299: In my view, this paragraph is too speculative and can be omitted. Moreover, real time forecast of one to multiple days is different from short-term forecasts of up to 6h used here. Positional errors larger than then the horizontal grid-spacing of the model (everything smaller than that does not really an influence) are hopefully not contributing to the spread because they do not occur (or rarely occur and should then be removed from the sounding data before doing the comparison).*

  We agree about the unnecessary speculation and the positional spacing errors. We have updated the paragraph to remove them in the revised paper.

  The global forecasts that supplied the lateral boundary conditions were run forward from midnight on each day for each 24-hour period, providing hourly forecasts for that day (24 in total), so the time gap between boundary conditions was 1 hour rather than 6 hours as the reviewer stated. We have updated the text for clarity.

- *L301: Comparison is concluded and then starts again with discussing the surface winds. They are already included in L281 and local topographic effects are mentioned as possible reason. So L301- L306 can be removed. Detailed discussion of topographic differences between model and reality would include a comparison of the model topography to high-quality elevation data of the island. I don't think it's relevant for the rest of the paper.*

  Agreed, some this paragraph has now been incorporated in to the discussion above, and the rest has been removed.

- *L227: Can you specify what scales are meant? Vertically it's clear to me from Sec. 2 (8-9km) but not horizontally (3 times footprint size, e.g. > approx. 80km?).*

  We agree. The sentence (line 327) was confusing, so we have removed it.

- *L330: I cannot follow the reasoning of this sentence. Is this because the model runs without GW parametrization or why/how does the generation of long scale waves depend on the smallest scales?*

  The sentence was badly phrased and is not needed so it has been removed. We meant that, particularly for the case of mountain waves from small islands, the large horizontal-scale wave structures observed by AIRS (10s to 100s of km) originate from small-scale perturbations induced by topography (1s of km and lower).

- *Figure 4: It is probably better not to show the model data above 58 km where the damping layer is located. With the saturated amplitudes and vertical phaselines, it distorts the visual perception.*

  We have included a dashed line showing the model damping layer above 58km, as was done in Fig. 15. This was an oversight, we should have added this. For this example however, we still prefer to show the model data in Fig. 4 up to the model top for completeness. We have stated this in the text. In the revision, all subsequent analysis results are presented for measurements that are well below this damping layer (altitudes less than 45 km).

- *L464: applied to the*

  Fixed, thanks.

- *L470ff: Can you provide some values for a more quantitative comparison? For example, max. amplitude (and later on horizontal and vertical wavelength) at 20 and 40 km above the island and the downstream values you are referring to for both AIRS and model.*

  Agreed, the paragraph has been revised to be more quantitative.

- *L523: Really "measurement error of AIRS" or rather an uncertainty in the analysis and determination of the sign of m?*

  The reviewer is correct, we meant uncertainty in the determination of the sign of $m$ in the analysis. We have updated the text to reflect this.

- *Sec 4 "results": Section 3 contains already plenty of results. 4.1 could be just labelled 3.4 and so on*

  Agreed, fixed in response to major comment 2.

- *L643: I cannot follow. Isn't this a conclusion resulting from comparing model to model as AIRS? There is clearly more MF in the model outside the observational window of AIRS.*

  Agreed, the sentence was badly phrased and has been removed.

- *L979: It would be interesting to repeat the analysis with the output of the UKMO global configuration in the near future. Or was something similar already done in the past? If yes, you could add the reference here.*

  This is a good suggestion. We don't know of any studies comparing resolved gravity waves in the UKMO global model to 3-D AIRS observations (or similar) yet. The closest studies we can think of are probably Preusse et al. (2014), who analysed resolved waves in the IFS model and Holt et al. (2017), who compared GWs globally in a high resolution GEOS-5 simulation to 2-D AIRS observations.

  The key step in our study is the sampling of the model using the horizontal sampling and vertical weighting functions of the observations, which eliminates observational filter differences between them (Wright and Hindley, 2018). This is perhaps not done as routinely it should be in model-observations comparisons for GWs. Our study points to a way forward for direct like-for-like comparisons of observed and simulated GWs in these high-resolution configurations used to investigate GW generation from sub-grid scale orography. We have a planned study to compare resolved waves globally in re-analysis (probably ERA5) to 3-D AIRS observations, but this could be tricky because ERA5 assimilates AIRS radiances, so the comparison might not be straightforward.

**References**

B. Ehard, B. Kaifler, A. Dörnbrack, P. Preusse, S. D. Eckermann, M. Bramberger, S. Gisinger, N. Kaifler, B. Liley, J. Wagner, and M. Rapp. Horizontal propagation of large-amplitude mountain waves into the polar night jet. *Journal of Geophysical Research: Atmospheres*, 122(3):1423–1436, 2017. doi: 10.1002/2016JD025621.

L. A. Holt, M. J. Alexander, L. Coy, C. Liu, A. Molod, W. Putman, and S. Pawson. An evaluation of gravity waves and gravity wave sources in the southern hemisphere in a 7km global climate simulation. *Quarterly Journal of the Royal Meteorological Society*, 143(707):2481–2495, 2017. doi: 10.1002/qj.3101.

Riwal Plougonven, Alvaro de la Cámara, Albert Hertzog, and François Lott. How does knowledge of atmospheric gravity waves guide their parameterizations? *Quarterly Journal of the Royal Meteorological Society*, 146(728):1529–1543, 2020. doi: 10.1002/qj.3732.

P. Preusse, M. Ern, P. Bechtold, S. D. Eckermann, S. Kalisch, Q. T. Trinh, and M. Riese. Characteristics of gravity waves resolved by ECMWF. *Atmos. Chem. Phys.*, 14, 2014. doi: 10.5194/acp-14-10483-2014.

S. B. Vosper. Mountain waves and wakes generated by south georgia: implications for drag parametrization. *QJRMS*, 141 (692):2813–2827, 2015. doi: 10.1002/qj.2566.

C. J. Wright and N. P. Hindley. How well do stratospheric reanalyses reproduce high-resolution satellite temperature measurements? *Atmos. Chem. Phys.*, 18(18):13703–13731, 2018. doi: 10.5194/acp-18-13703-2018.

---

## Author Comment (AC3) · 21 Dec 2020

**Authors' response to comments from Reviewer #3 on "Stratospheric gravity waves over the mountainous island of South Georgia: testing a high-resolution dynamical model with 3-D satellite observations and radiosondes"**

N. P. Hindley et al.

**General Comment for all Reviewers**

We would like to thank the reviewers for the hard work in preparing their reviews of our submission. Their helpful suggestions have significantly improved the study. Several main improvements are listed below:

- In response to the reviewers' comments, we have significantly improved the way the model is sampled to create the model-as-AIRS dataset in our study. We realised that it is not enough to simply apply the AIRS horizontal resolution to the model: the AIRS horizontal sampling must be considered too. By sampling the model on the AIRS horizontal grid and taking into account the different sampling locations of each overpass, we are able to remove the background temperatures in exactly the same way for the AIRS and model-as-AIRS temperatures (we no longer use the nSG model runs for this). This ensures that our analysis steps allow for the spectral range of GWs visible to the AIRS and model-as-AIRS to be consistent.

- We also apply specified AIRS retrieval noise to the model-as-AIRS, which is characterised from a realistic AIRS granule. By applying the noise to the model-as-AIRS, we can separate out the effects of retrieval noise. This is important for the area-averaged results upwind and downwind of South Georgia.

- We now keep GW results measured in the full-resolution model very separate from the comparison between the AIRS and the model-as-AIRS. GW momentum flux in the full-resolution model is now calculated using wind perturbations, rather than from down-sampled temperature perturbations as before, and no comparison is made between GWMF in the model and the model-as-AIRS. This an important distinction because it is not possible to apply consistent horizontal sampling and background removal methods to both datasets, so no fair comparison can be made.

- The above steps have greatly improved the agreement between the AIRS GW measurements and the model-as-AIRS. As a result, the paper has been substantially reduced in size from 16 figures to 11 with a $\sim$20% reduction in text. Inconclusive or superfluous results and discussions have been removed, and a new Fig. 11 showing a case study of a short-$\lambda_H$ GW event has been added.

**Response to Reviewer #3: Major Comments**

1. *L169-176: The vertical resolution applied in the model is extremely coarse related to the horizontal resolution. The vertical grid-spacing is 0.6-2 km in the stratosphere, versus a horizontal grid-spacing of 1.5 km. This vertical grid-spacing in the stratosphere in not even sufficient to simulate a self-induced QBO in GCMs. More importantly, GCMs with explicit simulation of GWs (e.g., Watanabe et al., 2008, JAS: General aspects of a T213L256 middle atmosphere general circulation model) employ a vertical level spacing of 300-600 m throughout the middle atmosphere while the resolvable horizontal wavelengths in these models are of the order of 200 km. The necessity for a small enough vertical grid-spacing derives from the fact that the GWs resolved by the horizontal grid must not be spectrally biased in the*

*vertical to too large vertical wavelengths. Indeed a too coarse vertical resolution artificially prevents the GWs from reaching dynamic or convective instability and thus being dissipating by the model's turbulent diffusion scheme.*

This is probably the reviewer's main point, so we will break down our response to it below. While we agree and acknowledge that high vertical resolution is very important for accurate GW modelling in the stratosphere, we argue that the vertical and horizontal grids used in our model are more than sufficient to accurately resolve stratospheric GWs over South Georgia.

Sensitivity tests for vertical grids of 70, 118 and 173 vertical levels were performed by Vosper (2015). They found that the resolved zonal GW momentum fluxes from the surface to altitudes near 40 km for the 118 and 173 level simulations were highly similar. Both of these exhibited more realistic GWMF values in the lower stratosphere than the 70 level simulation. As can be seen from Fig. 2b of Vosper (2015), the 70 level simulation exhibits increased GWMF above 25 km. This is highly indicative of the issues relating to coarse vertical grids that the reviewer highlights here. Very little further improvement was found going from 118 vertical levels to 173, so the 118 level configuration was selected to reduce the computational load and permit the use of a fine horizontal grid over the island. We apologise that we neglected to mention this explicitly in the original submission. This is updated in the revised manuscript.

It is clear from our study however that a high horizontal spatial resolution is essential for accurate simulation of orographic GWs from the small mountainous island of South Georgia. It would always be nice to have more vertical levels, but we are limited by computational resources of what was feasible when the model was run.

*"This vertical grid-spacing in the stratosphere in not even sufficient to simulate a self-induced QBO in GCMs."*

The reviewer's comment here about the QBO is not relevant for our study. We do not need to simulate a realistic QBO over South Georgia near 54°S because (a) it is a primarily a tropical phenomenon and (b) we are only considering month-long time periods whereas the QBO has periods near two years.

It is worth mentioning however that the Kanto model of Watanabe et al. (2008) that the reviewer mentions did indeed resolve a QBO signal, but the period was close to 15 months rather than 28 months. Clearly this is far from perfect, despite the high number of vertical levels.

Ideally of course, one would always have many more vertical levels, but our point is that the 118 vertical levels used here are more than sufficient to resolve a realistic orographic GW field over South Georgia (Vosper, 2015). Once the correct horizontal sampling and resolutions were applied to the model (see revised paper), we actually found the agreement between the model and observations to be quite good. This suggests that the vertical grid issues described by the reviewer do not significantly affect our results.

*More importantly, GCMs with explicit simulation of GWs (e.g., Watanabe et al., 2008) employ a vertical level spacing of 300-600 m throughout the middle atmosphere while the resolvable horizontal wavelengths in these models are of the order of 200 km.*

Despite their fine vertical grids, such models with coarse horizontal grids cannot be used for our study. The reviewer mentions the Kanto model of Watanabe et al. (2008), and later the model of Becker and Vadas (2018). The orography of South Georgia would be, at most, equivalent to one or two horizontal grid points in these spectral models with T213 and T240 respectively, if even resolved at all. Despite their high number of vertical levels (which is very good), they would be unable to realistically resolve orographic GW generation and propagation over the island at the short horizontal scales necessary for our study.

In the vertical, the global models mentioned actually have quite coarse vertical grid spacing in the troposphere, which is a problem for accurate GW simulation over orography. The Becker and Vadas (2018) model for example has a vertical grid spacing of 600m and doesn't even go down to the surface, instead stopping at the boundary layer. The local-area model used here has a 10m vertical grid at the surface, and only begins to exceed 600m above 20 km

altitude. This grid configuration is more practical for low-level wind flow over orography and realistic mountain wave generation and propagation.

The local-area model configuration of the UM used here, with a horizontal grid of 1.5 km, will out-perform the resolution of these global spectral models, which is a key benefit of non-spectral models because local refinement is possible. The horizontal grid used is around 20 times finer than the global spectral models the reviewer mentioned, so if we wanted the same ratio between horizontal and vertical grids as these global models (to avoid spectral biasing), we would have to increase our number of stratospheric vertical levels by a similar amount. This is clearly impractical, and well beyond what was feasible when these simulations were performed due to computational limitations. The model runs used here are computed on a 800×600×118 grid. A simultaneous run on a 750 m horizontal grid was also performed on a 1600×1200×118 grid. These runs were highly computationally intensive. An increased number of vertical levels in the stratosphere would of course be advantageous, but the trade-off here is necessary to investigate the effect of fine horizontal grids, while remaining practical to run.

We acknowledge that no model is perfect, but some are useful. The sensitivity tests and assessment of simulated GWs in previous studies demonstrate that our chosen configuration can be useful for our study of mountain waves over South Georgia (Vosper, 2015; Vosper et al., 2016; Jackson et al., 2018).

*The necessity for a small enough vertical grid-spacing derives from the fact that the GWs resolved by the horizontal grid must not be spectrally biased in the vertical to too large vertical wavelengths.*

We agree with this comment, but we believe that it is not relevant for our study. Of course, a high vertical resolution is important for all GWs, but it is especially important for inertia GWs with relatively short vertical wavelengths.

However, our study is focused on stratospheric mountain waves over South Georgia during winter, where strong zonal winds at southern high latitudes can refract GWs to relatively long vertical wavelengths in the stratosphere (e.g. $\lambda_H \sim$12–25 km for zonal winds 40–80 m/s). These GWs can also have short horizontal scales $\lambda_H \lesssim$50–100 km. The aspect ratio of these GWs is far from those of inertia GWs, and can be considered mid-frequency or perhaps even high-mid frequency GWs.

One could easily argue the reviewer's point but for horizontal resolution in the Kanto model or the Becker and Vadas (2018) model. In those models, waves from small sources (like South Georgia) will be spectrally biased to long horizontal wavelengths because the horizontal grid is too coarse to accurately simulate them. Here, we accept any limitations of our model grid and have explicitly discussed the reviewer's concerns in the revised manuscript.

*Indeed a too coarse vertical resolution artificially prevents the GWs from reaching dynamic or convective instability and thus being dissipating by the model's turbulent diffusion scheme.*

We agree with this point and have added this into the revised manuscript.

2. *L176-178: I do not find this statement very conclusive. The grid-spacing of a model as such does not say anything about the scales that are reliably resolved. It is the dynamical core (spatial resolution, numerics) combined with the subgrid-scale diffusion (either explicit or implicit) that determines the reliable scales of a model.*

We agree, this was phrased badly. This has been corrected in the revised manuscript.

3. *L193-196: See my 2 previous major comments and consider reformulation.*

See our responses above. The paper has been significantly revised to make this clearer.

4. *L137-348: When the model data are interpolated to a 15 km grid, the Fourier components with horizontal wavelengths shorter than 30 km must be filtered out beforehand to avoid aliasing errors from the scales below the 15 km grid. Did the authors apply this spectral filtering before re-griding the model data (for model and model-as-AIRS)? If yes, please mention this point in the text for the sake of clarity. If not, the resulting aliasing could be an explanation for*

*the high power in the GW amplitudes and in the MF at horizontal wavelengths of 30-40 km (e.g. Fig. 16a). In that case you might consider a substantial revision and re-submission of the paper.*

Firstly, we must apologise. As the reviewer correctly suspected, Fig. 16 had an error with the normalisation of the amplitude-horizontal wavelength bin widths (which were different sizes) which caused anomalously high power at short $\lambda_H$. When the bins are correctly normalised for their width, this anomalous high power is removed. We are grateful to the reviewer for spotting this. The figure has been correctly revised, but in the end we decided not to include it in the revised paper for brevity. Also, as a result of the new model-as-AIRS processing mentioned above, the full-resolution model cannot be fairly compared to the AIRS and model-as-AIRS.

Perhaps more importantly however, the reviewer's comment made us think about the effect of horizontal sampling. We realised that it is not enough to simply apply the AIRS horizontal resolution, but the horizontal sampling pattern must also be applied to the model to ensure a fair comparison. This led to a major overhaul of the model-as-AIRS processing to accommodate realistic horizontal sampling of the AIRS instrument, including the different sampling locations during different overpasses. Once this aspect was correctly applied, the agreement between AIRS and the model-as-AIRS was significantly improved (see revised paper). This is major improvement, and we are grateful to the reviewer for highlighting it - even though that maybe was not their intention!

With this is mind however (although this is not relevant any more), the reviewer is not correct that Fourier components with horizontal wavelengths shorter than 30km should be filtered out here to avoid aliasing problems. When AIRS samples the atmosphere, the horizontal sampling pattern samples where it samples, including any effects of aliasing. There is no post-hoc removal of Fourier components in AIRS when it samples the real atmosphere, so it would be inconsistent to apply such things to the model. For a fair comparison, we should simply sample at the same locations as AIRS and allow aliasing effects to take their course in both datasets.

Finally, we would also like to direct the reviewer to the new Fig. 11, where GWs with very large amplitudes with $\lambda_H \sim$30–40 km are observed in AIRS and simulated in the model directly over the island. These waves can only be resolved in the model due to the fine horizontal resolution, and in this example they are essentially validated by AIRS observations due to favourable viewing geometry.

Even after the original Fig. 16 was fixed, the figure did not make the cut for the revised manuscript, because its results were not very useful. Instead, the new Fig. 11 shows that although the high power at $\lambda_H \sim$30–40 km in the original Fig. 16 was in error, mountain waves with large amplitudes can be found at these short scales directly over the island, if the resolution is high enough to support them.

We should also mention that the improved sampling approach has also benefited the AIRS results. In the original submission, we applied a 3×3 horizontal boxcar filter to the AIRS data to suppress any spurious pixel-scale noise, as per the approach of previous studies (e.g. Wright et al., 2017). But after close inspection we found that some GWs directly over the island at the pixel-scale in AIRS were actually realistic (see new Fig. 11), so it was a mistake to smooth these out. In the revised paper, we do not apply this which results in a much improved agreement between GWMF in AIRS and the model-as-AIRS directly over the island.

5. *L399: The authors should not only mention that model-as-AIRS produces too small amplitudes compared to AIRS, but also that the GW-phases of the MWs over the Island differ significantly in the two data sets (Figs. 4 and 5). Moreover, the slopes of the phase lines from x=100 km to 600 km in Fig. 4 differ in sign(!); that is, these GWs must propagate in different directions when comparing model-as-AIRS to AIRS. Please mention and discuss these dissimilarities.*

   Figures 4 and 5 show different overpass times 14 hours apart. This was stated in the caption, but we have clarified it in the main text to help to make this clearer. Also, the new revised figure (now Fig. 5) now shows the horizontal area around the island which makes this non-orographic wave (NGW) clearer to see. The slope of the phase lines that

the reviewer refers to is identified as a part of a transitory NGW, and not related to the mountain wave field over the island. 14 hours later in the next overpass, this wave is no longer present. This non-orographic (or at least, clearly not from South Georgia) wave does not appear in the model-as-AIRS. The apparent under-representation of NGWs this is one of the results discussed in the revised paper.

6. *L429-430: See my comments above: The horizontal structures in model-as-AIRS and AIRS are at best qualitatively similar over the mountain; they are dissimilar farther downstream. Please describe your comparison of results from model-as-AIRS and AIRS consistently with your high-quality figures.*

See response above. These overpass times are 14 hours apart.

We should mention though that in the text of the revised paper we now make a clearer distinction between qualitative and quantitative comparisons as a result of the reviewer's point, so this has been very constructive. We are also grateful for the reviewer's complement about the quality of the figures, we hope they will find the revised figures equally good.

7. *Fig. 7: How did you apply averaging over the GW scales when calculating the MF. Furthermore, the regions of phases going upward with increasing x in Fig. 4c and f should give rise to a reversal from westward to eastward MF in Fig. 7c. Please clarify.*

The GWMF values estimated via Eqn. 1 are assumed to be averaged over one GW wavecycle (Ern et al., 2004). We have GW wavelength measurements for the dominant (largest spectral amplitude) wave at every location in the 3-D volume from the 3DST (Hindley et al., 2019), so we have an estimated GWMF value everywhere. The isosurfaces then show cuts through these values.

We acknowledge that this is not ideal, but this is standard practise for estimating GWMF from measured GW amplitudes and wavelengths. Later (and in the revised paper), we take the area average over a well-defined 3-D volume, which again is not ideal but provides a reasonable average over GW scales, as the reviewer later suggests.

Regarding the regions of upward sloping phase in the model-as-AIRS, recall again that Figs. 4 and 7 show different overpass times 14 hours apart. We also direct the reviewer to the revised figure in the new Fig. 5. Once the horizontal sampling and resolution is correctly applied, we can see that this upward sloping phase with increasing $x$ is no longer apparent. Again, we are very grateful to the reviewer for prompting this revision.

8. *L489-493: The wave refraction argument can be applied for either upward propagating GWs (negative vertical wavenumber) or downward propagating GWs (positive vertical wavenumber). Here you apply this argument even though the longer vertical wavelengths that you expect for a westward MW in an increasing stratospheric eastward jet show up in your plot with reversed sign. How do you explain the reversal from negative to positive vertical wavenumber at 20-30 km in Fig. 6d? Why is there a noisy mixture of positive and negative vertical wavenumbers in Fig.6h? These wavenumber (wavelength) results need to be revisited.*

We agree with the issues highlighted in this comment. Note however that when we try to measure GWs with long vertical wavelengths, only a small amount of horizontal directional error is required to flip the horizontal direction because the phase fronts are aligned so near to the vertical.

We thought it might be useful to discuss these changes in sign of the vertical wavenumber, just in case they were physical, but as the reviewer points out they are probably simply due to horizontal directional error in measurement of very long vertical wavelengths.

In the revised figure (new Fig. 6), we do not discuss positive or negative vertical wavenumbers and instead we accept these regions as experimental error, and have clearly mentioned this is the text. Our revised results in later sections (see new Fig. 8) however indicate that the area-average GWMF results are not significantly affected by this error.

9. *L510-515 and L528-L532: This discussion relates to my previous comment. Please give a hint on why you possibly have positive vertical wavennumbers in AIRS. One possibility is that the background wind in the lower atmosphere shows accelera- tion/decceleration which can cause the phase lines of MWs sloping upward/downward in time-height cross-sections. Another possibility is the generation of secondary GWs from MW breaking causing downward propagating GWs (which are no longer MWs). See also Vadas and Becker (2018, JGR Atmos.: Numerical Modeling of the Excitation, Propagation, and Dissipation of Primary and Secondary Gravity Waves during Wintertime at McMurdo Station in the Antarctic), as well as Vadas et al. (2018).*

See response above. Both of these suggestions are possible, and in the original submission we wondered if we could be could measuring secondary GWs or some kind of reflection. But upon reflection, the data do not fully support an investigation into this, so in the revised paper we instead accept any directional errors as measurement error, rather than discussing the possibility of 2GWs here. It is something that could be considered in future, but is beyond the scope of what can be addressed in this paper.

10. *L599: Note that the wind in the lower troposphere is crucial for MW generation, while the wind at higher altitudes facilitates propagation (strongly eastward) or dynamical instability (weakly eastward or westward). Again, it is unclear how dynamical instability (including critical levels) are handled by the model, given its coarse vertical level spacing in the stratosphere and the lack of information about subgrid-scale processes.*

See our response above regarding vertical level spacing in the model and the sensitivity tests conducted in Vosper (2015). We respectfully disagree with the reviewer here. Of course it would always be good to have more vertical layers, but we do not agree that the model has a "coarse" vertical grid spacing that could significantly affect our results in this specific study. We have however included these possible issues in the text for discussion.

The model configuration used here is well-described in Vosper (2015). A comprehensive description of the dynamical core of the Met Office Unified Model is provided in Wood et al. (2014) and citation therein. Please consult these descriptions for information about subgrid-scale processes.

11. *L619-623: This is another example of a very speculative discussion about suspicious features in the model data. Are stationary, non-orographic GWs indeed present around the island in the global model? Are these waves artificial? Please clarify.*

Agreed. This discussion, and other speculative discussions like it, have been removed in the revised paper. NGWs are not expected to be well-simulated in the local area model due to the coarse resolution of the global forecast. Even if there were realistic NGWs in the global forecast, it is not clear how well these waves would be "transferred" into the local area domain due to the 1hr time integration used for the lateral boundary conditions. These aspects are described in the revised paper.

12. *L638-645: How is the simulated very large MF at scales close to the horizontal grid scale possibly related to the coarse vertical level spacing and, in addition, to insufficient parameterization of dissipation processes in the stratosphere below the sponge layer? Your model results would imply that the vast majority of MW momentum flux resides at horizontal scales not even observable by AIRS. Hence, according to your model results, observations from AIRS are essentially useless to estimate the orographic GW MF from small Islands that is missing in global models? Please clarify.*

Sensitivity tests by Vosper (2015) suggest that stratospheric vertical grid spacing in the 118-level configuration chosen here has no significant effect on the resolved GWMF, because increasing the number levels to 173 made no significant difference.

The large GWMF at short horizontal scales ($\lambda_H \sim 50\,\text{km}$) found in the full-resolution model is because the vast majority of GWs in the model are mountain waves from the small island of South Georgia, which is less that 37 km

across. The characteristic horizontal wavelengths of mountain waves are primarily determined by the horizontal size of the obstacle, as the reviewer later mentions. Largest mountain waves amplitudes occur directly over the island, where these short wavelengths are found. This results in very large GWMF measurements via Eqn. 1. We direct the reviewer to the new Fig. 11 in the revised manuscript, where the large amplitudes of GWs at short $\lambda_H$ are found and validated by AIRS measurements in one example with favourable viewing geometry.

*Hence, according to your model results, observations from AIRS are essentially useless to estimate the orographic GW MF from small Islands that is missing in global models? Please clarify.*

No single instrument can observe the full GW spectrum, but no other instrument can yet provide global 3-D measurements needed to constrain GWMF. We can only compare models to the observations we have.

Regarding the "usefulness" of the 3-D AIRS measurements, the horizontal scales of GWs from South Georgia can be resolved equally well or better in 3-D AIRS measurements than they can be resolved in the Kanto model of Watanabe et al. (2008) or the model of Becker and Vadas (2018), so these measurements are useful.

13. *Fig. 15: This is a very nice figure (like most of the other figures)! I cannot see the grey lines mentioned in the caption. My comment is this: The AIRS curves nicely indicate wave dissipation from about 25 km on. This wave dissipation is not reflected by the model results. Therefore, this figure supports my major concerns about the model: Too large vertical level spacing combined with possible shortcomings in subgrid-scale parameterization leads to insufficient dissipation.*

Thanks! We have made the grey lines thicker.

We have included these possible issues in the discussion of the results in the revised paper.

14. *L858–868: Ditto.*

Ditto above. We have included these possible issues in the discussion of the results in the revised paper.

15. *Fig. 16a: This figure suggests that you would get a reversed power spectrum of the wave amplitude with respect to the horizontal wavenumber, i.e., increasing (instead of decreasing) power with increasing wavenumber? Please check. If this is so, this would imply that the model results at these small scales are not reliable at all.*

Once again we apologise for the error in the bin-width normalisation in this figure. The revised figure did not feature such a prominent distribution, but we have not included this analysis in the paper because it was not useful. The average GWMF at a given wavelength and amplitudes is more or less determined by Eqn. 1, and the differences between the AIRS and the model-as-AIRS in the revised manuscript were less significant. This analysis has instead been replaced with the new Fig. 11, which provides a good indication of some of the smallest horizontal scales that occur over the island.

But we should consider that the largest GWMF values for mountain waves over South Georgia do indeed occur at short horizontal wavelengths (large wavenumbers) up to near the characteristic size of the island ($\lambda_H \sim$30–40 km).

We should also remember that the model configuration used here is not some global model with a full spectrum of resolved GWs in the middle atmosphere and a well-behaved GW power law spectrum. This is a small regional model (with under-represented NGWs) over South Georgia in which the single largest source of GWs is flow over mountainous orography of the island. This creates a spectrum of GWs that is unique to the physical size and characteristics of the island. If we applied the same analysis to a large oceanic region, we would expect a much better behaved power law spectrum.

16. *L927: Ditto*

Ditto above.

17. *L978-981: As mentioned above, it is not just horizontal grid-spacing (and model numerics, as you mention in L992) that determines how well a model simulates GWs. You have to consider the vertical grid spacing as well. Most importantly, inviscid fluid dynamics cannot handle GW breakdown and wave-mean flow interaction. You need an explicit dissipative process for non-transient wave-mean flow interaction (see the non-acceleration theorem, Lindzen's GW saturation theory, or the classical McFarlane paper about orographic GW parmameterization). That is why the parameterization of subgrid-scale processes (turbulent diffusion) is very important in any GW-resolving circulation model (e.g., Becker and Vadas, 2018).*

    Agreed, thanks. We have added this to the revised paper. See our points above relating to the vertical grid spacing.

**Minor Comments**

- *L73-75: I agree with this statement. However, the authors miss the opportunity to put the orographic GW momentum flux from South Georgia into the context of the general circulation in SH winter.*

  The introduction has been revised in the resubmission.

- *L92: Please point out that the model used in this study is a real-date regional model that is forced by a global forecast model via lateral boundary conditions. Therefore, this regional model is not "essentially free running".*

  Added, thanks for the correct terminology! This is an important distinction that allows the model to be compared directly to observations.

- *L135: Please be specific whether the vertical resolution relates to wavelength or grid-spacing.*

  Fixed, thanks.

- *L179: The vertical resolution of the global model is presumably too coarse to represent inertia GWs in the stratosphere. This could be the reason why the regional model misses these waves when compared to the AIRS data.*

  AIRS is very unlikely to see inertia GWs (IGWs) either, due to the deep vertical weighting functions of the AIRS instrument, so this is not likely to affect our comparison.

- *L205-214: This paragraph is hard to follow and distracts a bit from the very good writing otherwise in the paper.*

  Agreed, this has been fixed now.

- *Figure 2: Please plot the zonal wind with the same color coding as the meridional wind (blue for minus, red for plus)? Can you use a nonlinear color scale to make the accelerations and deccelerations of the tropospheric wind visible? Note that the wind in the lower troposphere determines the forcing of orographic GWs.*

  Agreed, the colour scale has been fixed now for consistency. We tried a non-linear colour scale for this figure but we found that, visually, it placed too much emphasis on whether the wind was positive/negative at low speeds and less emphasis on the large wind speeds in the stratosphere, which are important for GW propagation and refraction to long vertical wavelengths visible to AIRS and the model-as-AIRS. The surface winds are reasonably strong for most of the campaign (i.e. reasonable orographic forcing), but the stratospheric wind speeds can have a first order effect on the measured GWMF in AIRS due to GW refraction effects (Hindley et al., 2020), so it is important to highlight this.

- *L245: The radiosonde observations do not provide a horizontal average over the domain covered by the model. Please reformulate correspondingly.*

  We did not do a horizontal average over the model domain, we traced each individual radiosonde through a 4-dimensional model space $(x, y, z, t)$ and evaluated the model temperatures along that path using linear interpolation (as stated in l.245-250). These model-as-Sondes paths were then compared to the radiosonde observations.

The wind contours in Fig. 3a are provided for illustration of the local wind conditions only.

- *L266: Figure 3 is very well composed. However, Fig. 3g illustrates that the simulated winds are not in good agreement with the radiosonde data. Rather, the agreement is only reasonable. The mean meridional wind in Fig. 2d is predominantly southward from 30 to 60 km and is of the order of a few -10 m/s. The corresponding wind in Fig. 3d shows a bias of about 10 m/s.*

  Thank you for spotting this. We assume the reviewer means Fig. 3g. Reviewer #2 also mentioned that this needed reformulating and you are both right. We have since removed obvious anomalies from the radiosonde measurements and the resulting comparison provides a much clearer result.

  There is indeed (on average) a southward bias in the model winds compared to the radiosonde observations. This now forms one of our key results of the paper, since a small corresponding northward bias in simulated stratospheric GWMF is also found later.

- *L279: Short-timescale variability would average out when comparing time-averaged wind profiles. I suggest to accept these discrepancies and to discuss the possible implications for orographic forcing and vertical propagation of GWs in the model.*

  The discussions regarding short-timescale variability have been removed in the revised paper. As the reviewer suggests, we instead accept these discrepancies and focus on what we can say with the time-averaged results. The implications for a persistent southward bias in the model may be a resulting northward bias in the simulated GWMF, which is discussed in the new results section.

- *L284-L290: See my comment with respect to L226 above.*

  Thanks, this discussion is revised now. We assume the reviewer means L266 above.

- *L300-306: The differences between model and radiosonde data are not minor. Invoking the "climatological level" of simulated wind in case studies of orographic GWs, which are subject to extreme intermittency, does not sound conclusive.*

  Agreed, we now consider the discrepancies between the model winds and the radiosonde observations more carefully in the revised paper. Note however that because the zonal wind is (usually) so much stronger than the meridional, a meridional bias in wind speed only corresponds to a small directional bias, and the magnitude of the mean wind in the model and the sondes is reasonably close. This is what we were trying to say (albeit badly), but this has been thoroughly revised now in the revised paper.

- *L360-363: These sentences are hard to understand (e.g., "vertical resolution for that vertical layer"). Please reformulate.*

  In this sentence, we meant horizontal layer, sorry. The AIRS vertical resolution changes with altitude. So for a given layer in the retrieval, this layer will have its own vertical resolution that must be applied to the model. The description of the model-as-AIRS process has been substantially revised for clarity.

- *L377: This statement is not conclusive. What about model errors?*

  We meant in terms of time separation between the AIRS overpass and the model timestep. The text has been improved in the revised paper.

- *L470-471: A "reasonable apparent similarity" is not observed when considering the dissimilarity of individual phase lines between the two data sets in Fig. 6a and e.*

  See above. As discussed, Figs. 4 and 5 show two different overpasses 14 hours apart.

- *page 25: Why is this new section called "Results". The previous Section 3 contained plenty of results, not just methodology.*

  Agreed, the sections were not well arranged in the original submission. This was a major comment from Reviewer #2. This has been substantially revised in the resubmission, and more care has been taken to separate methods from results and results from discussions.

- *L535-538: This description of secondary GW generation from MW breaking does not seem consistent with the aforementioned papers by Vadas and coauthors.*

  The sentence has been removed.

- *L539-542: This sounds very vague. I recommend to simply discard speculations of this kind. Furthermore, if you want to discuss secondary GWs in your model, then you need to consider how the model simulates dynamical instability and dissipation of resolved GWs and, hence, the necessary body forces for secondary GW generation. As discussed earlier, the very coarse vertical resolution of the model combined with the lack of knowledge about the built-in (presumably implicitly numerical) dissipation casts doubts on whether the model reliably simulates body forces from GW dissipation in the stratosphere.*

  The sentence has been removed, and we have included in the model description the following:

  "It should be mentioned that this although this vertical grid spacing is sufficient to resolve wintertime orographic waves over South Georgia, the vertical grid spacing of around 1.5–2 km in the upper stratosphere is unlikely to accurately simulate body forces under wave breaking that are necessary for secondary GW (2GW) generation (e.g. Becker and Vadas, 2018)."

- *L553: Note that this equation holds strictly only for a monochromatic GW or, at best, for a narrow spectrum of GWs. As soon as you have a broad spectrum, the wavelengths to be used at the rhs become arbitrary. More importantly: I am missing the Reynolds-type average of (T')**2 (see my comment on Fig. 7 above). Please clarify.*

  Ern et al. (2017) showed that this equation is valid for GWs visible to AIRS, which is all we apply it to in the model and the model-as-AIRS in the revised paper. This relation is not perfect, but the mid-frequency approximation on which is derived is certainly not a "narrow spectrum" of GWs regarding those visible in satellite observations.

  We are well aware that it is valid for a monochromatic GW only. This is why we only apply it to the dominant (largest spectral amplitude) wave at each location, before taking the area-average. This is standard practice in observational GW studies. We do not pretend that this is an ideal method, but for observational of GWs where only temperature perturbations are available, to our knowledge there exists no other reliable method to estimate GWMF from measured GW temperature amplitudes and wavelengths.

  The terms in Equation 1 (where $T'$ is defined as the temperature perturbation amplitude of a GW) are consistent with previous studies involving estimates of GWMF from temperature perturbations (e.g. Ern et al., 2004). The 3DST method of Hindley et al. (2019) delivers spatially-localised phase-invariant "packet" amplitude for a GW at a given length scale (or wavelength here) which is equivalent to the average perturbation amplitude usually describe for wind perturbations (see new Eqn. 1 in the revised manuscript for the model wind perturbations).

- *L582-585: This information clarifies my previous comment at least for Fig. 8-10. Given the size of the island relative to the model domain and the GW scales in AIRS and model-as-AIRS, you use the area-average to compute the MF. I think that is the right choice here. How would the resulting MF contribute to the zonal mean parameterized in global models?*

  Agreed, the area-average approach is probably the only reasonable choice we can do regarding GW scales.

  *How would the resulting MF contribute to the zonal mean parameterized in global models?*

A similar approach would need to be taken to that employed by Hindley et al. (2019) and Hindley et al. (2020), who used a latitudinal band approach to show that ~75% of the total GWMF during winter near 60°S was found over the ocean, including over small islands. If we have an area-average GWMF value for one segment of a latitudinal band, we can get it's contribution to the zonal mean by considering the fraction (zonally) of the latitude band that it occupies. A future regional study aiming to break these fractions down further into individual islands to try and constrain the contribution of each is currently planned.

- *L588: I can not see the red markers in Fig. 8-14.*

  We have made these markers bigger for clarity.

- *L681: Again, I disagree that "observed and simulated wave fields are quite similar". As mentioned earlier, there are even qualitative differences.*

  See above. Figs. 4 and 5 are 14 hours apart in time.

- *L700: What about spontaneous emission from the upper tropospheric jet stream? See Plougonven and Zhang, 2014, Rev. Geophys: Internal gravity waves from atmospheric jets and fronts.*

  Agreed, reference added.

- *L827-829: These differences could simply result from errors in the background wind (driven by the global model) in the lower troposphere, leading to errors in orographic forcing of MWs in the model. I believe the authors should discuss this role of the tropospheric winds somewhere in the paper.*

  A southward wind bias in the model is now thoroughly discussed in the revised paper, in particular relating to an observed northward bias in model GWMF, which may be related as the reviewer suggests.

- *L845-849: See my previous major and minor comments regarding the obvious and possible shortcomings of the model.*

  See our responses above.

- *L928-935: It is hard to follow these arguments. Of course, MWs can be forced by non- stationary background winds. Furthermore hourly fluctuations of the background wind would correspond to non-orographic GWs that you force at the lateral boundaries. Your discussion of possible reasons for the model shortcomings (see also L936-940) do not mention the concerns that I raised above.*

  Agreed, these arguments were poor. This discussion has been removed, because the improved sampling method for generating the model-as-AIRS resulted in significant improvements in this regard.

- *L946-947: This sentence seems not logical. Consider reformulation.*

  This whole discussion has been revised for clarity.

- *L949-950: "not so commonly"? Which observations are you aware of that show this feature of very large MW amplitudes in the stratosphere at very small horizontal scales?*

  We direct the reviewer to the new Fig. 11, where very large amplitude mountain waves at horizontal scales $\lambda_H$ ~30–40 km are simulated and observed directly over the island in the model and AIRS observations. The measurement of these short-$\lambda_H$ waves in AIRS is only possible in this example due to favourable viewing geometry of the specific overpass. The measured wavelength agrees well between all three datasets. The fact that the measured AIRS amplitudes agree reasonably well between the AIRS and model-as-AIRS suggests that the wave amplitudes in the full-resolution model ($T' \sim 45$ K near 45 km altitude) may have occurred in reality.

  GWs like this do not appear in global high resolution models like those of Watanabe et al. (2008) or Becker and Vadas (2018) because the horizontal resolution is too coarse to resolve them, however fine their vertical grids.

We did not find them in AIRS observations in the original submission because we foolishly smoothed them out with the 3×3 horizontal boxcar filter. The discovery of this example has shifted the conclusions of the paper considerably.

- *L954-955: Why should intermittency of MW forcing give rise to shorter horizontal wavelengths than stationary forcing? Usually, the structure of the topography determines the spectrum that can be forced.*

  This discussion was weak and has been superseded in the revised paper.

- *L963-964: Now you argue that an "overly-stable wind vector" could give rise to the high power of MWs at very small scales in models.*

  This discussion was weak and has been superseded in the revised paper. We thought that perhaps the MW field was too idealised in the model compared to reality.

  However, the high power of MWs at small scales near 30–40 km in the model has been shown to be realistic in the revised paper.

- *L993-997: I think that here you reveal a misconception about semi-implicit time stepping in circulation models. Semi-implicit time stepping is applied to suppress the artificial generation of very fast anelastic waves and sound waves; otherwise, smaller time steps would be required for numerical stability. In any event, the time step is always small enough to properly resolve the time scales of anelastic GWs that are well described by the representation of the model equations in gridspace.*

  Thanks for the information. Apologies if I have misunderstood, but I'm not sure if this is consistent with the description of the process employed in the Unified Model as described by (Shutts and Vosper, 2011). It's probably my misunderstanding, so don't worry. In any case, I think the use of a relatively short time step here (30s) means that these issues are not likely to be significant for our configuration.

- *L999-1000: Here you finally come up with a critical comment about the lack of dissipation in the model stratosphere.*

  We have added this as a possibility in the revised paper.

- *L1002: You did not run the model at very high spatial resolution. Your vertical resolution in the stratosphere is much coarser than even in GW-resolving global models run at moderate horizontal resolution (e.g., Sato et al. 2012., JAS: Gravity Wave Characteristics in the Southern Hemisphere Revealed by a High-Resolution Middle-Atmosphere General Circulation Model). Again, your coarse vertical level spacing is certainly not adequate to support your very high horizontal resolution.*

  We meant that we ran the model at a high *horizontal* spatial resolution. As mentioned, the sensitivity tests in Vosper (2015) did not reveal any issues with our chosen vertical grid for mountain wave simulations.

- *L1005: As long as we do not solve the (viscid) Navier-Stokes equations with a resolution of 1 cm in the troposphere, the performance of our circulation models will always depend on how unresolved (subgrid-scale) dynamical processes are parameterized.*

  We are specifically referring here to GW drag parameterisations, such as parameterised GW generation from flow over small sub-grid scale islands, not the accurate parameterisation of GW dissipation processes (which will always need parameterising), but the reviewer's point is fair. We do not claim here that increased horizontal resolution GCMs will be able to remove GW drag parameterisations altogether, we simply suggest that as resolution improves we may be able to reduce reliance on parameterised GW drag for small orographic sources, which are almost impossible to fully constrain by observations.

- *L1022-1023: Yes! See my comments above.*

  We have expanded on this significantly in the revised manuscript.

- *L1030: You did not perform sensitivity experiments using the same model with different horizontal resolutions.*

  Sensitivity tests on this model configuration (and the actual July 2013 run) are described by Vosper (2015). For the runs used here, the same model was run on both a 1.5 km and 750 m horizontal grid was simultaneously. Jackson et al. (2018) reported that the general characteristics of the mountain wave field were the same between each run. See also Vosper et al. (2016), where the balance between resolved and parameterised GW drag for varying horizontal grid resolutions is investigated directly using an identical model set up.

**Typos/Suggestions**

All suggested typos and edits have been added, thank you.

**References**

E. Becker and S. L. Vadas. Secondary gravity waves in the winter mesosphere: Results from a high-resolution global circulation model. *Journal of Geophysical Research: Atmospheres*, 123(5):2605–2627, 3 2018. doi: 10.1002/2017JD027460.

M. Ern, P. Preusse, M. J. Alexander, and C. D. Warner. Absolute values of gravity wave momentum flux derived from satellite data. *J. Geophys. Res.*, 109:D20103, 2004. doi: 10.1029/2004JD004752.

M. Ern, L. Hoffmann, and P. Preusse. Directional gravity wave momentum fluxes in the stratosphere derived from high-resolution airs temperature data. *Geophy. Res. Lett.*, 44(1):475–485, 2017. doi: 10.1002/2016GL072007.

N. P. Hindley, C. J. Wright, N. D. Smith, L. Hoffmann, L. A. Holt, M. J. Alexander, T. Moffat-Griffin, and N. J. Mitchell. Gravity waves in the winter stratosphere over the southern ocean: high-resolution satellite observations and 3-d spectral analysis. *Atmospheric Chemistry and Physics*, 19(24):15377–15414, 2019. doi: 10.5194/acp-19-15377-2019.

N. P. Hindley, C. J. Wright, L. Hoffmann, T. Moffat-Griffin, and N. J. Mitchell. An 18-year climatology of directional stratospheric gravity wave momentum flux from 3-d satellite observations. *Geophysical Research Letters*, 47(22), November 2020. doi: 10.1029/2020gl089557.

D. R. Jackson, A. Gadian, N. P. Hindley, L. Hoffmann, J. Hughes, J. King, T. Moffat-Griffin, A. C. Moss, A. N. Ross, S. B. Vosper, C. J. Wright, and N. J. Mitchell. The south georgia wave experiment: A means for improved analysis of gravity waves and low-level wind impacts generated from mountainous islands. *Bulletin of the American Meteorological Society*, 99(5):1027–1040, 2018. doi: 10.1175/BAMS-D-16-0151.1.

G. J. Shutts and S. B. Vosper. Stratospheric gravity waves revealed in NWP model forecasts. *Quart. J. Roy. Meteor. Soc.*, 137:303–317, 2011. doi: 10.1002/qj.763.

S. B. Vosper. Mountain waves and wakes generated by south georgia: implications for drag parametrization. *QJRMS*, 141 (692):2813–2827, 2015. doi: 10.1002/qj.2566.

S. B. Vosper, A. R. Brown, and S. Webster. Orographic drag on islands in the nwp mountain grey zone. *Quarterly Journal of the Royal Meteorological Society*, 142(701):3128–3137, 2016. doi: 10.1002/qj.2894.

S. Watanabe, Y. Kawatani, Y. Tomikawa, K. Miyazaki, M. Takahashi, and K. Sato. General aspects of a T213L256 middle atmosphere general circulation model. *J. Geophys. Res.*, 113:D12110, 2008. doi: 10.1029/2008JD010026.

N. Wood, A. Staniforth, A. White, T. Allen, M. Diamantakis, M. Gross, T. Melvin, C. Smith, S. Vosper, M. Zerroukat, and J. Thuburn. An inherently mass-conserving semi-implicit semi-lagrangian discretization of the deep-atmosphere global non-hydrostatic equations. *Quarterly Journal of the Royal Meteorological Society*, 140(682):1505–1520, 2014. doi: 10.1002/qj.2235.

C. J. Wright, N. P. Hindley, L. Hoffmann, M. J. Alexander, and N. J. Mitchell. Exploring gravity wave characteristics in 3-d using a novel s-transform technique: Airs/aqua measurements over the southern andes and drake passage. *Atmospheric Chemistry and Physics*, 17(13):8553–8575, 2017. doi: 10.5194/acp-17-8553-2017.

---

## Referee Report (RR1)

**2nd Review on Hindley et al. Stratospheric gravity waves over the mountainous island of South Georgia: testing a high-resolution dynamical model with 3-D satellite observations and radiosondes**

In the extensively revised version of the manuscript, the authors have greatly improved the model data processing that now results in a persuasive comparison between AIRS and the local-area simulations. My previous comments were adequately addressed and the structure of the manuscript improved. I only have a few technical corrections for this revised version of the manuscript.

Technical corrections:

L22: … agreement with the model-as-AIRS.

L123: no need to introduce 2GW abbreviation because it is never used elsewhere in the paper

L267: (Fritts and Alexander, 2003; Ern et al, 2004).

L267: Eq1 is the general definition and not restricted to mid-frequency GWs. Applies for Eq. 2 later in the paper.

L269: extracted → separated

L277: during periods where surface zonal winds are low (Figs. 2a,b)…

L460: Subsection "6.3.1 3DST analysis" could be introduced to have not only one subsection in 6.3

L781: could be "Summary and Conclusions"

L811: 30-40 km

Fig.7: You can add that the assumption of westward propagation is based on model results, e.g. In this example, westward propagation has been assumed based on sequential model results in order to…

---

## Author Response (AR2)

**Authors' response to minor comments on "Stratospheric gravity waves over the mountainous island of South Georgia: testing a high-resolution dynamical model with 3-D satellite observations and radiosondes"**

N. P. Hindley et al.

Once again we would like to thank the reviewers for their detailed and useful comments in their first and subsequent reviews. It is due to their care and attention that the manuscript is significantly improved, so we would like to extend our gratitude to them for taking the time to prepare their reviews.

**Response to Reviewer #1**

*This is an improved version of the manuscript, where several aspects have been modified or corrected. Among them, some merely speculative arguments without substantiation. I have only one final request which may not have been clear enough in my first review. The redundancy issue regarding the 3DST method (former lines 449-453) should be clarified. The calculated GWs amplitudes cannot be considered reliable if a non-orthogonal basis is used. See e.g. Stockwell (2007) https://doi.org/10.1016/j.dsp.2006.04.006 as discussed for the Stockwell transform. So please explain this issue or cite an earlier discussion of this topic for the 3DST method.*

The reviewer raises an important point. We apologise for not addressing this in our previous reply. We can provide a discussion of this issue below, and we have included this information in the revised paper.

The basis functions for the $S$-transform are normalised Gaussian-windowed sinusoidal functions. These basis functions are specified by frequency $\alpha$ in the spectral domain (which are at the discrete Fourier frequencies, if the discrete Fourier transform is used), and translation $\tau$ in the spatial domain. The width of the Gaussian window is scaled as $1/\alpha$, so all basis functions in the $S$-transform can be fully described by the two values $\alpha$ and $\tau$.

But for a discrete-sampled timeseries, we do not need to consider all values of $\alpha$ at all translations $\tau$ to compute an accurate $S$-transform spectrum that can be readily inverted to recover the original signal with no loss of information.

Indeed, Stockwell (2007) showed that for carefully chosen sets of orthogonal basis functions for the $S$-transform , the $S$-transform could be successfully computed and inverted to recover the original signal for only a fraction of the calculations. This approach is highly efficient and can result in a very significant reduction in computation time by removing this redundancy. This forms the basis for the Discrete Orthonormal $S$-transform (DOST), which can be found widely online.

Our application of the $S$-transform however is different. Here we use the $S$-transform to exhaustively "probe" the internal structure of the measured wave field in the maximum possible detail. We are not interested in inverting the $S$-transform to recover the original signal, we are trying to "break down" and localise the spectral components of the wave field as much as possible at all locations.

To do this, our method considers all $S$-transform basis functions for all values of $\alpha$ at all translations $\tau$. Of course, this approach is highly redundant, and much slower to compute, but it is actually advantageous for our purposes because it allows us to retain the maximum precision in $\alpha$ and $\tau$ at all locations. By inspecting the resulting $S$-transform $S(\alpha,\tau)$ spectrum, we are able to detect changes in the dominant (largest spectral amplitude) up to the scale of the data sampling. The scaling parameter for the Gaussian window, as discussed in Hindley et al. (2016, 2019), helps to adjust this capability, but the precision remains as fine as the spatial sampling and Fourier frequency sampling. Note that this is not the same

as oversampling, since we do not sample between sample points or between Fourier frequencies, as we might do with a continuous wavelet transform.

Regarding the accuracy of GW amplitudes, the reviewer is correct that if we used sets of orthogonal basis functions (which would be much faster), the sets of coefficients must be considered together to recover accurate GW amplitudes. But because we consider all possible basis functions singly and one at a time, we are able to extract amplitudes from the $S$-transform coefficients directly. Of course, not all basis functions will provide a good estimate of the amplitude of a specific GW, so at each translation $\tau$ we select the coefficient of $S(\alpha,\tau)$ with the largest spectral amplitude and take the GW amplitude and frequency from that.

Naturally, there are disadvantages to extracting GW amplitudes directly from the $S$-transform spectrum, such as the issue of measuring GWs at "in-between" frequencies in the Fourier sampling, or the effect of spectral leakage for GW packets (see appendix of Hindley et al. (2019)). The maximum spectral amplitude method therefore is nominally an underestimate of GW amplitudes, depending on frequency sampling, but can work well in most cases, as demonstrated in Sect 3.5 of Hindley et al. (2019).

The measured GW amplitudes from our $S$-transform method are thoroughly tested for real and synthetic waves in Hindley et al. (2016, 2019) and Wright et al. (2017), and in preparatory work for the present study. This is very interesting topic however and we welcome the reviewer to contact us directly if there is a way we could improve GW amplitude measurement or reduce computation time.

**Response to Reviewer #2**

*In the extensively revised version of the manuscript, the authors have greatly improved the model data processing that now results in a persuasive comparison between AIRS and the local-area simulations. My previous comments were adequately addressed and the structure of the manuscript improved. I only have a few technical corrections for this revised version of the manuscript.*

We once again thank the reviewer for their careful and useful review of the revised submission.

**Technical corrections:**

- *L22: ... agreement with the model-as-AIRS.*
  Fixed, thanks.

- *L123: no need to introduce 2GW abbreviation because it is never used elsewhere in the paper*
  Fixed, thanks.

- *L267: (Fritts and Alexander, 2003; Ern et al, 2004).*
  Fixed, thanks.

- *L267: Eq1 is the general definition and not restricted to mid-frequency GWs. Applies for Eq. 2 later in the paper.*
  Fixed, thanks!

- *L269: extracted > separated*
  Fixed, thanks.

- *L277: during periods where surface zonal winds are low (Figs. 2a,b)...*
  Fixed, thanks.

- *L460: Subsection "6.3.1 3DST analysis" could be introduced to have not only one subsection in 6.3*

  Fixed, thanks. The new sections headings are "6.3.1 3DST measurements of GW amplitude and wavelength" and "6.3.2 Zonal and meridional momentum fluxes".

- *L781: could be "Summary and Conclusions"*

  Agreed, thanks.

- *L811: 30-40 km*

  Fixed, thanks.

- *Fig.7: You can add that the assumption of westward propagation is based on model results, e.g. In this example, westward propagation has been assumed based on sequential model results in order to...*

  Good suggestion. Added, thanks.

**Response to Reviewer #3**

*The authors provided a thoughtful and extensive revision. This paper is now of very high quality, and it is packed with insight and useful information. My former comments on the submitted manuscript were accounted to a very high degree. I recommend publication of revised manuscript in its present form.*

*I wish to comment on the authors' answer to my previous critical remarks on vertical resolution and subgrid-scale processes. I still believe that the paper could have mentioned how dissipation processes are represented in the model in terms of either numerical filters (implicit methods) and/or physics-based parameterizations (explicit methods like, for example, the Smagorinsky scheme).*

Our apologies for not including this information in the paper. The Met Office Unified Model (UM) uses a semi-Lagrangian dynamical core, so there is some implicit numerical diffusion as a result of the interpolation methods used to determine the departure points. In the local-area simulations used here, the "Smagorinsky-type" 3-D subgrid horizontal turbulence scheme is used, as the reviewer correctly suggests. Useful descriptions of the 3-D Smagorinsky scheme used in high-spatial resolution configurations of the UM, and some comparisons with other schemes, can be found in Pearson et al. (2014), Boutle et al. (2014) and citations therein. We have included this information in the revised paper.

*Below are some miscellaneous remarks that the authors may wish to consider in their page proofs:*

- *L172: exceeded*

  Changed to "travelled beyond", thanks.

- *L276-277: incomplete sentence*

  Fixed, thanks.

- *L283: What is "southward component of the characteristic GW pattern" ?*

  Apologies, this was unclear. For a typical mountain wave pattern from an isolated island source, a chevron-shaped "bow-wave" pattern is formed. This wave field has GWMF directed opposite to the prevailing wind, but it also has both northward and southward components of GW momentum to the north and south. The text has been revised to make this clearer.

- *L317,320,336: "time step" -> "snapshot" (your model time step is certainly much smaller than your snapshot interval of 1 hour)*

  Agreed, thanks, this is an important distinction. We have changed instances like this to "hourly model output".

- *L396: A GW with a horizontal wavelength of 50-150 km is usually not termed a "large-scale GW". Consider to use "small-to-medium-scale GW" instead.*

  Apologies, we simply meant "larger horizontal scales". The text has been fixed, thanks.

- *L583: define "NGWs" or just write "non-orographic GWs"; same in Sec. 9.2*

  Fixed, thanks.

- *L696: "upwind"?*

  Changed to "measured upwind of the island to the west".

- *L764-768: The simulation result should depend on the time step. If such a dependence is found, the numerical solution has not yet converged with respect to the time step.*

  The reviewer is correct, simulation results do certainly depend on model time step in the general case, but we did not find this to be an issue in the Unified Model for time step choices close to the value considered here (30 seconds).

  As discussed by Vosper (2015): "the results of the local area simulation were not sensitive to choices around this time step, which is expected because of the second-order accuracy of the time integration scheme in the ENDGame dynamical core (Wood et al., 2014). ... The improved stability of the dynamical core [over previous versions] allows the model to be run with reduced temporal off-centring in the semi-implicit method, such that less weight is placed on the future time step. As discussed by Shutts and Vosper (2011), off-centring can cause heavy damping in time, damping gravity-wave motion when long time steps are used. The reduced off-centring allows longer time steps to be used without causing significant damping of gravity waves that are spatially well-resolved."

  We apologise if we have misunderstood. We encourage the reviewer to contact us directly to clarify this or any other aspects of the Met Office model.

**References**

I. A. Boutle, J. E. J. Eyre, and A. P. Lock. Seamless stratocumulus simulation across the turbulent gray zone. *Monthly Weather Review*, 142(4):1655–1668, March 2014. doi: 10.1175/mwr-d-13-00229.1.

N. P. Hindley, N. D. Smith, C. J. Wright, D. A. S. Rees, and N. J. Mitchell. A two-dimensional stockwell transform for gravity wave analysis of AIRS measurements. *Atmospheric Measurement Techniques*, 9(6):2545–2565, June 2016. doi: 10.5194/amt-9-2545-2016.

N. P. Hindley, C. J. Wright, N. D. Smith, L. Hoffmann, L. A. Holt, M. J. Alexander, T. Moffat-Griffin, and N. J. Mitchell. Gravity waves in the winter stratosphere over the southern ocean: high-resolution satellite observations and 3-d spectral analysis. *Atmospheric Chemistry and Physics*, 19(24):15377–15414, 2019. doi: 10.5194/acp-19-15377-2019.

K. J. Pearson, G. M. S. Lister, C. E. Birch, R. P. Allan, R. J. Hogan, and S. J. Woolnough. Modelling the diurnal cycle of tropical convection across the 'grey zone'. *Quarterly Journal of the Royal Meteorological Society*, 140(679):491–499, 2014. doi: https://doi.org/10.1002/qj.2145.

G. J. Shutts and S. B. Vosper. Stratospheric gravity waves revealed in NWP model forecasts. *Quart. J. Roy. Meteor. Soc.*, 137:303–317, 2011. doi: 10.1002/qj.763.

R. G. Stockwell. A basis for efficient representation of the s-transform. *Digital Signal Processing*, 17(1):371 – 393, 2007. ISSN 1051-2004. doi: http://dx.doi.org/10.1016/j.dsp.2006.04.006.

S. B. Vosper. Mountain waves and wakes generated by south georgia: implications for drag parametrization. *QJRMS*, 141 (692):2813–2827, 2015. doi: 10.1002/qj.2566.

N. Wood, A. Staniforth, A. White, T. Allen, M. Diamantakis, M. Gross, T. Melvin, C. Smith, S. Vosper, M. Zerroukat, and J. Thuburn. An inherently mass-conserving semi-implicit semi-lagrangian discretization of the deep-atmosphere global non-hydrostatic equations. *Quarterly Journal of the Royal Meteorological Society*, 140(682):1505–1520, 2014. doi: 10.1002/qj.2235.

C. J. Wright, N. P. Hindley, L. Hoffmann, M. J. Alexander, and N. J. Mitchell. Exploring gravity wave characteristics in 3-d using a novel s-transform technique: Airs/aqua measurements over the southern andes and drake passage. *Atmospheric Chemistry and Physics*, 17(13):8553–8575, 2017. doi: 10.5194/acp-17-8553-2017.